# Coulomb branch surgery: Holonomy saddles, S-folds and discrete symmetry gaugings

**Elias Furrer[♯], Horia Magureanu[♭]**

[♯] *School of Mathematics, University of Birmingham,*
*Watson Building, Edgbaston, Birmingham, B15 2TT, United Kingdom*

[♭] *Mathematical Institute, University of Oxford,*
*Andrew-Wiles Building, Woodstock Road, Oxford, OX2 6GG, United Kingdom*

*E-mail:* e.r.furrer@bham.ac.uk, magureanuhoria@gmail.com

ABSTRACT: Symmetries of Seiberg–Witten (SW) geometries capture intricate physical aspects of the underlying 4d $\mathcal{N} = 2$ field theories. For rank-one theories, these geometries are rational elliptic surfaces whose automorphism group is a semi-direct product between the Coulomb branch (CB) symmetries and the Mordell-Weil group. We study quotients of the SW geometry by subgroups of its automorphism group, which most naturally become gaugings of discrete 0- and 1-form symmetries. Yet, new interpretations of these *surgeries* become evident when considering 5d $\mathcal{N} = 1$ superconformal field theories. There, certain CB symmetries are related to symmetries of the corresponding $(p, q)$-brane web and, as a result, CB surgeries can give rise to (fractional) S-folds. Another novel interpretation of these quotients is the folding across dimensions: circle compactifications of the 5d $E_{2N_f+1}$ Seiberg theories lead in the infrared to two copies of locally indistinguishable 4d SU(2) SQCD theories with $N_f$ fundamental flavours. This extends earlier results on holonomy saddles, while also reproducing detailed computations of 5d BPS spectra and predicting new 5d and 6d BPS quivers. Finally, we argue that the semi-direct product structure of the automorphism group of the SW geometry includes mixed 't Hooft anomalies between the 0- and 1-form symmetries, and we also present some new results on non-cyclic CB symmetries.

# 1 Introduction

Supersymmetric quantum field theories have received great attention over the years due to their remarkable exact solutions and tractable non-perturbative effects. This is particularly evident in the case of four-dimensional $\mathcal{N} = 2$ supersymmetric gauge theories, where the Seiberg–Witten (SW) geometry [1, 2] fully solves the intricate dynamics of the strong-coupling regimes. Besides offering a detailed description of the vacuum moduli space [3–6], SW geometry has been shown to capture various other physical aspects such as flavour symmetries [7–10], spectra of BPS states [11–14], renormalisation group flows [15–21], and even global properties of these models [22–24].

## 1.1 Automorphisms of the SW geometry

In this paper, we extend the applications of SW geometry through a careful analysis of its symmetries. We direct our attention to 4d rank-one $\mathcal{N} = 2$ theories, which also include Kaluza–Klein (KK) theories. There, the total space of the SW curve fibred over the complex one-dimensional Coulomb branch, or $U$-plane, can be identified as a *rational elliptic surface* (RES) [10, 22]. In many instances, quotients of a RES by subgroups of the full automorphism group $\mathrm{Aut}(\mathcal{S})$ lead to other well-defined RES. As a result, a link between seemingly distinct SW geometries is realised. Rather remarkably, these *surgeries* lead to deep physical insights, going beyond discrete symmetry gaugings, contrary to naive expectations.

The full automorphism group $\mathrm{Aut}(\mathcal{S})$ for such surfaces $\mathcal{S}$ takes the form of a semidirect product [25–27],

$$\mathrm{Aut}(\mathcal{S}) = \mathrm{MW}(\mathcal{S}) \rtimes \mathrm{Aut}_\sigma(\mathcal{S}) . \tag{1.1}$$

The first component, $\mathrm{MW}(\mathcal{S})$, is the *Mordell–Weil group* of $\mathcal{S}$. Notably, this group has been argued to encode the 1-form symmetry of the theory whose SW geometry is described by the RES $\mathcal{S}$ [22] (see also [18, 28–30]). The conjecture was further refined in [24], where a procedure for gauging 1-form symmetries directly from the SW geometry was proposed and checked in all rank-one examples. This procedure involves compositions of so-called *isogenies*, which are generated as quotients of the underlying rational elliptic surface by subgroups of the Mordell-Weil group. Isogenies find another application in generating Galois covers [31], which are constructions that relate the spectra of BPS states between two theories. This topic will be the focus the our upcoming work [32].

The second factor appearing in (1.1) is the subgroup of automorphisms preserving the zero section $\sigma$ of the SW elliptic fibration, $\mathrm{Aut}_\sigma(\mathcal{S})$. It consists of symmetries acting non-trivially on the Coulomb branch, which, in the simplest case, are the cyclic symmetries $U \mapsto e^{\frac{2\pi i}{n}} U$. These discrete 0-form symmetries can have various physical origins, such as residual $R$-symmetries, charge conjugation, or remnants

of higher-form symmetries if the theory is obtained by toroidal compactification from a higher-dimensional theory. Gaugings of these discrete 0-form symmetries are naturally associated with a *folding* of the $U$-plane, which arises as a quotient of $\mathcal{S}$ by subgroups of $\mathrm{Aut}_\sigma(\mathcal{S})$, similar to how isogenies are defined.

Coulomb branch foldings corresponding to cyclic subgroups of $\mathrm{Aut}_\sigma(\mathcal{S})$ have been extensively studied within the classification programme of 4d $\mathcal{N} = 2$ rank-one SCFTs [18, 20, 33]. Yet, new interpretations of such foldings are revealed when considering the SW geometries of KK theories resulting from the toroidal compactification of higher dimensional SCFTs. In this regard, we discover intricate relationships between four and five-dimensional theories, some of which have previously been suggested by detailed computations of BPS spectra [34–37]. Additionally, for the five-dimensional theories admitting a $(p, q)$-brane web description we show how CB surgery relates to brane-web folding [38, 39]. Notably, $U$-planes contain more information than the underlying brane web, allowing thus for symmetries not visible at the level of the brane web.

Another new result of this work involves the semi-direct product structure of (1.1), which captures a very subtle feature of the underlying theory. This structure defines a non-trivial action $\mathrm{Aut}_\sigma(\mathcal{S})$ on $\mathrm{MW}(\mathcal{S})$. In the simplest interpretation of these groups as 0- and 1-form symmetries, the most natural interpretation of said action is that of a mixed anomaly between the two symmetries. We show that this is indeed the case and formulate the precise condition giving an anomaly. Lastly, we should also mention that the group $\mathrm{Aut}_\sigma(\mathcal{S})$ is in many cases non-cyclic. We study the action of these groups on the Coulomb branch geometry and discuss its physical interpretation. In the rest of the introduction, we expand on the results presented above.

## 1.2 Coulomb branch folding

As previously alluded to, it frequently occurs that quotients of the SW geometry by subgroups of $\mathrm{Aut}(\mathcal{S})$ lead to new rational elliptic surfaces, which are thus compatible with new 4d $\mathcal{N} = 2$ models. The simplest such constructions are the discrete gaugings of 0- and 1-form symmetries. Nevertheless, a single quotient can often times be given multiple physical interpretations: the same RES can describe several theories, typically involving the fine-tuning of mass deformation parameters. We will refer to the interpretations different from discrete gaugings as *geometric transitions*. Table 1 summarises the various interpretations of these surgeries.

The simplest class of 4d $\mathcal{N} = 2$ theories for which we can illustrate the various surgery constructions are $\mathcal{N} = 2$ supersymmetric QCD with gauge group SU(2) and $N_f \leq 4$ massive fundamental hypermultiplets. Their $U$-planes have been analysed in great detail in [1, 2, 4, 6, 22, 40–43]. Our reasoning naturally extends to any 4d rank-one $\mathcal{N} = 2$ SCFT, as well as to the family of Kaluza–Klein theories obtained

|  | Isogenies – MW($\mathcal{S}$) | Foldings – Aut$_\sigma(\mathcal{S}$) |
|---|---|---|
| **Discrete gaugings** | 1-form sym. gauging | 0-form sym. gauging |
| **Geometric transitions** | Galois covers | (Fractional) S-folds<br>Holonomy saddles |

Table 1: Summary of the physical action of quotients by the subgroups of Aut($\mathcal{S}$).

by circle compactifications of the 5d $\mathcal{N}=1$ Seiberg $E_n$ theories, which we denote by $D_{S^1}E_n$ [44, 45].

**Cyclic symmetries of the $U$-plane.** Coulomb branch 0-form symmetries are generally manifestations of discrete subgroups of the U(1)$_\mathbf{r}$ $R$-symmetry. Gaugings of these discrete symmetries have been extensively studied in the classification programme of rank-one SCFTs [18, 20, 33]. Nevertheless, the symmetries of the KK theories have not been explored yet to the same extent. Although five-dimensional $\mathcal{N}=1$ SCFTs only possess a SU(2)$_R$ R-symmetry, their effective 4d $\mathcal{N}=2$ KK theories can include discrete subgroups of the additional U(1)$_\mathbf{r}$ symmetry, which manifest themselves on the Coulomb branch as accidental 0-form symmetries. We analyse in detail the possible $\mathbb{Z}_k$ symmetries of the $D_{S^1}E_n$ theories and, using geometric constraints from their underlying surfaces $\mathcal{S}$, classify all allowed $\mathbb{Z}_k$ symmetries. These are listed in Table 2. Note that the maximal cyclic symmetries of $D_{S^1}E_n$ for $n \geq 3$ are precisely the centres of the simply connected flavour symmetry groups $E_n$, *i.e.* $\mathbb{Z}_{9-n} \cong Z(E_n)$.

| $D_{S^1}E_n$ | $Z(E_n)$ | $F_\infty$ | # sing | $\mathbb{Z}_2$ | $\mathbb{Z}_3$ | $\mathbb{Z}_4$ | $\mathbb{Z}_5$ | $\mathbb{Z}_6$ |
|---|---|---|---|---|---|---|---|---|
| $E_0$ | – | $I_9$ | 3 |  | ✓ |  |  |  |
| $E_1$ | $\mathbb{Z}_2$ | $I_8$ | 4 | ✓ |  | ✓ |  |  |
| $\widetilde{E}_1$ | U(1) | $I_8$ | 4 |  |  |  |  |  |
| $E_2$ | U(1) | $I_7$ | 5 |  |  |  |  |  |
| $E_3$ | $\mathbb{Z}_6$ | $I_6$ | 6 | ✓ | ✓ |  |  | ✓ |
| $E_4$ | $\mathbb{Z}_5$ | $I_5$ | 7 |  |  |  | ✓ |  |
| $E_5$ | $\mathbb{Z}_4$ | $I_4$ | 8 | ✓ |  | ✓ |  |  |
| $E_6$ | $\mathbb{Z}_3$ | $I_3$ | 9 |  | ✓ |  |  |  |
| $E_7$ | $\mathbb{Z}_2$ | $I_2$ | 10 | ✓ |  |  |  |  |

Table 2: Allowed $\mathbb{Z}_k$ automorphisms of the $U$-planes of the $E_n$ KK theories which lead to well-defined quotients of the underlying SW geometries. The red check is a symmetry that does not appear at the level of the $(p,q)$-brane web of the 5d $E_4$ theory.

From the perspective of the underlying RES, all $U$-plane foldings described in Table 2 are allowed, in the sense that the quotients lead to new well-defined rational surfaces. Remarkably, the allowed – and therefore foldable – symmetries of the $U$-plane almost coincide with the symmetries of the underlying $(p, q)$ brane webs of the respective 5d SCFTs, where the folding procedure corresponds to five-dimensional S-folding [38, 39, 46, 47]. In fact, the interpretation of CB foldings as S-folds was already apparent from the F-theoretic construction of these models [48–53], having been already discussed in [18] for purely four-dimensional SCFTs. Nevertheless, the link between the CB geometry and the brane-webs of the five-dimensional models sheds new light on these S-folds.

The astute reader might have already noticed that Table 2 includes a $\mathbb{Z}_5$ symmetry which exists on the Coulomb branch of the 4d $D_{S^1} E_4$ theory, which does not manifest as a symmetry of the 5d brane web. This symmetry is peculiar for the following reason. The fixed point of the $\mathbb{Z}_5$ symmetry is a type $II$ singular fibre in Kodaira's classification (see Appendix A), and thus a CB folding would correspond to a $\mathbb{Z}_5$ discrete gauging of the $H_0$ Argyres-Douglas theory [3, 4]. However, the existence of this discrete gauging was argued in [33] to be rather speculative.

Five-dimensional $\mathbb{Z}_k$ S-folds involve cutting the 5-brane web into $k$ slices, keeping only one of these slices, and inserting a specific 7-brane at the fixed point of the $\mathbb{Z}_k$ action [38]. As recently argued in [39], such quotients can be generalised by keeping $k-1$ slices and only removing one, while inserting appropriate 7-brane configurations at the fixed point. In the 4d KK theory, we understand this procedure of *fractional folding* as a composition of a folding and an *unfolding*, the latter being a formal inverse to a folding. This reproduces the list of rank-one examples found in [39].

**Folding across dimensions.** The mathematical formalism of rational elliptic surfaces identifies a theory by fixing one singular fibre of the SW geometry: the fibre at 'infinity', $F_\infty$ [10, 22]. This singular fibre specifies the UV physics of a field theory. An operation that can change the fibre at infinity of a given RES $\mathcal{S}$ is the *quadratic twisting*, which is a particular reparametrisation of the Weierstraß invariants of the surface. Such operations can thus give relations between theories of seemingly different origins.

Of particular interest to us will be combinations of CB foldings and quadratic twistings. These constructions are perfectly exemplified by the foldings of the KK theories, which are described by $F_\infty = I_n$, where $I_n$ are the multiplicative type Kodaira singularities (see Table 4 in Appendix A). Coulomb branch $\mathbb{Z}_k$-foldings of these models would naturally suggest the mapping $I_n \to I_{n/k}$, but our interest lies in the slightly modified version $I_n \to I^*_{n/k}$. In this latter example, the new fibre at infinity describes a 4d SQCD model, leading thus to a link between 4d and 5d theories. Specifically, we show that this scenario can only occur for $\mathbb{Z}_2$-foldings, being closely related to the concept of *holonomy saddles* [54, 55]. In particular,

such foldings imply that the circle compactification of particular 5d SCFTs leads to two copies of locally indistinguishable 4d SQCD theories in the infrared. This phenomenon was studied in [35, 36] for the 5d pure SU(2) gauge theory, and we extend this analysis to theories with $2N_f$ flavours,

$$\text{5d SU(2)} + 2N_f \text{ fund } \xrightarrow{\mathbb{Z}_2} 2 \text{ copies of 4d SU(2)} + N_f \text{ fund .} \qquad (1.2)$$

The fact that there are always two copies is related to the rank of the gauge group SU(2) [36], and it can be derived from the rationality condition of the underlying surfaces. We prove the existence of these $\mathbb{Z}_2$-foldings (1.2) by finding an explicit dictionary between the 4d and 5d mass parameters, and obtain the universal relation

$$U_{2N_f+1} = \sqrt{\mathcal{K}_{N_f} u_{N_f} + \mathcal{P}_{N_f}} \, , \qquad (1.3)$$

with $U_{2N_f+1}$ the CB parameter of the $D_{S^1}E_{2N_f+1}$ theory, while $u_{N_f}$ is the SQCD Coulomb branch parameter, and $\mathcal{K}_{N_f}, \mathcal{P}_{N_f}$ are normalisation factors.

This analysis also leads to a conjecture on the BPS spectrum of these 5d theories. For massless SQCD it generalises the results of [34, 36, 37], while for generic masses it gives a multitude of new correspondences between 4d and 5d BPS quivers. Of particular interest is the $N_f = 3$ case, which would allow the computation of the BPS spectrum for the non-toric BPS quivers of the $E_7$ theory. Furthermore, we expect additional relations between the 5d $\mathcal{N} = 1^*$ gauge theory and 4d $\mathcal{N} = 2^*$ SYM, as well as between the 6d E-string theory and superconformal 4d $N_f = 4$ SCQD. These results would not only predict 6d BPS quivers [24], but also lead to insights into their spectra of BPS states.

## 1.3 Outline

The remainder of the paper is organised as follows. In Section 2, we present the physical and mathematical ingredients that we aim to combine: we give a short introduction to the $U$-plane of 4d rank-one $\mathcal{N} = 2$ theories and the formalism of rational elliptic surfaces $\mathcal{S}$. We further introduce the two components of the automorphism group of $\mathcal{S}$. Section 3 discusses the constraints and the explicit construction of discrete gaugings in 4d and 5d theories, the relation to symmetries of $(p, q)$ brane webs, as well as the construction of fractional foldings. We further comment on non-cyclic symmetries of the Coulomb branch. Section 4 describes the folding across dimensions and the relation to holonomy saddles. In Section 5, we conjecture how mixed 't Hooft anomalies are encoded in $\text{Aut}(\mathcal{S})$. We conclude and give some directions for future work in Section 6.

Several details and definitions of elliptic surfaces as well as modular forms are given in Appendix A. In Appendix B, we provide a more general description of how $U$-plane foldings can be obtained from the perspective of fundamental domains. Appendix C discusses further aspects of non-cyclic Coulomb branch symmetries by

analysing several examples. Appendix D finally lists some explicit expressions for the Seiberg–Witten curves. We also include a list of Mathematica notebooks with explicit curves for all $D_{S^1}E_n$ theories, as well as various other routines used throughout this work in the GitHub repository [56].

## 2 Symmetries of the Coulomb branch

In this section, we introduce the Coulomb branch of 4d rank-one $\mathcal{N} = 2$ theories and initiate the study of CB symmetries. Section 2.1 includes a gentle introduction to the formulation of the SW geometry as a rational elliptic surface. Meanwhile, in Section 2.2 we discuss the automorphisms of elliptic surfaces and the relations to isogenies and foldings.

### 2.1 Introducing the $U$-plane

The low-energy dynamics on the Coulomb branch of four-dimensional $\mathcal{N} = 2$ super-symmetric quantum field theories is famously solved by the Seiberg–Witten geometry [1, 2]. SW geometry provides a beautiful synthesis of the notions of duality, elliptic curves and modular forms. See [57–59] for some reviews of the topic.

**The Seiberg–Witten solution.** For any rank-one 4d $\mathcal{N} = 2$ theory, the low-energy effective U(1) field theory on the Coulomb branch $\mathcal{B}$ can be expressed in terms of a scalar field $a$. The CB itself is parametrised by a gauge invariant operator $U \in \mathcal{B}$, being thus one-complex dimensional for this class of models.[1] The celebrated Seiberg–Witten solution expresses the scalar $a$ and its magnetic dual $a_D$ as periods of a differential form $\lambda$ over one-cycles of an elliptic curve,

$$a = \oint_{\gamma_A} \lambda \, , \qquad a_D = \oint_{\gamma_B} \lambda \, . \tag{2.1}$$

These periods determine the low-energy effective gauge coupling $\tau$ as well as the exact prepotential $\mathcal{F}$ as

$$a_D = \frac{\partial \mathcal{F}}{\partial a} \, , \qquad \tau = \frac{\partial a_D}{\partial a} \, . \tag{2.2}$$

The SW geometry is then defined as the elliptic fibration of the SW curve over the Coulomb branch, $E \to \mathcal{S} \to \mathcal{B}$. The total space of this fibration, $\mathcal{S}$, forms a *rational elliptic surface* (RES). We leave to Appendix A more details about the structure of such surfaces (see also the nice review [60], as well as [61–65]).

The elliptic fibres of the SW geometry can be brought into Weierstraß normal form,

$$y^2 = 4x^3 - g_2(U, \boldsymbol{m})x - g_3(U, \boldsymbol{m}) \, , \tag{2.3}$$

---

[1]For strictly 4d theories, the coordinate $U$ is more commonly denoted by $u$. We also consider circle compactifications of 5d theories, and use $U$ for both 5d and 4d Coulomb branch parameters.

where $g_2$ and $g_3$ are functions of the CB parameter $U$, as well as of various mass parameters $\boldsymbol{m}$, and possibly dynamical scales and gauge couplings that can appear in the theory. These fibres can become singular above special loci on the CB, where the low-energy effective U(1) description breaks down. The types of singularities that can appear on the $U$-plane are covered by Kodaira's classification of singular fibres, as we discuss in more detail in Appendix A. In short, the various types of singular fibres can be identified based on the orders of vanishing of the Weierstraß invariants $g_2, g_3$ and the discriminant $\Delta$. We should mention that a configuration of singular fibres for a given theory will typically change as the mass parameters vary.

The low-energy effective coupling $\tau$ (2.2) is realised as the complex structure of the elliptic curve. This identifies the $\mathcal{J}$-invariant $\mathcal{J}(U)$ of the elliptic curve (2.3) with the modular $J$-invariant $J(\tau)$:[2]

$$\mathcal{J}(U(\tau)) = J(\tau) \ , \tag{2.4}$$

For fixed parameters $\boldsymbol{m}$, the family of elliptic curves (2.3) is parametrised by the coordinate $U$, and so the $\mathcal{J}$-invariant of the corresponding RES becomes a rational function $\mathcal{J}(U)$ on the base $\mathcal{B}$. This promotes $U$ to a function of $\tau$ through (2.4). Solving this equation has been crucial for the study of topological correlation functions [42, 66–76], and for obtaining BPS quivers directly from the SW geometry [14, 22, 31, 77, 78]

While in the generic case of arbitrary $\boldsymbol{m}$ there are no analytical solutions to (2.4), in many cases, the duality and modular properties of the Coulomb branch of any 4d $\mathcal{N} = 2$ theory $\mathcal{T}$ can be found by rewriting (2.4) as $P_{\mathcal{T}}(X) = 0$, where

$$P_{\mathcal{T}}(X) = \left(g_2(X, \boldsymbol{m})^3 - 27g_3(X, \boldsymbol{m})^2\right) J(\tau) - g_2(X, \boldsymbol{m})^3 \ , \tag{2.5}$$

is a polynomial specified by the RES of the theory $\mathcal{T}$ [6, 14, 43].

**Classes of 4d $\mathcal{N} = 2$ theories.** The quantum field theories of interest throughout this paper are the 4d $\mathcal{N} = 2$ field theories with a one complex dimensional Coulomb branch. We focus on two particular classes of such theories. The possibly simplest class is 4d $\mathcal{N} = 2$ supersymmetric QCD with gauge group SU(2) and $N_f \leq 4$ fundamental hypermultiplets. The SW curves of these theories have been studied in detail in [1, 2, 4, 6, 22, 40–43], and we present them explicitly in Appendix D. The second family of interest is represented by the Kaluza–Klein (KK) theories obtained by a circle compactification of the UV completion of the 5d $\mathcal{N} = 1$ SU(2) theories, with $N_f = n - 1$ fundamental flavours [44, 45]. These theories are the simplest examples of 5d SCFTs, having $E_n$ flavour symmetry, where the low flavour-rank

---

[2]See Appendix A for the precise definitions of both $\mathcal{J}(U)$ and $J(\tau)$.

groups are given by:

$$\begin{aligned}
E_1 &= \mathrm{SU}(2) \ , & \widetilde{E}_1 &= \mathrm{U}(1) \ , \\
E_2 &= \mathrm{SU}(2) \times \mathrm{U}(1) \ , & E_4 &= \mathrm{SU}(5) \ , \\
E_3 &= \mathrm{SU}(3) \times \mathrm{SU}(2) \ , & E_5 &= \mathrm{Spin}(10) \ ,
\end{aligned} \qquad (2.6)$$

while for $n = 6$, 7 and 8, $E_n$ are the exceptional Lie groups. We will denote their circle compactifications by $D_{S^1} E_n$.

These 5d rank-1 SCFTs are geometrically engineered in $M$-theory compactifications on non-compact Calabi-Yau threefolds, which are the canonical line bundles over del Pezzo surfaces $dP_n$ or the Hirzebruch surface $\mathbb{F}_0$. More precisely, the $E_n$ theories with $n = 2, \ldots, 8$ are engineered on local $dP_n$, while the $E_1$ theory corresponds to local $\mathbb{F}_0 \cong \mathbb{P}^1 \times \mathbb{P}^1$. The $E_0$ theory is engineered on local $\mathbb{P}^2$ and does not have a gauge theory interpretation, but can be obtained from the $\widetilde{E}_1$ theory, which itself corresponds to local $\mathbb{F}_1 \cong dP_1$.

From the $D_{S^1} E_n$ theories, by deformations, geometric-engineering limits and RG flows one can obtain 4d SQCD, as well as the Argyres-Douglas (AD) [3, 4] and Minahan-Nemeschansky (MN) [7, 8] superconformal field theories in 4d. Their SW geometries have been determined and analysed in great detail [15, 17, 79–90]. We list the explicit curves of the toric $D_{S^1} E_n$ theories in Appendix D. See also the Mathematica notebook [56] for the toric as well as the non-toric curves.

The SW geometry of any rank-one 4d $\mathcal{N} = 2$ SQFT is a rational elliptic surface, being partially characterised by its configuration of singular fibres. One particular dedicated singular fibre is the fibre at infinity $F_\infty$, which corresponds to the point $U = \infty$ on the CB [10, 22]. This singularity fixes the 'UV physics', and will be singled out from the rest of the singular fibres. As such, we use the notation $(F_\infty; F_{v_1}, \ldots)$, for describing a RES with fibre at infinity $F_\infty$, and bulk singular fibres $(F_{v_1}, \ldots)$. For the asymptotically free gauge theories with $N_f$ fundamentals for instance, the point at infinity is characterised by the one-loop beta function coefficient, leading to the following identifications [22]:

$$\begin{aligned}
F_\infty &= I^*_{4-N_f} : & \text{4d SU(2) SQCD with } N_f \text{ fundamentals} \ , \\
F_\infty &= I_{9-n} : & D_{S^1} E_n \ .
\end{aligned} \qquad (2.7)$$

The remaining possibilities describe 4d $\mathcal{N} = 2$ SCFTs, such as the AD and MN theories [3, 4, 7, 8, 14, 18, 22].

**Higher-form symmetries.** Finally, let us comment on the higher-form symmetries of the theories described above (see also [91–94] and references therein). The candidate theories that have 1-form symmetries are those which allow a gauge theory description where the matter fields do not transform under the centre of the gauge group [95]. The first such example is the pure 4d SU(2) theory, which has a $\Gamma^{(1)} = \mathbb{Z}_2$

1-form symmetry. For future reference, let us mention that this theory also has a mixed anomaly between the 1-form symmetry and the $\mathbb{Z}_8 \subset \mathrm{U}(1)_{\mathbf{r}}$ 0-form symmetry, originating from the classical $R$-symmetry and broken by an ABJ anomaly [96]. Another example of a purely 4d theory having a 1-form symmetry is the $\mathcal{N} = 2^*$ theory, which is the $\mathcal{N} = 2$ preserving mass deformation of $\mathcal{N} = 4$ SYM.

It is by now well-known that the 5d $E_1$ SCFT also exhibits a $\Gamma^{(1)} = \mathbb{Z}_2$ 1-form symmetry. This might not come as a surprise since the theory admits a deformation to the 5d $\mathcal{N} = 1$ supersymmetric gauge theory with gauge algebra $\mathfrak{su}(2)$. Consequently, the circle compactification to the $D_{S^1} E_1$ theory yields both a $\mathbb{Z}_2$ 0-form and a 1-form symmetry.

Finally, the 5d $E_0$ theory has a $\Gamma^{(1)} = \mathbb{Z}_3$ 1-form symmetry [97, 98], leading to both a $\mathbb{Z}_3^{(1)}$ and $\mathbb{Z}_3^{(0)}$ 1-form and 0-form symmetries in the 4d KK theory. In this case, however, there is a cubic anomaly in 5d, leading to a mixed anomaly in the 4d theory [24]. We will discuss the relation between the 0-form and 1-form symmetries in more detail in Section 5.

## 2.2 Automorphisms of elliptic surfaces

The Seiberg–Witten geometry of rank-one 4d $\mathcal{N} = 2$ theories is a rational elliptic surface (RES), whose base corresponds to the $U$-plane, together with the point at infinity. As previously explained, by a RES we mean an elliptic fibration over the complex projective plane with a section (referred to as the *zero-section*) and at least one singular fibre [60]. The main focus of this work will be on the group of automorphisms of these surfaces, and on quotients of the SW geometry by such automorphisms [99–108].

Perhaps a natural interpretation of these quotients is as discrete gaugings of the underlying field theories, but this dictionary can be further expanded. The group of automorphisms of a RES $\mathcal{S}$ is isomorphic to the semi-direct product [25]:

$$\boxed{\mathrm{Aut}(\mathcal{S}) = \mathrm{MW}(\mathcal{S}) \rtimes \mathrm{Aut}_\sigma(\mathcal{S}) \ .}$$

(2.8)

Here $\mathrm{MW}(\mathcal{S})$ is the Mordell–Weil group of $\mathcal{S}$, while $\mathrm{Aut}_\sigma(\mathcal{S})$ is the subgroup of automorphisms preserving the zero section $\sigma$. We will analyse the two subgroups separately in the rest of this section, and collect the precise definitions and relations in Appendix A.2.

An important distinction between the two groups on the RHS of (2.8) is the following: the group $\mathrm{Aut}_\sigma(\mathcal{S})$ is not fixed by the singular fibres of $\mathcal{S}$, as it is the case for $\mathrm{MW}(\mathcal{S})$.[3] The automorphisms $\mathrm{Aut}_\sigma(\mathcal{S})$ induce automorphisms on the base $\mathbb{P}^1 \cong \{U\}$ of the SW geometry. Thus, these automorphisms appear whenever the $U$-plane exhibits certain symmetries, which occur by tuning the mass parameters of

---

[3]This is not entirely accurate, as there are some exceptions of configurations having the same singular fibres but different MW groups, as we discuss later.

a theory. As such, perturbations of the mass parameters can explicitly break the CB symmetry, while preserving the configuration of singular fibres, leading thus to a different $\mathrm{Aut}_\sigma(\mathcal{S})$ group. However, if the singular fibres are not changed, the MW group is not modified.

### 2.2.1 Isogenies

Consider a section $P \in \mathrm{MW}(\mathcal{S})$ of the elliptic fibration $\pi : \mathcal{S} \to \mathbb{P}^1$. For any such section we can define an automorphism $t_P$ of $\mathcal{S}$ as the translation by $P$ on the elliptic fibres [109]. On the smooth fibres $E$, this automorphism acts as a simple translation by $P$, implemented through the usual addition on elliptic curves. Note that this translation does not preserve the zero-section of the elliptic surface, but can be thought of as an automorphism of the underlying surface by forgetting the elliptic structure.

Our primary focus will be on the torsion sections of the MW group. For any torsion section $P$ of finite order $k \in \mathbb{N}$, the automorphism $t_P$ will also be of finite order and will operate without any fixed points. For the generic smooth fibre $E$ of the elliptic surface $\mathcal{S}$, the quotient by the torsion subgroup $\langle t_P \rangle$ generated by $t_P$ leads to a $k-isogenous$ elliptic curve $\widetilde{E}$. Here, we define a $k$-isogeny $\varphi_k : E \to \widetilde{E}$ as a $k$-to-1 homomorphism, mapping the marked point of $E$ to that of $\widetilde{E}$. An isogeny is accompanied by a unique dual isogeny $\hat{\varphi}_k : \widetilde{E} \to E$ such that their composition corresponds to multiplication by $k$:

$$\varphi_k \circ \hat{\varphi}_k = k \, \widetilde{E} \, , \qquad \hat{\varphi}_k \circ \varphi_k = k \, E \, , \tag{2.9}$$

where by $k \, E$ we mean the multiplication-by-$k$ map on the corresponding elliptic curve.

Such quotients will also affect the singular fibres of the RES. Recall that in the Weierstraß model, the singular fibres will be either nodal or cuspidal curves and upon resolving these singularities the exceptional divisors will intersect according to an extended ADE Dynkin diagram, as indicated in Table 4. To each fibre, we also associate an Euler number $e_v$, which is equal to the order of vanishing of the discriminant $\Delta$. Then, an important constraint for a rational elliptic surface is that:

$$e(\mathcal{S}) = \sum_v e(F_v) = 12 \, . \tag{2.10}$$

Consider now a singular fibre of type $I_n$. Then, under a $k$-isogeny generated by the $k$-torsion section $P$, for prime $k$, the action on the singular fibre will depend on how it is intersected by $P$, as follows [109]:

- For trivial intersections: $\quad I_n \longrightarrow I_{nk} \, ,$
- For non-trivial intersections: $\quad I_n \longrightarrow I_{n/k} \, .$

$$\tag{2.11}$$

Here, an intersection is said to be *non-trivial*[4] whenever the section $P$ intersects the singular point of the fibre in the Weierstraß model. We should also stress that the resulting isogenous elliptic surface must have a torsion section of the same order for the dual isogeny to exist. Importantly, while the Euler numbers of the singular fibres may change, the isogeny does not affect the rationality condition [109]. The explicit form of the isogeny can be determined using Vélu's formula (see e.g. Appendix A.2 of [24] for a review).

**Semi-direct product structure.** Let us also briefly mention the semi-direct product structure of (2.8). Recall that from any section $P \in \mathrm{MW}(\mathcal{S})$ we can define an automorphism $t_P$ of $\mathcal{S}$. Then, given an element $\alpha \in \mathrm{Aut}_\sigma(\mathcal{S})$, there is an action of $\mathrm{Aut}_\sigma(\mathcal{S})$ on $\mathrm{MW}(\mathcal{S})$, defined by:

$$\alpha \cdot t_P = t_{\alpha(P)} \ . \tag{2.12}$$

This subtle difference between a direct product and a semi-direct product will be important in the context of mixed anomalies, which will be discussed in Section 5.

### 2.2.2 Foldings

Let us now consider the automorphisms preserving the zero-section. As mentioned above, any such automorphism $\tau : \mathcal{S} \to \mathcal{S}$ induces an automorphism on the base $\mathbb{P}^1 \cong \{U\}$. We denote by $\mathrm{Aut}_\mathcal{S}(\mathbb{P}^1)$ the collection of all induced automorphisms on the CB. The group in $\mathrm{Aut}_\sigma(\mathcal{S})$ (2.8) is then a $\mathbb{Z}_2$ extension of $\mathrm{Aut}_\mathcal{S}(\mathbb{P}^1)$, where the $\mathbb{Z}_2$ acts as the inversion on the elliptic fibre, *i.e.* it maps $(x, y) \mapsto (x, -y)$ in the Weierstraß form (2.3) [25]:

$$1 \to \mathbb{Z}_2 \to \mathrm{Aut}_\sigma(\mathcal{S}) \to \mathrm{Aut}_\mathcal{S}(\mathbb{P}^1) \to 1 \ . \tag{2.13}$$

**Induced automorphisms.** For any rational elliptic surface $\mathcal{S}$, the group $\mathrm{Aut}_\mathcal{S}(\mathbb{P}^1)$ is isomorphic to only one of the following [25]:

- The cyclic group $\mathbb{Z}_k$ with $k \leq 12$ and $k \neq 11$.

- The Klein four-group $\mathbb{Z}_2 \times \mathbb{Z}_2$.

- The dihedral group $D_k$ (with $2k$ elements) for $k = 3, 4$ or $6$.

- The alternating group $A_4$.

---

[4]It can be shown that whenever a $k$-torsion section, for prime $k$, intersects non-trivially an $I_n$ fibre, then $k|n$ – see Corollary 7.5 of [109].

For the rational elliptic surfaces having constant $\mathcal{J}$-map, namely the configurations having only elliptic elements and/or $I_0^*$ singular fibres, there are a few more possibilities [26], which, however, will not be relevant for our purposes.

Note that the group of induced automorphisms is always a subgroup of the symmetric group $\text{Aut}_\mathcal{S}(\mathbb{P}^1) \subset S_k$, with $k$ being the number of singular fibres of $\mathcal{S}$. This is, of course, just a statement about the maximal symmetry exchanging the $U$-plane singularities.

As already alluded to, the configuration of singular fibres does not automatically determine the group of automorphisms $\text{Aut}_\sigma(\mathcal{S})$. As an example, consider the $D_{S^1}E_1$ theory, for which there exists a point in the moduli space where the $U$-plane singularities arrange themselves as the roots of $U^4 = c$, for some $c \in \mathbb{C}$ [22]. In this case, the group of induced automorphisms is enhanced to $\text{Aut}_\mathcal{S}(\mathbb{P}^1) = \mathbb{Z}_4$, while, otherwise, this is only $\text{Aut}_\mathcal{S}(\mathbb{P}^1) = \mathbb{Z}_2$, for the same configuration of singular fibres: $(I_8; 4I_1)$. We will return to this example in the following sections.

**Base change.** Let us now consider quotients of rational elliptic surfaces by subgroups of $\text{Aut}_\sigma(\mathcal{S})$. Generally, such quotients are well-defined only if the generating automorphism $\alpha \in \text{Aut}_\sigma(\mathcal{S})$ has the same order $k$ as the one induced on the base curve. The induced map on the base is then a rotation $U \mapsto \omega_k U$ by the angle $\frac{2\pi}{k}$, that is, $(\omega_k)^k = 1$. We should point out that there exist CB symmetries which are not induced by automorphisms of the underlying RES. Quotients by such automorphisms will not generate new RES, and thus should also not be allowed physically.

Given the quotient projection $\epsilon : \mathcal{S} \to \tilde{\mathcal{S}} = \mathcal{S}/\langle\alpha\rangle$, for $\alpha \in \text{Aut}_\sigma(\mathcal{S})$, there is an induced projection map on the CB [25]:

$$g_k : \mathbb{P}^1 \longrightarrow \mathbb{P}^1 \ , \qquad U \mapsto z = U^k \ . \tag{2.14}$$

The complete list of such quotients can be found in Table 6 of [25] for non-constant $\mathcal{J}$, and Table 6 of [26] for constant $\mathcal{J}$-maps. In the mathematical literature, these constructions are usually referred to as *base changes*, and are important for determining the MW groups of elliptic surfaces [60]. Similar quotient projections for non-cyclic symmetries have not been considered to the same extent in the mathematical literature, and we discuss them further in Section 3.5 and Appendix C.

Base changes are reflected as *k-foldings* of the $U$-plane, according to (2.14). Let us also mention that in this case, the degrees of the $\mathcal{J}$-maps of the two rational elliptic surfaces are related by

$$\deg(\mathcal{J}_\mathcal{S}) = k \cdot \deg(\mathcal{J}_{\tilde{\mathcal{S}}}) \ , \tag{2.15}$$

for $k > 1$.[5] As a result, these quotients are quite different from isogenies, as the latter preserve the degree of the $\mathcal{J}$-map. Foldings can be also understood from the

---

[5]We define the degree of the rational function $\mathcal{J}(U)$ as the maximum of the degrees of the numerator and denominator polynomials of $U$. See Appendix A.

perspective of fundamental domains, where the identity (2.15) is particularly relevant – we leave this discussion to Appendix B.

**Quadratic twists.** Finally, there is an interesting interplay between base changes and quadratic twists which allows for a richer structure when applied to Seiberg–Witten geometry. A quadratic twist of an elliptic curve is another elliptic curve obtained by rescaling the relative invariants $g_2$ and $g_3$ in the Weierstraß model (see Appendix A.1). This transfers 'stars' across singular fibres as follows:

$$I_n \longleftrightarrow I_n^* \, , \qquad II \longleftrightarrow IV^* \, , \qquad III \longleftrightarrow III^* \, , \qquad IV \longleftrightarrow II^* \, . \qquad (2.16)$$

It is known that two elliptic curves with the same $\mathcal{J}$-invariant are either isomorphic or quadratic twists of each other. Likewise, two elliptic surfaces with the same $\mathcal{J}$-map have the same singular fibres, up to a quadratic twist (or *transfer of star*). In particular, the smooth fibres are isomorphic.

As we explain in more detail below, CB foldings of SQCD or KK theories are associated with changes in the type of fibre at infinity $F_\infty$. The only way to preserve the type of fibre at infinity is to combine the folding with a simultaneous quadratic twist. This therefore gives two possibilities for the behaviour of the type of fibre at infinity $F_\infty$:

1. Folding: $\qquad\qquad\qquad\qquad\qquad I_n \mapsto I_{n/k}^* \, , \text{ or } I_n^* \mapsto I_{n/k} \, ,$

2. Folding with quadratic twist: $\quad I_n \mapsto I_{n/k} \, , \text{ and } I_n^* \mapsto I_{n/k}^* \, .$

$$(2.17)$$

We will study these foldings in detail in Section 4.

Combining this base-change with a quadratic twist rescales the Weierstraß invariants,

$$g_2(U) \to f^2(z) \, g_2\left(z^{\frac{1}{k}}\right) \, , \qquad g_2(U) \to f^3(z) \, g_3\left(z^{\frac{1}{k}}\right) \, . \qquad (2.18)$$

Then in order for the fibre at infinity $F_\infty$ to be preserved under a base change, we need to apply a quadratic twist[6] with $f(z) = z^{2-\frac{2}{k}} = U^{2k-2}$. Indeed, in this case $g_2(U)$ becomes $g_2(z) \sim z^4$ and $g_3(U)$ becomes $g_3(z) \sim z^6$ for all $k$. This generalises the discussion of the $\mathbb{Z}_3$ and $\mathbb{Z}_2$-foldings with quadratic twists of the $D_{S^1}E_0$ and $D_{S^1}E_1$ theories discussed in [24, (4.23) & (5.21)].

To summarise, any $k$-folding has the effect of mapping a rational elliptic surface $\mathcal{S}$ to another surface $\widetilde{\mathcal{S}}$ such that their $\mathcal{J}$-maps are related as (2.15). Without a quadratic twist, this changes the fibre at infinity. Including the specific quadratic twist $f(z) = z^{2-\frac{2}{k}}$ however, we can preserve the type of fibre at infinity.

---

[6]Up to proportionality constants.

# 3  Folding the $U$-plane

As outlined in Section 2, the Coulomb branch of a theory $\mathcal{T}$ can have certain discrete $\mathbb{Z}_N$ symmetries. This prompts a natural question about the interpretation of the quotients of the SW geometry by such symmetries. Note that the $U$-plane symmetries are generally subgroups of $H \subset S_n$, where $n$ is the number of singularities, but quotients $\mathcal{S}/H$ of the associated rational elliptic surface $\mathcal{S}$ by such subgroups do not typically lead to other rational elliptic surfaces [25]. We will thus consider mainly $\mathbb{Z}_N$ symmetries, and comment on non-cyclic symmetries in Appendix C.

In Appendix B, we discuss how $U$-plane foldings can be obtained from both modular and non-modular configurations. For the modular cases, we relate the discrete $\mathbb{Z}_N$ symmetries to a pair of subgroups of $\mathrm{SL}(2,\mathbb{Z})$. Generic configurations are however non-modular, in which case the $\mathbb{Z}_N$ symmetries can be read off from a consistent choice of the fundamental domain. Our discussion in the appendix can be applied to any rational elliptic surface, and can likely be generalised to other (non-rational) elliptic surfaces, such as K3 surfaces, giving rise to fundamental domains of index larger than 12 [65, 77, 78, 110].

The simplest interpretation of $U$-plane foldings is in terms of discrete gaugings. Such a Coulomb branch analysis for the five-dimensional theories has not been discussed in the literature, to the best of our knowledge. Nevertheless, the gaugeable CB symmetries turn out to be related to the symmetries of the corresponding $(p,q)$-brane webs, as we will show below.

## 3.1  Discrete gaugings of 4d theories

The continuous internal symmetries of 4d $\mathcal{N} = 2$ superconformal field theories are subgroups of the direct product $\mathrm{U}(1)_{\mathbf{r}} \times \mathrm{SU}(2)_R \times F$ of the R-symmetry and the flavour symmetry algebra, $F$. It was argued in [19, 33] that discrete subgroups of $F$ cannot be gauged without adding new degrees of freedom, while gaugings of subgroups of $\mathrm{SU}(2)_R$ break (some of the) supersymmetry. As a result, the most general discrete gaugings that one can thus consider are those of symmetries generated by transformations:

$$C = (\rho, \sigma, \varphi) \in \mathrm{U}(1)_{\mathbf{r}} \times \mathrm{SL}(2,\mathbb{Z}) \times \mathrm{Out}(F) , \qquad (3.1)$$

with $\mathrm{Out}(F)$ the outer automorphism group of the flavour algebra.[7] The three generators must be chosen in such a way that the $\mathcal{N} = 2$ supersymmetry is preserved. Let us note that the $\sigma$ and $\varphi$ factors do not act on the CB parameter, but instead on the gauge coupling $\tau$ and the mass deformations.

---

[7]For instance, the 4d SU(2) theory with $N_f = 4$ fundamental hypermultiplets has $F = \mathrm{Spin}(8)$ flavour symmetry, with outer automorphism group $\mathrm{Out}(F) = S_3$ the triality group. See [43, 111] for detailed discussions of this symmetry.

**Discrete gauging.** Consider first the generator $\sigma$ of some subgroup of the modular group, $\mathbb{Z}_k \subset \mathrm{SL}(2, \mathbb{Z})$. It is straightforward to see that the only such subgroups are $\mathbb{Z}_2, \mathbb{Z}_3, \mathbb{Z}_4$ and $\mathbb{Z}_6$. Such symmetries can only exist for fixed values of $\tau$, the only exception being that of $\mathbb{Z}_2$, where the action of $\sigma$ on $\tau$ is trivial.

This transformation acts on the supercharges as a simple phase factor; as such, in order to preserve the $\mathcal{N} = 2$ supersymmetry, we need to accompany it by a $\mathbb{Z}_k \subset \mathrm{U}(1)_{\mathbf{r}}$ action generated by some element $\rho \in \mathrm{U}(1)_{\mathbf{r}}$. This action manifests directly on the CB parameter $u$. Namely, the gauged theory will be described by $\widetilde{u} = u^r$, for $r \in \mathbb{N}$ the smallest positive integer necessary to build a $\mathrm{U}(1)_{\mathbf{r}}$ invariant operator [33]. That is:

$$r = \frac{k}{\Delta_u} \, , \tag{3.2}$$

with $\Delta_u$ being the scaling dimension of the CB parameter. This leads to a new SCFT whose CB parameter has scaling dimension:

$$\Delta_{\widetilde{u}} = r\Delta_u = k \, . \tag{3.3}$$

Finally, the discrete gauging must involve some $\mathrm{Out}(F)$ action relating the mass parameters of the two theories. Note that here $\mathbb{Z}_r$ is not the symmetry being gauged, but only a subgroup of it, $\mathbb{Z}_r \subset \mathbb{Z}_k = \mathbb{Z}_{r\Delta_u}$. Nevertheless, we will sometimes refer to such CB foldings as discrete gaugings.

In the language of rational elliptic surfaces, the CB scaling dimension is set by the fibre at infinity as follows:

| $\boldsymbol{F_\infty}$ | $II$ | $III$ | $IV$ | $I_0^*$ | $II^*$ | $III^*$ | $IV^*$ |
|---|---|---|---|---|---|---|---|
| $\boldsymbol{\Delta_u}$ | $6$ | $4$ | $3$ | $2$ | $\frac{6}{5}$ | $\frac{4}{3}$ | $\frac{3}{2}$ |

$$\tag{3.4}$$

Then, the discrete gauging relating the rational elliptic surfaces $\mathcal{S}$ and $\widetilde{\mathcal{S}} = \mathcal{S}/\mathbb{Z}_r$, for $\mathbb{Z}_r \subset \mathrm{Aut}_\sigma(\mathcal{S})$, will be one of the following possibilities [18]:

| $\boldsymbol{r}$ | $5$ | $4$ | $3$ | $2$ | $2$ | $2$ |
|---|---|---|---|---|---|---|
| $\boldsymbol{F_\infty(\widetilde{\mathcal{S}})}$ | $II^*$ | $IV^*$ | $I_0^*$ | $IV$ | $I_0^*$ | $IV^*$ |
| $\boldsymbol{F_\infty(\mathcal{S})}$ | $II$ | $II$ | $II$ | $II$ | $III$ | $IV$ |

$$\tag{3.5}$$

These follow from the physical constraint on the scaling dimensions (3.3). Out of these, the $r = 5$ case is not allowed, as there are no such quotients from the perspective of rational elliptic surfaces [25]. The lists of [25] can be used to find all 4d $\mathcal{N} = 2$ SCFTs in the classification of [9, 19–21], which was already detailed in [18].

Given a Coulomb branch with a $\mathbb{Z}_r$ symmetry, we distinguish two distinct folding scenarios depending on the fixed point of the $\mathbb{Z}_r$ symmetry. That is, we can either have a smooth fibre $I_0$, or a singular fibre as the fixed point of the symmetry. We shall discuss the two cases below and give a list of how these fixed points change

under discrete gaugings from the perspective of 4d $\mathcal{N}=2$ theories. In the context of 5d SCFTs, these results can be interpreted as insertions of certain 7-branes in the geometry.

**Foldings with smooth fibres at the fixed point.** Consider first the case where the fixed point of the $\mathbb{Z}_r$ symmetry is not a CB singularity. This will correspond to gaugings of free U(1) theories. In particular, $\mathbb{Z}_r$ gaugings of a free vector multiplet lead to so-called *frozen* singularities:

$$I_0 \xrightarrow{\mathbb{Z}_r} \begin{cases} I_0^*, & \text{for } r=2 , \\ IV^*, & \text{for } r=3 , \\ III^*, & \text{for } r=4 , \\ II^*, & \text{for } r=6 . \end{cases} \tag{3.6}$$

For a simple illustration, consider the $r=4$ case, for which there is a single cover, namely:

$$(IV^*; 4I_1) \longrightarrow (II; III^*, I_1) . \tag{3.7}$$

This corresponds to the discrete gauging of the $A_2$ Argyres-Douglas theory – sometimes also denoted $H_2$ or $(A_1, D_4)$ – by a $\mathbb{Z}_6 \subset \mathrm{U}(1)_{\mathbf{r}}$ symmetry, acting on the CB as a $\mathbb{Z}_4$ symmetry [33]. In Section 3.3, we will give an alternative interpretation to these gaugings for 5d theories in terms of insertions of 7-branes.

**Foldings with singular fibres at the fixed point.** The second scenario is when the fixed point of the CB symmetry (which is usually the origin of the $U$-plane) has a singularity. In the context of 4d theories, we ought to mention the case of $\mathcal{N}=2$ U(1) gauge theories with massless matter, whose CB is described by a $I_n$ cusp. The $\mathbb{Z}_2$ gauging of such theories can only be implemented for even $n=2n'$ (being anomalous for odd $n$), in which case the CB transforms as [33]:

$$I_{2n'} \xrightarrow{\mathbb{Z}_2} I_{n'}^* . \tag{3.8}$$

The daughter theories are SU(2) gauge theories with $\Delta(\widetilde{u}) = 2$. Importantly, note that IR free theories cannot have other $\mathbb{Z}_k$ global symmetries in $\mathrm{SL}(2,\mathbb{Z})$ with $k > 2$, as $\tau$ is allowed to vary on the Coulomb branch.

One can also have $II, III,$ or $IV$ singularities at the fixed point of the discrete symmetry. In four dimensions, these singularities correspond to the $H_0, H_1$ and $H_2$

Argyres-Douglas theories, which admit the following discrete gaugings [33]:

| $\mathcal{T}$ | Flavour | $\mathbb{Z}_r$ | $\mathcal{T}/\mathbb{Z}_r$ | Deformation | Flavour |
|---|---|---|---|---|---|
| $II$ | $-$ | $\mathbb{Z}_5$ | $II^*$ | $(II^*)$ | $-$ |
| $III$ | $A_1$ | $\mathbb{Z}_3$ | $III^*$ | $(IV^*, I_1)$ | $A_1$ |
| $IV$ | $A_2$ | $\mathbb{Z}_2$ | $IV^*$ | $(I_0^*, 2I_1)$ | $A_2$ |
| $IV$ | $A_2$ | $\mathbb{Z}_4$ | $II^*$ | $(III^*, I_1)$ | $A_1$ |

$$(3.9)$$

Consider first the $H_1$ theory, with the massless singularity $III$. Whenever the chiral deformation of the SW curve is frozen, the CB singularities organise in a $\mathbb{Z}_3$ symmetric way. As a result, the gauged theory simply freezes this deformation, without acting on the $A_1$ flavour symmetry. For the discrete gaugings of the $H_2$ AD theory, the flavour symmetry changes only in the $\mathbb{Z}_4$ case [33].

A rather interesting example is the $\mathbb{Z}_5$ gauging of the $H_0$ theory proposed in [33]. This symmetry is part of the $U(1)_{\mathbf{r}}$ R-symmetry and freezes the chiral deformation:

$$II \xrightarrow{\ \mathbb{Z}_5\ } II^* \ . \tag{3.10}$$

This new theory, however, has been deemed as rather speculative, as it is not connected by RG flows to any other 4d $\mathcal{N} = 2$ SCFTs. We will come back to this example when discussing gaugings of 5d SCFTs. Let us also mention that the singular fibres $I_0^*$ or $IV^*$ can also appear as the fixed points of a $\mathbb{Z}_r$ symmetry. Then, the (bulk) $U$-plane singularities behave as follows [33]:

$$I_0^* \xrightarrow{\ \mathbb{Z}_r\ } \begin{cases} III^*, & \text{for } r = 2 \ , \\ II^*, & \text{for } r = 3 \ , \end{cases} \qquad IV^* \xrightarrow{\ \mathbb{Z}_2\ } II^* \ . \tag{3.11}$$

As a final check, we consider the fibres of type $I_n^*$, for $n > 0$. From the classification of rational elliptic surfaces [112, 113], these are limited to $n \leq 4$ and, in four dimensions, they will either describe SU(2) gauge theories with $N_f = 4 - n$ fundamentals, or, alternatively $\mathbb{Z}_2 \times U(1) = O(2)$ gauge theories with matter (*i.e.* the previously mentioned U(1) gauge theories from which a $\mathbb{Z}_2$ symmetry was gauged; this is a mix of the $\mathbb{Z}_2$ charge conjugation and $\mathbb{Z}_2 \subset U(1)_{\mathbf{r}}$). In the former case, the classical $U(1)_{\mathbf{r}}$ R-symmetry is anomalous being completely broken for $N_f > 0$ with generic masses of the flavour multiplets. However, in the massless case, a discrete $\mathbb{Z}_{4(4-N_f)}$ symmetry is anomaly-free. Since a $\mathbb{Z}_2$ subgroup of this acts the same as the centre of the $SO(2N_f)$ flavour symmetry, the $u$-plane will only have a $\mathbb{Z}_{4-N_f}$ symmetry (or $\mathbb{Z}_2$ for $N_f = 0$). Let us note, however, that in these theories there are mixed 't Hooft anomalies between the residual discrete R-symmetry and the $SU(2)_R$ symmetry, or with gravity, as well as a cubic anomaly (see *e.g.* [96]). Hence these residual R-symmetries cannot be gauged.

## 3.2 Discrete gaugings of 5d theories

The next step in our analysis is to generalise the above picture for KK theories. Five-dimensional $\mathcal{N} = 1$ SCFTs only possess an $SU(2)_R$ R-symmetry, but the SW geometries we are considering correspond to effective 4d $\mathcal{N} = 2$ theories, which can include discrete subgroups of the additional $U(1)_{\mathbf{r}}$ symmetry. Such 0-form symmetries on the CB arise as accidental R-symmetries of the KK theory. Note also that, as opposed to the purely four-dimensional theories, the CB parameter $U$ is now dimensionless, and thus there is no analogue for (3.3).

### 3.2.1 Discrete symmetries and the prepotential

Before we discuss explicit examples of foldings, we first constrain the cyclic symmetries which lead to well-defined quotients of the SW geometry. These quotients lead to a simple effect on the periods and the prepotential of the respective theory.

**Gaugeable $\mathbb{Z}_k$ symmetries.** For the $D_{S^1} E_n$ theories, the fibre at infinity is of $I_{9-n}$ type, and, thus, a $\mathbb{Z}_k$ discrete gauging must transform the fibre at infinity as:

$$\mathbb{Z}_k \text{ gaugings for } D_{S^1} E_n : \qquad I_{9-n}^\infty \xrightarrow{\;\mathbb{Z}_k\;} I_{(9-n)/k}^\infty \; . \tag{3.12}$$

Thus, it must be the case that $k|(9-n)$. Combining this with the fact that the $U$-plane has $n+3$ simple singularities (*i.e.* $I_1$) for generic mass parameters, the allowed $\mathbb{Z}_k$ symmetries are shown in Table 2. Note that these conditions do not explicitly exclude the symmetries of $D_{S^1} \widetilde{E}_1$, for which a more careful argument is needed. We will come back to this matter in Section 3.2.3.

The symmetries listed in Table 2 are in precise agreement with the possible induced cyclic automorphisms $\text{Aut}_\mathcal{S}(\mathbb{P}^1) \cong \mathbb{Z}_k$ on the $\mathbb{P}^1$ base of the SW geometry [25]. For $n \geq 3$, the gaugeable $\mathbb{Z}_k$ symmetries of the $D_{S^1} E_n$ theories are subgroups of the centre of the simply connected flavour symmetry groups,[8]

$$Z(E_n) \cong \mathbb{Z}_{9-n} \; . \tag{3.13}$$

This relation (3.13) of course only holds for the theories whose maximal flavour algebra does not involve a $U(1)$ factor. Note also that the folding by the centre of the simply connected flavour symmetry group gives a map from the $D_{S^1} E_n$ theory to a $D_{S^1} E_8$ theory.

**Periods and prepotential.** Such discrete quotients of the SW geometry do lead to tractable changes in the prepotential and the periods. As shown in *e.g.* [22], the Picard-Fuchs differential equations for rank-one theories reduce to a single second

---

[8]Note also that the $D_{S^1} E_n$ theories are geometrically engineered on local $dP_n$ surfaces, where the del Pezzo surface $dP_n$ has degree (*i.e.* self-intersection number of the canonical class) $9-n$.

order differential equation, with the solutions being referred to as the 'geometric periods', i.e. the periods defined as:

$$\omega_D = \frac{da_D}{dU} \ , \qquad \omega_a = \frac{da}{dU} \ , \tag{3.14}$$

with $(a_D, a)$ as introduced in (2.1). For the local $dP_n$ geometries, in a conveniently chosen basis, these are the only two periods that receive quantum corrections and coincide with the classical D-brane periods in the large volume limit (see e.g. [114]). Namely, we have $\Pi_{D4} = a_D$, and $\Pi_{D2} = 2a$, where $\Pi_{D4}$ and $\Pi_{D2}$ are the periods associated with D4 and D2-branes wrapping the 4-cycle and a 2-cycle of the local threefold, respectively. In this limit the so-called 'mirror map' [89, 90] becomes:

$$a \approx \frac{1}{2\pi i} \log U \ , \qquad \omega_a = \frac{da}{dU} \approx \frac{1}{2\pi i} \frac{1}{U} \ , \tag{3.15}$$

in the region $U \to \infty$. Moreover, to leading order, we have [22]:

$$\Pi_{D4} \approx \frac{9-n}{2} a^2 + \mathcal{O}(a) \ , \tag{3.16}$$

where we neglect the contribution due to the mass parameters. This leads to the following asymptotics for the dual geometric period:

$$\omega_D \approx -\frac{9-n}{4\pi^2} \log U + \dots \ . \tag{3.17}$$

As the Coulomb branch of $\mathcal{T} = D_{S^1} E_n$ contains $n+3$ singularities for generic values of the mass parameters, the analytic continuation of $(\omega_D, \omega_a)$ throughout the entire $U$-plane is highly non-trivial. In the cases of interest, the $U$-plane shows a $\mathbb{Z}_k$ symmetry, in which case it becomes simpler to solve for the periods on the $z = U^k$ plane. Evidently, these would be the periods of the folded theory, and we will refer to them as $(\widetilde{\omega}_D, \widetilde{\omega}_a)$ and $(\widetilde{a}_D, \widetilde{a})$ [22].

Consider a theory $\mathcal{T} = D_{S^1} E_n$ whose $U$-plane possesses a $\mathbb{Z}_k$-symmetry, and let $\mathcal{T}/\mathbb{Z}_k$ denote the new theory arising from the $\mathbb{Z}_k$-folding of the $U$-plane of $\mathcal{T}$. We will also use $z = \widetilde{U} = U^k$. From the above considerations, by matching asymptotics it is not difficult to see that there is a region of the $U$-plane of $\mathcal{T}$ where the periods $(\omega_D, \omega_a)$ are related to $(\widetilde{\omega}_D, \widetilde{\omega}_a)$ by:

$$\omega_D \approx k\, \widetilde{\omega}_D \ , \qquad \omega_a = \widetilde{\omega}_a \ , \tag{3.18}$$

to leading order, where the additional factor in $\omega_D$ appears due to the logarithmic term $\log(U^k)$ inside the geometric period $\widetilde{\omega}_D$. Thus, the leading order terms in the prepotential will be related as follows:[9]

$$\mathcal{F}_{\mathcal{T}} \approx k\mathcal{F}_{\mathcal{T}/\mathbb{Z}_k} \ . \tag{3.19}$$

---

[9]The astute reader might have noticed that since $\tau = \frac{\omega_D}{\omega}$, the identity (3.18) appears to lead to a $k$-isogeny. This is, however, not the case, since our analysis only concerns leading order terms.

This expression is identical to the one derived in [38], in the context of *five-dimensional S-folds*. There, the $\mathbb{Z}_k$-folding procedure involved symmetries of the $(p, q)$-brane webs. We postpone the discussion about the relation between the two constructions to Section 3.3.

### 3.2.2 Symmetry enhancements

In the remainder of this section, we will show how the folding procedure is explicitly realised through some examples.

**The $\mathbb{Z}_3$-folding of $E_0$.** The $U$-plane of the $D_{S^1}E_0$ theory is $\mathbb{Z}_3$ symmetric and, thus, we can apply the folding procedure discussed above. The $\mathbb{Z}_3$ symmetry identifies the three bulk $I_1$ singularities, and thus the folded Coulomb branch $w = U^3$, corresponds to the transition:

$$(I_9^\infty; 3I_1) \xrightarrow{\mathbb{Z}_3} (I_3^\infty; I_1, IV^*) \ . \tag{3.20}$$

This discrete gauging was also discussed in [24]. While the three $I_1$ cusps are identified, the origin of the $w$-plane becomes a singular point of type $IV^*$, as expected from (3.6). We also note that the new configuration of singular fibres is identical to that of the massless $D_{S^1}E_6$ configuration, which is modular with monodromy group $\Gamma^0(3)$. The orbifold point $w = 0$ corresponds to an elliptic point of $\Gamma^0(3)$, and is smoothed out on the $\mathbb{Z}_3$ cover. This folding can be also argued from the perspective of modular elliptic surfaces, as detailed in Appendix B. We say, in particular, that $\Gamma^0(9) \sqsubset \Gamma^0(3)$, *i.e.* $\Gamma^0(9)$ is a subgroup of $\Gamma^0(3)$ which can be folded to the latter.

**The $\mathbb{Z}_2$-folding of $E_1$.** The $U$-plane of the $D_{S^1}E_1$ theory has a $\mathbb{Z}_2$ symmetry for generic values of the exponentiated inverse gauge coupling parameter $\lambda$. One can interpret this from a gauge theory point of view as follows: the CB parameter $U$ is the VEV of a fundamental Wilson line, which, under multiplication by elements of the centre of the gauge group does not change the physics as long as all fields are in the adjoint of the gauge group [86]. Equivalently, the $E_1$ five-dimensional SCFT is known to possess a $\mathbb{Z}_2$ 1-form symmetry, which, upon reduction on $S^1$ of the theory, becomes both a one and a 0-form symmetry [97, 98].

Let us focus for now on the massless configuration, which is modular, with monodromy group $\Gamma^0(8)$. The folding procedure corresponds to the transition:

$$(I_8^\infty; I_2, 2I_1) \xrightarrow{\mathbb{Z}_2} (I_4^\infty; I_1^*, I_1) \ , \tag{3.21}$$

where the new configuration can be interpreted as the massless $D_{S^1}E_5$ configuration. Note also that this is again a modular configuration, with monodromy group $\Gamma^0(4)$. Let us point out that this $\mathbb{Z}_2$-folding is different from a 2-isogeny, which would instead be a first step in gauging the 1-form symmetry [24]. The difference can be

understood, for instance, at the level of the monodromy groups. Whilst isogenies preserve the index of the monodromy group inside $\mathrm{PSL}(2, \mathbb{Z})$, for $\mathbb{Z}_k$-foldings the index of the monodromy group in the $\mathrm{PSL}(2, \mathbb{Z})$ group enlarges by a factor of $k$ – recall (2.15), and see also Appendix B.

**Higher-form symmetries.** The previous two examples offer valuable insight into the origin of the discrete symmetries on the Coulomb branch. The 5d $E_0$ and $E_1$ SCFTs have $\mathbb{Z}_3$ and $\mathbb{Z}_2$ 1-form symmetries [97, 98], respectively, which upon compactification to four dimensions turn into both 1-form and 0-form symmetries, depending on how the line operators wrap the circle direction. For these examples, isogenies can be thought of as discrete gaugings of 1-form symmetries, since the latter are encoded in the MW group of the Seiberg–Witten geometry, as elaborated in [22, 24].

These isogenies should not be mistaken for *Galois covers*. The fundamental distinction is that the SW geometry obtained after an isogeny corresponding to a gauging of a 1-form symmetry will generally contain *undeformable* singularities, in the sense of [19].[10] In contrast, when isogenies are interpreted as Galois covers the resulting singularities will still allow a deformation pattern. We will further analyse these in [32].

Consider for now the $U$-plane of the $D_{S^1} E_1$ theory. The CB singularities lie at the points $\pm 2 \pm 2\sqrt{\lambda}$, for generic values of the exponentiated inverse gauge coupling $\lambda$. Meanwhile, at $\lambda = -1$, the $\mathbb{Z}_2$ 0-form symmetry enhances to $\mathbb{Z}_4$. The existence of the short exact sequence

$$0 \to \mathbb{Z}_2 \to \mathbb{Z}_4 \to \mathbb{Z}_2 \to 0 \ , \tag{3.22}$$

would seem to suggest that the additional $\mathbb{Z}_2$ symmetry needed in this enhancement is a manifestation of the centre of the simply-connected group of the flavour symmetry algebra $\mathbb{Z}_2 \subset \mathrm{SU}(2)$. Note, in particular, that while the flavour symmetry group of the $E_1$ SCFT is the centreless version $\mathrm{SO}(3) \cong \mathrm{SU}(2)/\mathbb{Z}_2$, the Coulomb branch does have the full $\mathrm{SU}(2)$ symmetry – that is, there are massive BPS states in the fundamental representation of $\mathrm{SU}(2)$ [22]. We give some further evidence for this claim below.

**Characters and centre symmetries.** As discussed above, an interesting observation is that the maximal gaugeable $\mathbb{Z}_k$ symmetries for the Coulomb branches of the $D_{S^1} E_n$ theories are precisely the centres (3.13) of the corresponding simply connected flavour symmetry groups. Meanwhile, for $E_2$, the $\mathbb{Z}_2$ centre symmetry of the $\mathrm{SU}(2)$ factor in the flavour symmetry is 'masked' by the $\mathrm{U}(1)$ factor of the flavour symmetry. These maximal $\mathbb{Z}_k$ symmetries shown in Table 2 can be found by fine-tuning

---

[10]Recall that an undeformable or frozen $I_n$ singularity corresponds to a massless hypermultiplet of charge $\sqrt{n}$.

the mass parameters of the theories. Remarkably, an example of such a fine-tuning is achieved for vanishing values of *all* the characters of the $E_n$ groups, as we describe below.[11]

Recall first that the flavour symmetry characters are defined as

$$\chi_{\mathcal{R}}^{E_n} = \sum_{\rho \in \mathcal{R}} e^{2\pi i \rho(\nu)} \ , \tag{3.23}$$

for $\rho = (\rho_i)$ the weights of some fundamental representation $\mathcal{R}$ and $\nu = (\nu_i)$ the $E_n$ flavour parameters. These Lie algebra characters are invariant under Weyl group transformations, being functions on the weight spaces of the Lie algebra to complex numbers (see Chapter 13.5 of [115]). The weight zero characters $\nu = \mathbf{0}$ will thus lead to $\chi_{\mathcal{R}} = \dim(\mathcal{R})$, and correspond to an enhancement of the flavour symmetry algebra on the Coulomb branch.

To better understand the physical interpretation of these characters, recall the gauge theory definition of the CB parameter $U$ as the expectation value of a half-BPS line [22, 24]. As such, the origin of the extended CB corresponds to the vanishing of this Wilson line VEV, which is responsible for the symmetry enhancement at the fixed point of the RG flow. In the same way, the $\chi^{E_n}$ characters can also be interpreted as *flavour Wilson lines*. Accordingly, field theory intuition would suggest that the vanishing of all the $\chi^{E_n}$ characters should also be associated with a symmetry enhancement. However, the precise reason still remains an open question.

This symmetry enhancement can be made more precise by expressing the equations $\chi = 0$ in gauge theory language. In terms of the gauge theory parameters $\lambda$ and $M_i$, the vanishing of the characters can be achieved as follows:

$$\begin{aligned}
E_1 &: & (\mathbb{Z}_4) & & (\lambda) = (-1) \ , \\
E_3 &: & (\mathbb{Z}_6) & & (\lambda, M_1, M_2) = \left( e^{\frac{4\pi i}{6}}, e^{\frac{5\pi i}{6}}, e^{\frac{11\pi i}{6}} \right) \ , \\
E_4 &: & (\mathbb{Z}_5) & & (\lambda, M_1, M_2, M_3) = \left( e^{\frac{2\pi i}{5}}, e^{\frac{0\pi i}{5}}, e^{\frac{2\pi i}{5}}, e^{\frac{4\pi i}{5}} \right) \ , \\
E_5 &: & (\mathbb{Z}_4) & & (\lambda, M_1, M_2, M_3, M_4) = \left( e^{\frac{4\pi i}{4}}, e^{\frac{\pi i}{12}}, e^{\frac{7\pi i}{12}}, e^{\frac{13\pi i}{12}}, e^{\frac{19\pi i}{12}} \right) \ .
\end{aligned} \tag{3.24}$$

In all these examples, the coupling $\lambda$ satisfies $\lambda^{9-n} = 1$, while the masses $M_i$ lie on a unit circle and rotate into each other by a constant phase. Note that we have no a priori reason to believe that $\chi \equiv 0$ necessarily implies the striking result $|\lambda| = |M_i| = 1$. In fact, we will see that this result is not arbitrary, since such configurations correspond to massless theories, where non-trivial holonomies along the circle direction are turned on. These maximal (gaugeable) CB symmetries for the $D_{S^1} E_n$ theories with $n = 4, \ldots, 7$ are sketched in Figure 1, and the CB foldings can be carried out explicitly as before.

---

[11]This is not in disagreement with the $D_{S^1} E_2$ case, as the vanishing of the characters in this case is equivalent to the decoupling of the flavour hypermultiplet generating the abelian U(1) symmetry.

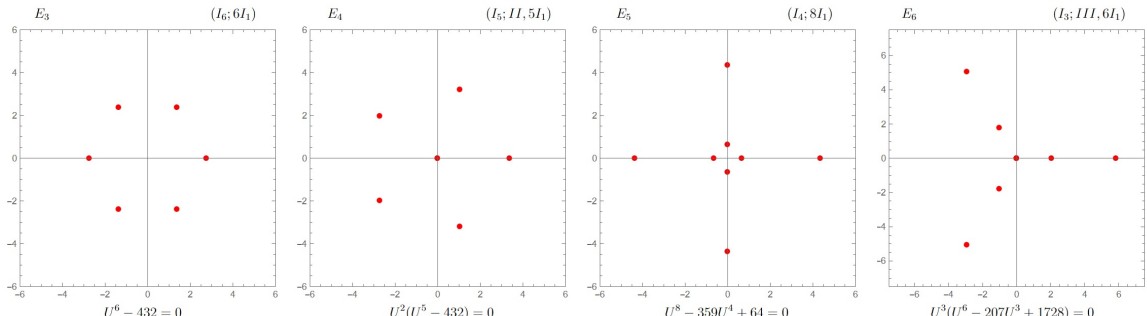

Figure 1: Structure of the $U$-plane for $\chi_i = 0$ for the $D_{S^1}E_n$ theories with $n = 3, \ldots, 6$. The equations below the diagrams give the discriminant locus.

Finally, it is also important to note that not all the $\mathbb{Z}_k$ symmetries of the $U$-plane discussed here are induced by automorphisms of the elliptic surface. In Section C.1, we will discuss an example where these are distinct symmetries.

**6d theories.** Let us briefly comment on the symmetries of $U$-planes originating from toroidal compactifications of six-dimensional theories. For this class of theories, we have $F_\infty = I_0$, and thus a folding will preserve the width of the fibre at infinity.[12] Thus, there are no a priori constraints on the 'foldable' symmetries of 6d theories. From the brane-web perspective we expect that the 6d $E$-string CB will at least allow $\mathbb{Z}_2, \mathbb{Z}_3, \mathbb{Z}_4$ and $\mathbb{Z}_6$ foldable symmetries [38].

Meanwhile, the M-string curve will always have a $\mathbb{Z}_2$ symmetry, inherited from the 1-form symmetry of the theory [24]. We will discuss the corresponding $\mathbb{Z}_2$ folding in Section 4. Let us also note that the so-called relative curve of the M-string theory, containing six $I_2$ fibres in the bulk, will have a $\mathbb{Z}_6$ symmetry. While the $\mathbb{Z}_3$ folding of this configuration would consist of two copies of the $\mathbb{Z}_3$ folded 4d $\mathcal{N} = 2^*$ theory [33], the $\mathbb{Z}_6$ folding does not appear to lead to a RES.

### 3.2.3 Peculiar foldings

Let us now discuss the peculiar cases of the $\widetilde{E}_1$ and $E_4$ theories, as mentioned in Table 2.

**Foldings of $\widetilde{E}_1$.** The $\widetilde{E}_1$ theory is the UV completion of the 5d $\mathcal{N} = 1$ $SU(2)_\pi$ gauge theory, thus sharing some similarities with the $E_1$ theory. While both of their SW geometries have $F_\infty = I_8$, the distinction is in the MW groups: only $D_{S^1}E_1$ has a $\mathbb{Z}_2$ 1-form symmetry, and thus $\mathbb{Z}_2 \subset MW$ for any configuration of $D_{S^1}E_1$. This is no longer the case for $D_{S^1}\widetilde{E}_1$.

The previously presented folding argument (3.12) involved the condition $k|(9-n)$, or $k|8$ for gaugeable $\mathbb{Z}_k$ symmetries of $D_{S^1}\widetilde{E}_1$. From the SW curve, by explicit

---

[12]The width of a fibre of type $I_n$ or $I_n^*$ is $n$.

computation, one can show that a $\mathbb{Z}_4$ symmetry cannot be realised for any value of the gauge theory parameter $\lambda$. However, there exist configurations that show a reflection symmetry along a line passing through the origin of the $U$-plane. Such an example is the configuration $(I_8; II, 2I_1)$, which is modular [14] and has a $\mathbb{Z}_2$ symmetry $U \mapsto -\bar{U}$, where $\bar{U}$ is the complex conjugate of $U$. In this case, the AD point is situated along the symmetry line, with the two $I_1$ singularities being identified under the reflection symmetry. Folding the $U$-plane along this line, one expects a configuration having $F_\infty = I_4$, as well as a bulk $I_1$ fibre and (at least) an elliptic point. The only such rational elliptic surfaces in Persson's classification [112, 113] are $(I_4; I_1, IV, III)$ and $(I_4; I_1, III, 2II)$, which are not promising candidates.

Note, moreover, that quotients of the underlying rational elliptic surface by this automorphism do not generate a new rational elliptic surface (as per Table 6 of [25]). In fact, this reflection symmetry is the first example of a symmetry that is *not induced* from an automorphisms of the elliptic surface preserving the zero section. This is because the transformation $U \mapsto -\bar{U}$ is not a Möbius transformation. Thus, this configuration does not exhibit a foldable $\mathbb{Z}_2$ symmetry. We can also rule out such a $\mathbb{Z}_2$-folding from monodromy considerations; that is, the monodromy group of the $(I_8; II, 2I_1)$ configuration is not an index 2 subgroup of any subgroup of $\mathrm{PSL}(2, \mathbb{Z})$ [14].

**The $\mathbb{Z}_5$-folding of $E_4$.** As pointed out in Table 2, a rather peculiar symmetry is the $\mathbb{Z}_5$ symmetry of the $U$-plane of $D_{S^1} E_4$. This $\mathbb{Z}_5$ symmetry manifests for vanishing characters, leading to the SW curve with Weierstraß invariants

$$g_2(U) = \frac{1}{12} U^4 , \qquad g_3(U) = U \left( 4 - \frac{1}{216} U^5 \right) , \qquad (3.25)$$

while the discriminant and $\mathcal{J}$-invariant read:

$$\Delta(U) = U^2 (U^5 - 432) , \qquad \mathcal{J}(U) = \frac{U^{10}}{12^3 (U^5 - 432)} . \qquad (3.26)$$

Thus, the $\mathbb{Z}_5$-folding of the $U$-plane is equivalent to the transition:

$$(I_5; 5I_1, II) \xrightarrow{\mathbb{Z}_5} (I_1; I_1, II^*) . \qquad (3.27)$$

In particular, at the fixed point of the $\mathbb{Z}_5$ symmetry, there is a type $II$ singular fibre (see Figure 2). Consequently, this folding is essentially describing a $\mathbb{Z}_5$ discrete gauging of the $H_0$ Argyres-Douglas theory. As already mentioned in Section 3.1, such a gauging is however speculative, as there are no other RG flows known that could validate this gauged theory [33].

Curiously, there is another reason why such a discrete gauging might be puzzling. Most gaugings outlined in Table 2 can be realised as five-dimensional S-folds, which involve certain operations on the underlying $(p, q)$-brane web. From a IIB

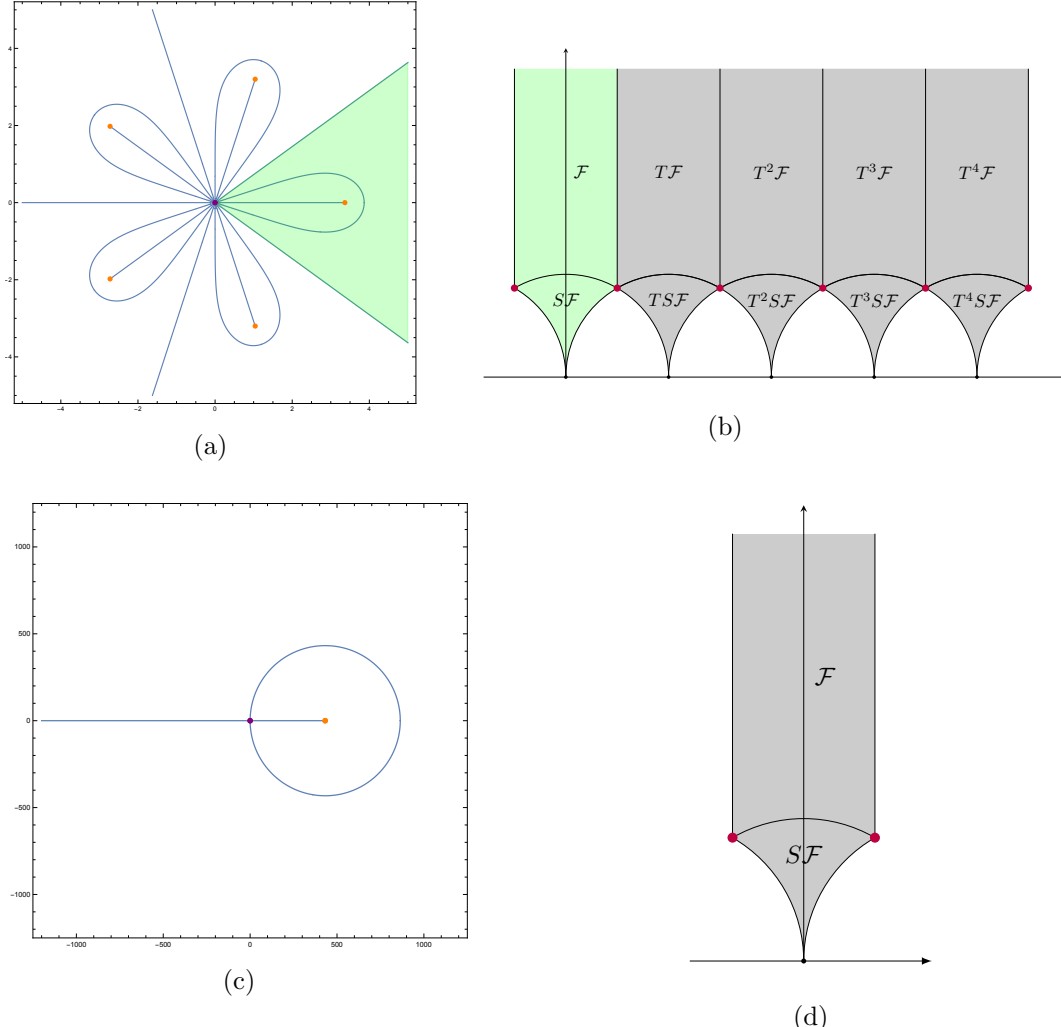

(a)

(b)

(c)

(d)

Figure 2: The $\mathbb{Z}_5$-folding of the $D_{S^1}E_4$ theory. Top: The $\mathbb{Z}_5$ symmetric configuration of $E_4$ with branch point (purple) at $U = 0$, which corresponds to the $II$ singular fibre of the $E_4$ configuration. From the fundamental domain, it is also clear that the branch point is the fixed point of the folding, as it is identified under $T : \tau \mapsto \tau + 1$. By quotienting the $\mathbb{Z}_5$ symmetry, we find the configuration $(I_1; I_1, II^*)$ of the $D_{S^1}E_8$ theory. On the $U$-plane, it is created by the image of a wedge (green) under that map. Bottom: The resulting $U$-plane and fundamental domain contain the same branch point, corresponding now to a $II^*$ singularity.

perspective, the appearance of new singular fibres at the origin of the fixed point of the symmetry corresponds to insertions of 7-branes. However, since there is no 7-brane configuration that would produce a deficit angle that corresponds to a $\mathbb{Z}_5$ symmetry, such a discrete gauging has not been discussed in [38]. We will revisit this construction in the following subsection.

## 3.3 Symmetries of $(p, q)$-brane webs

The foldable symmetries of the $U$-plane, as given in Table 2, almost coincide with the symmetries of the underlying $(p, q)$-brane webs, which were recently analysed in

[38, 39] (see also [46, 47]). These symmetries were discussed for the case of vanishing mass parameters of the 5d SCFTs, and the corresponding brane-web folding procedure was referred to as *S-folding*, being a generalisation of the four-dimensional S-folds introduced in [49–52]. In this section, we review five-dimensional S-folding and compare it to CB folding.

**S-folds.** Consider a five-dimensional superconformal field theory $\mathcal{T}$, whose brane web possesses a $\mathbb{Z}_k$ symmetry. That is, for a particular choice of the deformation parameters[13] and of the axio-dilaton, the brane web remains invariant under a $\mathbb{Z}_k \subset$ SL$(2, \mathbb{Z})$ action of the IIB string theory, combined with a $\mathbb{Z}_k$ spatial symmetry action. In this case, the web can be split into $k$ distinct parts, which are exchanged by the $\mathbb{Z}_k$ action. The S-fold $\mathcal{T}/\mathbb{Z}_k$ of the theory is then a quotient by this $\mathbb{Z}_k$ action, which produces a deficit angle at the fixed point of the $\mathbb{Z}_k$ symmetry that corresponds to a 7-brane configuration. The only possibilities for such 7-branes are listed below:

| **7-brane** | $D_4$ | $E_6$ | $E_7$ | $E_8$ |
|:---:|:---:|:---:|:---:|:---:|
| $\mathbb{Z}_{\boldsymbol{k}}$ | $\mathbb{Z}_2$ | $\mathbb{Z}_3$ | $\mathbb{Z}_4$ | $\mathbb{Z}_6$ |
| $\mathbb{M}_{\boldsymbol{F}}$ | $I_0^*$ | $IV^*$ | $III^*$ | $II^*$ |

$$(3.28)$$

In the last row of the above table, we also give the CB singularities that reproduce the correct monodromy matrix $\mathbb{M}_F$ of the 7-brane. In particular, the 7-brane insertions reproduce the singularities obtained from the discrete gauging of a free vector multiplet (3.6), as previously anticipated. Thus, these S-folds can only reproduce the CB foldings for which the fixed point of the $\mathbb{Z}_k$ symmetry is a smooth $I_0$ fibre. Importantly, however, S-folding and discrete 0-form symmetry gaugings are distinct operations. In the former, the 7-brane configuration introduces new degrees of freedom in the folded theory. As a result, the Kodaira singularity located at the fixed point of the $\mathbb{Z}_k$ symmetry can be further deformed, as opposed to the frozen singularity appearing in the discretely gauged model.

What remains unclear at this stage is the precise relation between the CB symmetries and the symmetries of the $(p, q)$-web. The link between the mirror curve and the brane web can be made explicit through the amoeba projection of the mirror curve [116]. Recall from [89, 90] that the CY3 mirror can be expressed as a double fibration over a complex plane $W$, with the elliptic fibre given by the SW curve, which, in its most natural form, is written in terms of two $\mathbb{C}^*$ variables, $t$ and $w$. The amoeba projection is obtained as

$$(t, w) \mapsto (r_t, r_w) = (\log |t|, \log |w|) , \qquad (3.29)$$

where $(r_t, r_w)$ are the amoeba coordinates, with $r_{t,w} \in \mathbb{R}$. Then, the brane web appears as the *spine* of the amoeba projection - *i.e.* the legs stretching out to infinity. To obtain the amoebæ, we need to fix both the mass parameters as well

---

[13]We follow the setup of [38], where the mass parameters are turned off.

as the Coulomb branch parameter $U$. We will implement the amoeba projection algorithmically following [117].

**The local $\mathbb{P}^2$ geometry.** For the toric phases of the $E_n$ theories with $n \leq 5$, the spine of the amoeba projection is always dual to the corresponding $dP_n$ or $\mathbb{F}_0$ geometry. Consider, for instance, the $\mathbb{P}^2$ geometry, whose mirror curve reads

$$\frac{1}{w} + \frac{1}{t} + wt - 3U = 0 \ . \tag{3.30}$$

In the Weierstraß form of the curve, the $U$-plane singularities reside at $U^3 = 1$. To find the loci of the singular points for the toric curve, one computes the minors of the Newton polygon $F(t, w) = t + w + w^2 t^2 - 3Utw$, for fixed values of $U$. These lead to the following solutions:

$$(1, 1, 1) \ , \qquad \left( e^{\frac{2\pi i}{3}}, e^{\frac{2\pi i}{3}}, e^{\frac{4\pi i}{3}} \right) \ , \qquad \left( e^{\frac{4\pi i}{3}}, e^{\frac{4\pi i}{3}}, e^{\frac{2\pi i}{3}} \right) \ , \tag{3.31}$$

for the tuples $(t, w, U)$. These points turn out to be the fixed loci of the $\mathbb{Z}_3^{\text{web}}$ symmetry exchanging the $\mathbb{C}^*$ coordinates as follows:

$$(t, w) \mapsto \left( w, \frac{1}{wt} \right) \ . \tag{3.32}$$

This symmetry leaves the mirror curve invariant, and, from (3.29), it is clear that it is preserved under the amoeba projection. Put differently, the $\mathbb{Z}_3^{\text{web}}$ symmetry of the brane web is nothing but the symmetry inherited from (3.32).

On the other hand, the $\mathbb{Z}_3^U$ symmetry of the $U$-plane implies, in particular, that under changes $U \to \rho U$, for $\rho^3 = 1$, the mirror curve remains unchanged. To see this explicitly, we accompany this action on $U$ by the coordinate rescaling $(t, w) \to \rho^{-1}(t, w)$ such that the curve becomes:

$$\frac{1}{w} + \frac{1}{t} + \frac{wt}{\rho^3} - 3U = 0 \ , \tag{3.33}$$

and since $\rho^3 = 1$, this is identical to (3.30). Then, the $\mathbb{Z}_3^U$ symmetry exchanges the points (3.31), as it ought to. Note, however, that under the amoebæ projection, this symmetry is lost. The amoebæ projections for various values of $U$ are depicted in Figure 3. Due to the $\mathbb{Z}_3^U$ symmetry of the $U$-plane, for any $U \in \mathbb{C}$ the amoebæ projections for $U$, $e^{\frac{2\pi i}{3}} U$ and $e^{-\frac{2\pi i}{3}} U$ are identical.

We have established that the Coulomb branch $\mathbb{Z}_3^U$ symmetry is inherently different from the brane-web symmetry $\mathbb{Z}_3^{\text{web}}$. This is perhaps not surprising, as the Coulomb branch symmetry is an artifact of the KK theory, instead of the purely five-dimensional model. That is, on the Coulomb branch, this symmetry appears as a subgroup of the $\text{U}(1)_{\mathbf{r}}$ symmetry of the effective 4d $\mathcal{N} = 2$ KK theory. Meanwhile, the brane-web symmetry $\mathbb{Z}_3^{\text{web}}$ is inherited from a symmetry of the mirror threefold.

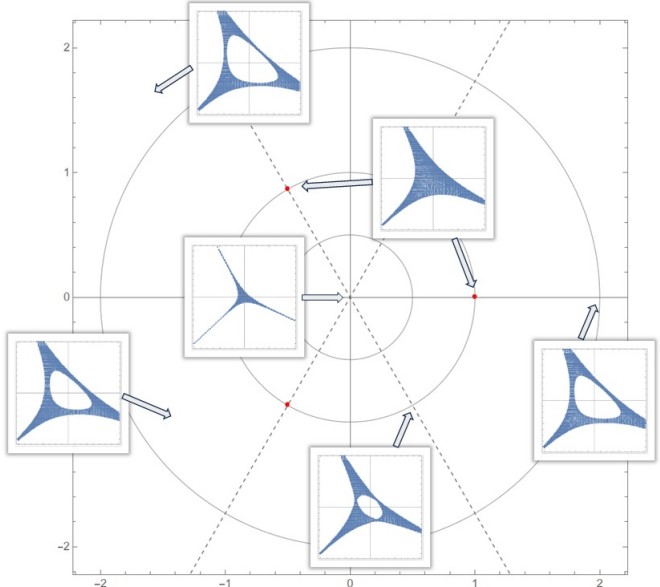

Figure 3: Amoebæ for $D_{S^1}E_0$, for various points on the $U$-plane. $U$-plane singularities lie at $U^3 = 1$, represented by red dots, while the dotted lines are axes of symmetry. The $\mathbb{Z}_3^U$ symmetry is preserved by the amoebæ projection projection: The amoebæ projections for $U$ and $U'$ are identical if $U$ and $U'$ are related by a $\mathbb{Z}_3^U$ rotation.

Nevertheless, brane-web symmetries still appear in general from a fine-tuning of the deformation parameters. Thus, a rather appealing question is whether a symmetry of the brane-web would necessarily imply the existence of a Coulomb branch symmetry. The reverse statement is clearly false, due to the above considerations, as can also be seen from the lack of a $\mathbb{Z}_5$ symmetry for the brane-web of the $E_4$ SCFT. As the $E_0$ theory is too simple from this perspective due to the lack of mass parameters, we will present some evidence that the direct implication is true by considering the $\mathbb{F}_0$ geometry.

**The local $\mathbb{F}_0$ geometry.** In the conventions of [22], the mirror curve of the $E_1$ theory can be written as

$$w + \frac{1}{w} + \sqrt{\lambda}\left(t + \frac{1}{t}\right) - U = 0 \ . \tag{3.34}$$

The singular points of the curve are given by the following tuples for the $(t, w, U)$ variables: $(-1, 1, 2-2\sqrt{\lambda})$, $(1, -1, -2+2\sqrt{\lambda})$, $(-1, -1, -2-2\sqrt{\lambda})$ and $(1, 1, 2+2\sqrt{\lambda})$. There are multiple symmetries for this curve. First, the symmetries

$$\mathbb{Z}_2^w : \ w \mapsto \frac{1}{w} \ , \qquad \mathbb{Z}_2^t : \ t \mapsto \frac{1}{t} \ , \tag{3.35}$$

leave the above singular points invariant, and are preserved by the amoebæ projection. Then, the CB symmetry $\mathbb{Z}_2^U$ is generated by $(t, w, U) \mapsto -(t, w, U)$, exchanging

the curve singularities pairwise. Note that this symmetry is not preserved by the amoebæ projection. Finally, the CB shows a $\mathbb{Z}_4^U$ symmetry whenever $\lambda = -1$ [22]. This symmetry acts as $(t, w, U) \rightarrow (w, -t, i\,U)$, but only a $\mathbb{Z}_2$ subgroup of this symmetry is preserved on the amoebæ.

Let us also note that the $\mathbb{Z}_4^{\text{web}}$ symmetry of the brane-web appears at $\lambda = 1$, being generated by $(t, w, U) \rightarrow \left(w, \frac{1}{t}, U\right)$. Importantly, this $\mathbb{Z}_4^{\text{web}}$ symmetry does not act on the CB parameter of the KK theory. Let us remark, however, that the mass parameters of the SW curve are exponentiated 5d mass parameters, which could also include non-trivial holonomies. Thus, from a five-dimensional point of view, the values $|\lambda| = 1$ would correspond to the same point on the extended Coulomb branch, namely the SCFT point.

This example appears to suggest that the brane-web symmetries do indeed imply the existence of similar symmetries on the Coulomb branch. This implication could be proved through a chain of string theory dualities, as exemplified in [46]. In fact, equation (3.24) already shows that most, if not all, of the CB symmetry enhancements can occur by only turning on some non-trivial holonomies along the $S^1$ direction. Furthermore, such holonomies can lead to new symmetry enhancements, which are not available in the brane-web picture.

## 3.4    Fractional foldings

Five-dimensional $\mathbb{Z}_k$ S-folds involve cutting the 5-brane web into $k$ slices (which are exchanged under the $\mathbb{Z}_k$ symmetry) and keeping only one of these slices while inserting a specific 7-brane at the fixed point of the $\mathbb{Z}_k$ action [38]. For the usual 5d $S$-folds, the only possible 7-branes are $D_4, E_6, E_7$ and $E_8$, which correspond to $\mathbb{Z}_2, \mathbb{Z}_3, \mathbb{Z}_4$ and $\mathbb{Z}_6$ quotients, respectively, as listed in (3.28). In [39], a new type of S-folds was introduced, which makes use of other types of 7-brane configurations:

$$
\begin{array}{c|c|c|c}
\textbf{7-brane} & H_2 & H_1 & H_0 \\
\hline
\mathbb{Z}_k & \mathbb{Z}_3 & \mathbb{Z}_4 & \mathbb{Z}_6 \\
\hline
\mathbb{M}_F & IV & III & II
\end{array}
\tag{3.36}
$$

As before, we indicate the brane-web symmetries relevant for each insertion, as well as the monodromy associated with the 7-branes [50]. This new construction was termed *fractional folding* and can be realised as follows. As before, consider a brane web with a $\mathbb{Z}_k$ symmetry (for $k = 3, 4$ or 6) and split it into $k$ slices exchanged by this symmetry. Then, the partial quotient procedure involves removing only one of these slices and introducing an appropriate 7-brane configuration at the fixed point of the $\mathbb{Z}_k$ action. The final configuration should be consistent with the deficit angle of the 7-brane.

**Fractional foldings on the $U$-plane.** To understand how this procedure could be implemented on the Coulomb branch, recall that for a theory $\mathcal{T}$ with fibre at infinity $F_\infty = I_n$, a $\mathbb{Z}_k$ symmetry is allowed only if $k$ divides $n$. As a result, there must exist a positive integer $\widetilde{n}$, such that

$$n = \widetilde{n}\,k \ . \tag{3.37}$$

Then, we naively associate a partial contribution $I_{\widetilde{n}}^\infty = I_{n/k}^\infty$ to every slice exchanged by the $\mathbb{Z}_k$ symmetry of the original theory. Thus, under the partial folding keeping $(k-1)$ slices, the new fibre at infinity becomes

$$F_\infty = I_n \ \xrightarrow[\text{partial}]{\mathbb{Z}_k} \ F_\infty = I_{(k-1)\widetilde{n}} = I_{n-\frac{n}{k}} \ . \tag{3.38}$$

This change in the fibre at infinity is, of course, accompanied by the previously mentioned change in the bulk fibres: the bulk singularities appearing in one of the slices are exchanged with a $IV, III$ or $II$ fibre at the fixed point of the $\mathbb{Z}_k$ symmetry (*i.e.* the origin of the $U$-plane), for $k = 3, 4$ or $6$, respectively. The remaining slices are glued together, leading to a $\mathbb{Z}_{k-1}$ symmetry in the new theory.

The construction from the point of view of the rational elliptic surface looks as follows. Let us denote by $\mathcal{S}_n$ a RES characterised by a fibre at infinity $F_\infty = I_n^\infty$. In the case where $\mathrm{Aut}_\sigma(\mathcal{S}_n)$ contains a $\mathbb{Z}_k$ group, the quotient map $f_k : \mathcal{S}_n \to \mathcal{S}_{\frac{n}{k}} = \mathcal{S}_n/\mathbb{Z}_k$ is well-defined, and results in a new RES with $F_\infty = I_{\frac{n}{k}}$. Then, we interpret the construction of fractional foldings as a composition of a folding and an *unfolding* (*i.e.* an inverse of a folding): the surface $\mathcal{S}_n$ has a fractional $k$-folding, if there exists a surface $\mathcal{S}_{\frac{n}{k}(k-1)}$ with $\mathrm{Aut}_\sigma(\mathcal{S}_{\frac{n}{k}(k-1)}) \supset \mathbb{Z}_{k-1}$, such that $f_{k-1}(\mathcal{S}_{\frac{n}{k}(k-1)}) = \mathcal{S}_{\frac{n}{k}}$. Then the fractional folding is the composition

$$\mathrm{f}_{\frac{k}{k-1}} := (f_{k-1})^{-1} \circ f_k : \mathcal{S}_n \longrightarrow \mathcal{S}_{\frac{n}{k}(k-1)} \ . \tag{3.39}$$

This reproduces precisely the list of rank 1 examples discussed in [39], namely:

$$
\begin{aligned}
\mathrm{f}_{\frac{2}{3}} : &\quad E_6 : (I_3^\infty; 3I_3) \dashrightarrow (I_1^\infty; IV^*, I_3) \dashrightarrow E_7 : (I_2^\infty; IV, 2I_3) \ , \\
\mathrm{f}_{\frac{3}{4}} : &\quad E_1 : (I_8^\infty; 4I_1) \dashrightarrow (I_2^\infty; III^*, I_1) \dashrightarrow E_3 : (I_6^\infty; III, 3I_1) \ , \\
\mathrm{f}_{\frac{5}{6}} : &\quad E_3 : (I_6^\infty; 6I_1) \dashrightarrow (I_1^\infty; II^*, I_1) \dashrightarrow E_4 : (I_5^\infty; II, 5I_1) \ ,
\end{aligned}
\tag{3.40}
$$

for the specific configurations with $\mathbb{Z}_3$, $\mathbb{Z}_4$ and $\mathbb{Z}_6$ symmetries. For completeness, we also introduce the intermediate step (3.39) of the above procedure. Note that the $\mathrm{f}_{\frac{5}{6}}$ fractional folding contains an unfolding of the peculiar $\mathbb{Z}_5$ symmetry of the $D_{S^1}E_4$ theory, as discussed in Section 3.2.3.

Since a $k$-unfolding maps the fibres at infinity as $I_{9-n}^\infty \mapsto I_{k(9-n)}^\infty$, this maps a $D_{S^1}E_n$ theory to a $D_{S^1}E_{9-k(9-n)}$ theory. We obtain the following Table 3 of theories related by foldings and fractional foldings.

|       | $E_0$ | $E_1$ | $E_2$ | $E_3$ | $E_4$ | $E_5$ | $E_6$ | $E_7$ | $E_8$ |
|-------|-------|-------|-------|-------|-------|-------|-------|-------|-------|
| $E_0$ |       |       |       |       |       |       | $\mathbb{Z}_3$ |       |       |
| $E_1$ |       |       |       | $f_{\frac{3}{4}}$ |       | $\mathbb{Z}_2$ |       | $\mathbb{Z}_4$ |       |
| $E_2$ |       |       |       |       |       |       |       |       |       |
| $E_3$ |       |       |       |       | $f_{\frac{5}{6}}$ |       | $\mathbb{Z}_2$ | $\mathbb{Z}_3$ | $\mathbb{Z}_6$ |
| $E_4$ |       |       |       |       |       |       |       |       | $\mathbb{Z}_5$ |
| $E_5$ |       |       |       |       |       |       |       | $\mathbb{Z}_2$ | $\mathbb{Z}_4$ |
| $E_6$ |       |       |       |       |       |       |       | $f_{\frac{2}{3}}$ | $\mathbb{Z}_3$ |
| $E_7$ |       |       |       |       |       |       |       |       | $\mathbb{Z}_2$ |
| $E_8$ |       |       |       |       |       |       |       |       |       |

Table 3: Matrix of $E_n$ theories related by $\mathbb{Z}_k$ foldings or $f_{\frac{k}{k-1}}$ fractional foldings. The action is always taken on the theories represented at the beginning of each row. We omit the inverse directions.

It is again relatively straightforward to compute the prepotential of the new theory, given the change in the fibre at infinity in (3.38). Indeed, from (3.17), we have for the rank-one theories:

$$\mathcal{F}_{\mathcal{T}^{k-1}/\mathbb{Z}_k} \approx \frac{k-1}{k}\mathcal{F}_{\mathcal{T}} \ , \qquad (3.41)$$

in agreement with [39], where by $\mathcal{T}^{k-1}/\mathbb{Z}_k$ we denote the $(k-1)/k$ partial quotient of $\mathcal{T}$.

As it was the case for the 5d S-folds, the CB singularity appearing at the fixed point of the $\mathbb{Z}_k$ symmetry can be further deformed, as it corresponds to a 7-brane configuration. Thus, while the above prescription can be used to reproduce the results of [39], it would be interesting to understand if a field-theoretic parallel to these partial foldings exists. Importantly, in this construction, the new CB singularities should be undeformable, similar to the comparison between S-folds and CB discrete gaugings.

We point out that $\mathbb{Z}_{k-1}$ *unfoldings* of (effective) 4d Coulomb branches can be viewed as gaugings of discrete 2-form symmetries. Nevertheless, this interpretation cannot describe a field theory analogue of a partial $\mathbb{Z}_k$-folding, as $\mathbb{Z}_{k-1}$ is not a subgroup of $\mathbb{Z}_k$. In general, these symmetries of course occur for a fine-tuning of the masses. We leave this problem for future work.

## 3.5 Non-cyclic symmetries

The previously studied Coulomb branch singularities and their quotients all correspond to cyclic subgroups of the induced automorphism group $\mathrm{Aut}_\sigma(\mathcal{S})$. As is clear

from the full classification [25], this group $\mathrm{Aut}_\sigma(\mathcal{S})$ also admits *non-cyclic* subgroups, such as the dihedral groups $D_n$ (of order $2n$) with $n = 2, 3, 4, 6$, and the tetrahedral group $A_4$. The family of rational elliptic surfaces $\mathcal{S}$ supporting non-cyclic symmetries is classified by their singular fibres, as given in [25, Proposition 4.2.3]. In this section, we briefly discuss these non-cyclic symmetries, as well as their physical interpretation, and present some examples for the $D_{S^1}E_n$ theories. A more detailed study, including an organisation of various symmetry groups preserving parts of the surface, can be found in Appendix C.

Recall from Section 2.2.2 that $\mathrm{Aut}_\mathcal{S}(\mathbb{P}^1)$ is defined as the automorphisms on the base $\mathbb{P}^1$ induced by all elements of $\mathrm{Aut}(\mathcal{S})$, being a subgroup of $\mathrm{Aut}_\sigma(\mathcal{S})$. These induced automorphisms act as Möbius transformations

$$\mathrm{Aut}(\mathbb{P}^1) \ni \varphi : U \mapsto \frac{aU + b}{cU + d} \,, \qquad \begin{pmatrix} a & b \\ c & d \end{pmatrix} \in \mathrm{PGL}(2, \mathbb{C}) \,, \qquad (3.42)$$

and, in particular, they preserve the $\mathcal{J}$-invariant of the surface $\mathcal{S}$. The examples studied so far are the cases where $\varphi(U) = \omega_n U$, with $\omega_n$ an $n$-th root of unity. Here, we will study cases where the induced Möbius transformations are not cyclic.

The simplest example of a non-cyclic symmetry is the Klein four-group $D_2 = \mathbb{Z}_2 \times \mathbb{Z}_2$ [25]. This group acts by Möbius transformations on $\mathbb{P}^1$ and is generated by $U \mapsto 1/U$ and $U \mapsto -U$, which correspond to the $\mathrm{PGL}(2, \mathbb{C})$ transformations $s = \left(\begin{smallmatrix} 0 & 1 \\ 1 & 0 \end{smallmatrix}\right)$ and $r = \left(\begin{smallmatrix} -1 & 0 \\ 0 & 1 \end{smallmatrix}\right)$. Clearly, these are inversion and reflection symmetries, with the inversion symmetry being non-cyclic.

This simplest non-cyclic abelian group is realised as the induced automorphism group $\mathrm{Aut}_\mathcal{S}(\mathbb{P}^1)$ for many different surfaces $\mathcal{S}$. Crucially, the inversion $U \mapsto 1/U$ necessarily interchanges the fibre at infinity, defined by $U = \infty$, with a bulk singularity $U = 0$. In order for this inversion to be an induced symmetry, the fibres corresponding to the singularities $U = 0$ and $U = \infty$ must be of the same Kodaira type. However, since we identify the fibre at infinity $F_\infty$ as a characterisation of the 'UV definition' of a given theory, any proper symmetry of a theory must preserve this fibre $F_\infty$. Clearly, from (3.42) any Möbius transformation fixing $U = \infty$ is cyclic, and consequently any non-cyclic automorphism can not be a proper physical symmetry. Nonetheless, the existence of non-cyclic Coulomb branch symmetries is rather curious and they are arguably worthwhile to explore. In the following, we study two explicit examples of curves with both the 'smallest' and the 'largest' possible non-cyclic symmetry.

**Non-cyclic abelian symmetry.** Let us consider a surface whose $U$-plane symmetry is the Klein four-group $D_2 = \mathbb{Z}_2 \times \mathbb{Z}_2$, with precisely the inversion and reflection symmetries described above. An example of such a surface is the $D_{S^1}E_5$ curve with gauge theory parameters $\lambda = 1$ and $M_j = i$ for all $j = 1, \ldots, 4$. Alternatively, this curve is found by setting the $E_5$ characters to $\boldsymbol{\chi} = (-2, -3, 0, 8, 0)$, being a modular

surface, with monodromy group $\Gamma^0(4) \cap \Gamma(2)$ [22]. Rescaling the CB parameter $U = 4u$, the $\mathcal{J}$-invariant becomes

$$\mathcal{J}(u) = \frac{4\left(u^4 + u^2 + 1\right)^3}{27u^4\left(u^2 + 1\right)^2} \ . \tag{3.43}$$

It is straightforward to check that it is invariant under $r : u \mapsto -u$ and $s : u \mapsto 1/u$. The singular fibres are $(I_4^\infty; I_4, 2I_2)$, where $s$ interchanges $I_4^\infty$ with the bulk $I_4$, while exchanging the two $I_2$ factors. We plot the $u$-plane with this $\mathbb{Z}_2 \times \mathbb{Z}_2$ symmetry in Figure 4a. In Appendix C, we explore the possibility of a quotient of the surface by this non-cyclic abelian group.

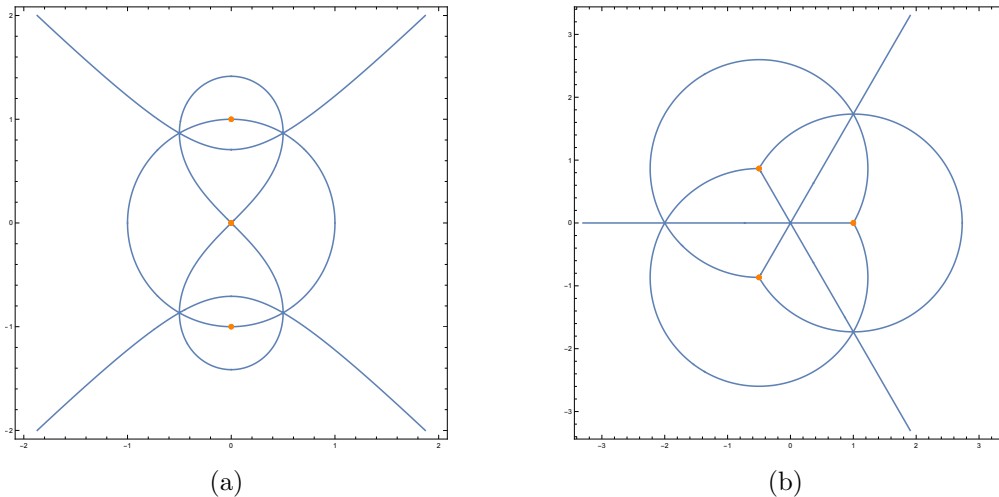

(a)                                 (b)

Figure 4: (a) $D_{S^1}E_5$ configuration with $\mathbb{Z}_2 \times \mathbb{Z}_2$ symmetry. The blue lines constitute the so-called *partitioning* of the $U$-plane, that is, the preimage of the boundary pieces of the $\mathrm{SL}(2, \mathbb{Z})$ fundamental domain inside the fundamental domain for this configuration under the map $u = u(\tau)$, as put forward in [6]. It can be found directly from the $\mathcal{J}$-invariant, and makes the symmetry therefore manifest. (b) $D_{S^1}E_7$ configuration with tetrahedral symmetry $A_4$.

**Tetrahedral symmetry.** Except for the cyclic groups and $\mathbb{Z}_2 \times \mathbb{Z}_2$, the groups $\mathrm{Aut}_{\mathcal{S}}(\mathbb{P}^1)$ are non-abelian – either the dihedral groups $D_n$ for $n = 3, 4, 6$, or the tetrahedral group $A_4$. The 'largest' symmetry groups are therefore the dihedral group $D_6$ and the tetrahedral group $A_4$. Rational elliptic surfaces with tetrahedral symmetry are highly restricted: the possible singular configurations are $(4I_3)$ and $(12I_1)$. The latter, $(12I_1)$, is the generic configuration of the 6d $E$-string curve curve [15, 17]. Meanwhile, the configuration $(I_3; 3I_3)$ can be realised for the $D_{S^1}E_6$ curve with characters $\boldsymbol{\chi} = (0, 0, -3, 9, 0, 0)$. It is a modular elliptic surface, with monodromy group $\Gamma(3)$. In order to make the $A_4$ symmetry manifest, let us rescale $U = 4u$. Then we find that

$$\mathcal{J}(u) = \frac{\left(u^4 + 8u\right)^3}{2^6\left(u^3 - 1\right)^3} \tag{3.44}$$

is invariant under the Möbius transformations $g_1 : u \mapsto \omega_3 u$ and $g_2 : u \mapsto \frac{u+2}{u-1}$, with $\omega_3 = e^{2\pi i/3}$. These indeed generate the tetrahedral group $A_4$ (see also [25, Section 5.3.5]).

The action of $A_4$ on $\mathbb{P}^1$ can be thought of as orientation-preserving transformations of a tetrahedron inside the Riemann sphere $\mathbb{P}^1$, where the four vertices are the $u$-plane singularities $P = (1, \omega_3, \omega_3^2, \infty)$. In terms of the monodromy group $\Gamma(3)$, the $A_4$ symmetry is carried by the cosets, as can be seen from the fact that $\mathrm{SL}(2, \mathbb{Z})/\Gamma(3) \cong A_4$. We plot the $u$-plane with $A_4$ symmetry in Figure 4b.

**Non-induced symmetries.** In order to shed some light on the origin of such non-cyclic symmetries, it is useful to consider all Möbius transformations preserving the $\mathcal{J}$-invariant. It turns out that not all such transformations are induced by $\mathrm{Aut}(\mathcal{S})$. An example is 4d SU(2) SQCD with $N_f = 2$ massless hypermultiplets. Here, in some normalisation, the SW curve is given by

$$\mathcal{J}(u) = \frac{(u^2 + 3)^3}{27 (1 - u^2)^2} , \tag{3.45}$$

which has a configuration $(I_2^*; 2I_2)$ and is modular with monodromy group $\Gamma(2)$. This curve is invariant under the Möbius transformations $m : u \mapsto \frac{u-3}{u+1}$ and $r : u \mapsto -u$. We have $m^3 = r^2 = (mr)^2 = 1$, giving a presentation of the dihedral group $D_3$ of order 6. In this case, the $m$-transformation is induced by a modular transformation $ST : \tau \mapsto -\frac{1}{\tau+1}$, while the $r$-transformation is induced by $T : \tau \mapsto \tau + 1$. Similar to the $\Gamma(3)$ modular surface described above, here the symmetry is recovered as well by a quotient with respect to the monodromy group, $\Gamma/\Gamma(2) \cong S_3 \cong D_3$. Thus in this case again the group of Möbius maps preserving the $\mathcal{J}$-invariant is given by the $\mathrm{SL}(2, \mathbb{Z})$ duality group. That is, in these cases, all $\mathrm{SL}(2, \mathbb{Z})$ electric-magnetic duality transformations act on the Coulomb branch by Möbius transformations. For the $\Gamma(2)$ surface, this $D_3$ symmetry is however not an induced symmetry, since it exchanges the fibre $F_\infty = I_2^*$ with a bulk $I_2$, and therefore does not originate from an automorphism. As was remarked in [118, 119], duality transformations generally do *not* act as Möbius transformations on the base, but rather as rational radical functions. We elaborate further on these aspects in Appendix C.

## 4 Folding across dimensions

Supersymmetric quantum field theories exhibit a rich structure whenever considered on spacetimes containing an $S^1$ factor. Specifically, in the small radius limit, a supersymmetric $d$-dimensional theory can reduce to a disjoint sum of multiple $(d-1)$-dimensional theories differing by their holonomies. In [54, 55], these $(d-1)$-dimensional theories were dubbed *holonomy saddles*. For 5d $\mathcal{N} = 1$ theories on a circle, such phenomena manifest directly on the SW geometry as multiple decoupling

limits of 4d SW theories [35, 36]. The SW geometry of the 5d pure $SU(N)$ theory, for instance, comprises $N$ copies of locally indistinguishable 4-dimensional pure $SU(N)$ theories. Each such holonomy saddle comes with its own 4d BPS quiver, which gives a natural prediction for the 5d BPS quivers in terms of 4d BPS quivers.

In this section, we present a simple argument for detecting holonomy saddles in the formalism of rational elliptic surfaces. This extends the analysis of [35, 36] to theories with fundamental matter, as well as to 6d theories. For rank-one theories, holonomy saddles appear from $\mathbb{Z}_2$-foldings of the $U$-plane, which we discuss in Section 4.1. Meanwhile, in Sections 4.2 and 4.3, we discuss all examples included in this class of theories, which corroborate with massless and massive 4d SQCD theories, respectively.

## 4.1 The 5d–4d map

In Section 2 we have argued that a $\mathbb{Z}_k$-folding of the $U$-plane is given by a map:

$$U \mapsto z = U^k \ . \tag{4.1}$$

This map corresponds to a base-change at the level of the Seiberg–Witten geometry. The discrete gaugings previously discussed in this context were realised as mappings $I_n^\infty \longrightarrow I_{n/k}^\infty$ between the fibres at infinity for the KK theories. These mappings required a quadratic twist on top of the base-change (4.1), in order to preserve the type of fibre at infinity. If such quadratic twists were not added, we would instead have the mapping

$$I_n^\infty \longrightarrow \left(I_{n/k}^*\right)^\infty \ . \tag{4.2}$$

Following (2.7), this appears as a relation between 5d KK geometries and 4d SQCD theories. In fact, we will show that this link has highly non-trivial implications, and could be used to determine the spectra of BPS states of 5d theories based on that of 4d SQCD theories.[14] Having this in mind, the map (4.2) can be argued as follows.

**Foldings across dimensions.** The BPS quiver of a 4d $\mathcal{N} = 2$ theory can be determined directly from the singular fibres of the SW geometry, especially when the CB is *modular* [14]. This map between $U$-plane singularities and nodes of BPS quivers can be directly realised when these CB singularities are of multiplicative type, *i.e.* of $I_n$ type. We would like to give a new interpretation to CB foldings, where the folded and unfolded geometries correspond to seemingly distinct theories. Namely, these should not be related by a discrete gauging. The utility of such an interpretation will be the BPS quiver description.

Accordingly, we are seeking foldings that do not introduce elliptic points, and only involve multiplicative fibres in the bulk. Recall first from (2.17) that the *width* $n$ of the cusp at infinity $F_\infty = I_n$ reduces by a factor of $k$ under a $\mathbb{Z}_k$-folding. At

---

[14]In some particular chamber, and up to a tower of KK states.

the same time, $k$ identical bulk fibres get identified under this operation. Thus, the only way to preserve the Euler number (2.10) of the RES while also avoiding the introduction of elliptic points, is to introduce $I_m^*$ singular fibres in the folded geometry. Nevertheless, in order to have a clear BPS quiver description, these fibres should be the fibres at infinity, describing thus 4d SQCD theories [10, 14, 22].

Let us first assume that all bulk fibres are of $I_1$ type and are related by the $\mathbb{Z}_k$ symmetry. Schematically, we are then looking at the following folding:

$$\left(I_n^\infty;\, (12-n)I_1\right) \xrightarrow{\ \mathbb{Z}_k\ } \left((I_{n/k}^*)^\infty;\, \frac{12-n}{k}I_1\right) . \tag{4.3}$$

The Euler number constraint (2.10) for the new surface reads:

$$\frac{12-n}{k} + \frac{n}{k} = 6 , \tag{4.4}$$

which thus restricts $k$ to the unique value $k = 2$. In turn, this implies that such foldings exist only for even $n$. As the starting point is a field theory with fibre $I_n^\infty$ at infinity with $n$ even, the obvious candidates are the $D_{S^1}E_{2N_f+1}$ theories with $N_f = 0, \ldots, 3$. Thus, the folding realizes the map:

$$\text{5d } SU(2) + 2N_f \text{ fund} \xrightarrow{\ \mathbb{Z}_2\ } 2 \text{ copies of 4d } SU(2) + N_f \text{ fund} . \tag{4.5}$$

In fact, these foldings are related to the notion of holonomy saddles [36, 55], and correspond to decompositions of the 5d BPS quivers in $k = 2$ copies of 4d quivers. Schematically, the 5d quivers take the following form, where $Q$ is a given 4d quiver:

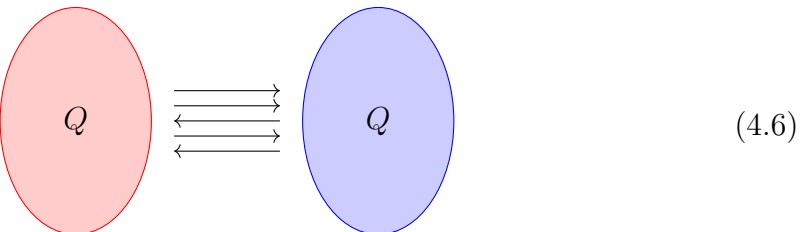

$$\tag{4.6}$$

**Flavour symmetry.** Before we make the folding explicit, let us comment briefly on the relation between the flavour symmetries of the theories. For this, we can consider a more general setup than (4.3) by allowing other types $I_{m_j}$ of singularities, with $m_j \geq 1$. The singular configurations of the $E_{2N_f+1}$ theory that we can $\mathbb{Z}_2$-fold are then necessarily of the form $(I_{8-2N_f};\, \{2n_j I_{m_j}\})$, where $n_j$ and $m_j$ are some integers satisfying

$$\sum_j n_j\, m_j - N_f = 2 . \tag{4.7}$$

This identity simply follows from the Euler number constraint (2.10). As before, the corresponding 4d configurations will be $(I_{4-N_f}^*;\, \{n_j I_{m_j}\})$. Since the folding from

5d to 4d pairwise identifies two $I_{m_j}$ singularities, the action of the $\mathbb{Z}_2$-folding is a reduction of the flavour symmetry algebra $\mathfrak{g}_F$,

$$\mathfrak{u}(1)^{\mathrm{rk}(\Phi)} \oplus \bigoplus_j A_{m_j-1}^{\oplus 2n_j} \xrightarrow{\mathbb{Z}_2} \mathfrak{u}(1)^{\frac{\mathrm{rk}(\Phi)-1}{2}} \oplus \bigoplus_j A_{m_j-1}^{\oplus n_j} \ . \tag{4.8}$$

Clearly, the rank of the MW group also changes across the dimensions. Indeed, for generic masses, the MW group of the $E_{2N_f+1}$ curve has rank $2N_f + 1$, while the corresponding 4d SW surface with $N_f$ generic masses has MW rank $N_f$.

In the following, we characterise the geometry changing $\mathbb{Z}_2$-foldings by quadratic relations between the CB parameters in 4d and 5d.[15] First, we discuss the case of massless SQCD, after which we lay out the structure of the massive theories.

## 4.2 The massless cases

**The $D_{S^1}E_1$ theory.** We start by considering the $D_{S^1}E_1$ theory, with the mirror curve given explicitly in [22]. We find for generic values of $\lambda$ (see also [86, 120, 121]), that

$$U(\tau, \lambda) = \sqrt{8\sqrt{\lambda}\frac{u_0(\tau)}{\Lambda_0^2} + 4(\lambda + 1)} \ , \tag{4.9}$$

where $u_0$ is the CB parameter of the pure 4d $N_f = 0$ theory, being a Hauptmodul of $\Gamma^0(4)$. This offers an alternative proof to the decomposition of the $U$-plane of the $D_{S^1}E_1$ theory in two identical copies of the $u$-plane for the 4d SU(2) theory. This relationship was also recently shown in [36] and holds true for generic values of the deformation parameter $\lambda$. In particular, the $U$-plane has a $\mathbb{Z}_2$ symmetry for any values of the gauge coupling, which is also the centre symmetry of the 5d gauge group SU(2). Thus, the folding that we are considering is:

$$E_1 : \quad (I_8; 4I_1) \xrightarrow{\mathbb{Z}_2} (I_4^*; 2I_1) \ . \tag{4.10}$$

In order to see the decomposition of the BPS quiver, we consider the specific value $\lambda = -1$ of the gauge coupling, where the $\mathbb{Z}_2$ symmetry enhances to $\mathbb{Z}_4$. As a result, it is not difficult to compute the $(a, a_D)$ periods on the $w = U^4$ plane, which lead to the following basis of BPS states (as computed explicitly in [22, 36]):

$$\begin{array}{llll}
\gamma_1 : & (1, 0) \ , & \gamma_2 : & (-1, 2) \ , \\
\gamma_3 : & (-1, 0) \ , & \gamma_4 : & (1, -2) \ ,
\end{array} \tag{4.11}$$

---

[15]An intricate issue regarding the CB parameters in 5d is their normalisation, see [14, Section 3.3] for a discussion. As is clear from the explicit expressions, the normalisation does not affect the structure of the results.

Thus, the associated BPS quiver becomes:

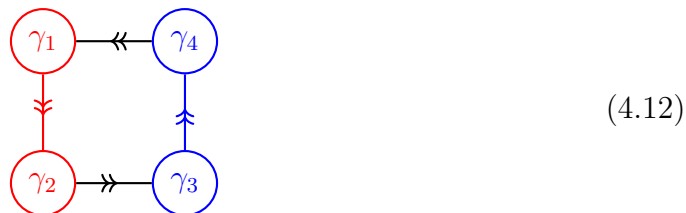

$$(4.12)$$

The two colours used in the above figure indicate the two BPS quivers of the 4d pure SU(2) gauge theory. The full spectrum of this quiver was recently computed in [34, 37] for some particular tame chamber. These works have already indicated that the spectrum of the $D_{S^1}E_1$ theory can be organised as two copies of the maximal chamber for the Kronecker quiver.

**The $D_{S^1}E_3$ theory.** From the previous arguments, another example of (4.3) is:

$$E_3 : \quad (I_6; 6I_1) \xrightarrow{\mathbb{Z}_2} (I_3^*; 3I_1) \ . \tag{4.13}$$

To realise the above map, we need $E_3$ configurations that have a $\mathbb{Z}_2$ symmetry. This can be achieved by setting $M_1 = -M_2 = M$ in the SW curve.

Let us consider the massless 4d $N_f = 1$ theory, whose $u$-plane has a $\mathbb{Z}_3$ symmetry, which is a remnant of the $U(1)_\mathbf{r}$ R-symmetry. Then, if the above map were to exist, we would need a configuration of $D_{S^1}E_3$ exhibiting a $\mathbb{Z}_6 = \mathbb{Z}_2 \times \mathbb{Z}_3$ symmetry. As discussed before, such configurations do exist, but only appear for isolated values of $(\lambda, M)$. For instance, for $(\lambda, M) = (e^{\frac{4\pi i}{3}}, e^{\frac{7\pi i}{6}})$, we find the following identity:

$$U_{E_3}(\tau) = \sqrt{-16 \frac{u_1(\tau, 0)}{\Lambda_1^2}} \ , \tag{4.14}$$

where $u_1$ is the order parameter for the massless 4d SU(2) $N_f = 1$ theory. For these values of the mass parameters it is again not difficult to compute a basis of light BPS states (see [22]), leading to:

$$\begin{array}{llll}
\gamma_1 : & (1, 0) \ , & \gamma_2 : & (1, -1) \ , & \gamma_3 : & (0, -1) \ , \\
\gamma_4 : & (-1, 0) \ , & \gamma_5 : & (-1, 1) \ , & \gamma_6 : & (0, 1) \ ,
\end{array} \tag{4.15}$$

The associated quiver is given below:

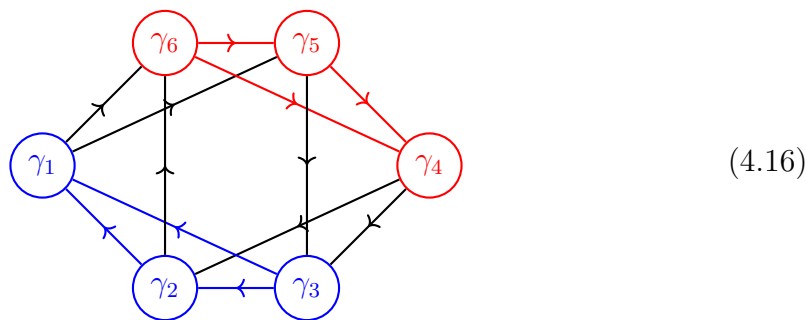

$$(4.16)$$

Fortunately, it was shown in [37, 122] that for this particular quiver there exists a chamber where the BPS spectrum decomposes into two copies of the 4d SU(2) massless $N_f = 1$ theory, together with a tower of KK states. The corresponding 4d sub-quivers are coloured in the above diagram.

We can also find the explicit holonomy saddles from the $D_{S^1}E_3$ theory to *massive* $N_f = 1$ SQCD with a generic mass. For $M_1 = -M_2 = M$, we find agreement of the curves for

$$U(\tau) = \sqrt{16(M\lambda)^{\frac{2}{3}}\frac{u_1(\tau,m)}{\Lambda_1^2} + 4\lambda(1 - M^2) + 4}, \tag{4.17}$$

where $u_1(\tau, m)$ corresponds to the massive 4d $N_f = 1$ curve, and we relate $\Lambda_1 = 4m(M^4\lambda)^{\frac{1}{3}}/(1 - M^2(1 + \lambda))$. This agrees with (4.14) in the massless case. We will extend this example to other massive cases below.

**The $D_{S^1}E_5$ theory.** Consider next the case of $D_{S^1}E_5$, where (4.3) specializes to:

$$E_5 : \quad (I_4; 8I_1) \xrightarrow{\mathbb{Z}_2} (I_2^*; 4I_1) . \tag{4.18}$$

We immediately note that the RHS is a configuration of the 4d SU(2) $N_f = 2$ theory. To realise this folding explicitly, we are again looking for $E_5$ configurations that have at least a $\mathbb{Z}_2$ symmetry.

Consider the massless 4d theory, in which case the $\mathbb{Z}_2$-folding corresponds to:

$$E_5 : \quad (I_4; 4I_2) \xrightarrow{\mathbb{Z}_2} (I_2^*; 2I_2) . \tag{4.19}$$

This $E_5$ configuration was discussed in detail in [22], and can be found, for instance, for the following values of the mass parameters: $M_1 = M_2 = M$, $M_3 = M_4 = -\frac{1}{M}$, with $\lambda = 1$. In this case the $U$-plane has a $\mathbb{Z}_2 \times \mathbb{Z}_2$ symmetry, which turns into $\mathbb{Z}_4$ for the fixed value $M = e^{\frac{\pi i}{4}}$. Note that these $\mathbb{Z}_2$ factors are extensions of the $\mathbb{Z}_2$ residual R-symmetry present on the Coulomb branch of the massless 4d $N_f = 2$ theory. When the $\mathbb{Z}_4$ symmetry is present, the light BPS states are given by [22]:

$$\begin{array}{llll}
\gamma_1, \gamma_5 : & (1, 0) , & \gamma_2, \gamma_6 : & (-1, 1) , \\
\gamma_3, \gamma_7 : & (-1, 0) , & \gamma_4, \gamma_8 : & (1, -1) .
\end{array} \tag{4.20}$$

The associated quiver is depicted below:

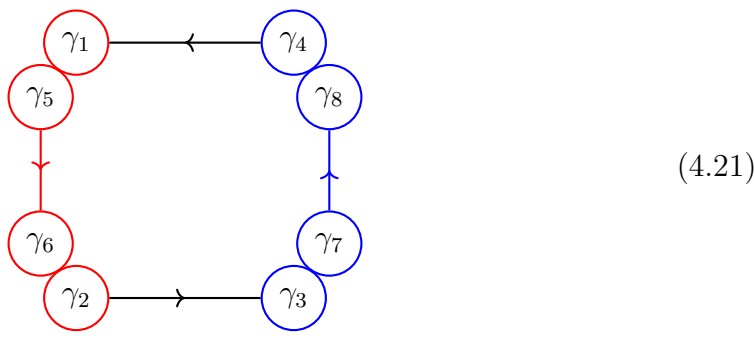

$$\tag{4.21}$$

with the massless 4d SU(2) $N_f = 2$ subquivers coloured. Here we introduce the *block notation* for a BPS quiver [123, 124]. Namely, we group together in a single block the nodes that share the same incidence with all the other nodes in the quiver. For instance, there are four morphisms involving node 1: $X_{12}, X_{16}$, as well as $X_{41}, X_{81}$. We also find:

$$U_{E_5}(\tau, M) = \sqrt{64\frac{u_2(\tau,0)}{\Lambda_2^2} + 4\frac{1+M^4}{M^2}} \ , \tag{4.22}$$

with $u_2(\tau, 0)$ the order parameter of the massless 4d SU(2) $N_f = 2$ theory. As for the $E_1$ and $E_3$ quivers discussed above, it was also shown in [37] that there exists a chamber of the above quiver where the BPS spectrum organises as two distinct copies of spectra of the massless 4d SU(2) $N_f = 2$ quivers, as explained around (4.12).

**The $D_{S^1}E_7$ theory.** Another example of the folding (4.3) involves the $D_{S^1}E_7$ and the 4d SU(2) $N_f = 3$ theories. For the massless 4d theory, this reads:

$$E_7 : \quad (I_2; 2I_4, 2I_1) \xrightarrow{\ \mathbb{Z}_2\ } (I_1^*; I_4, I_1) \ . \tag{4.23}$$

To find the above $E_7$ configuration, we first express the characters in terms of gauge theory parameters. Then, we set the masses to $M_i = 1$ for $i = 1, 2, 3$ and $M_j = -1$ for $j = 4, 5, 6$, while the $\lambda$ parameter does not need to be fixed. In this case, we find:

$$U_{E_7}(\tau, \lambda) = \sqrt{-2^{12}\frac{u_3(\tau,0)}{\Lambda_3^2} - 4\frac{(\lambda^2-1)^2}{\lambda^2}} \ . \tag{4.24}$$

The relevant $\mathbb{Z}_2$ symmetric $D_{S^1}E_7$ quiver is given by:[16]

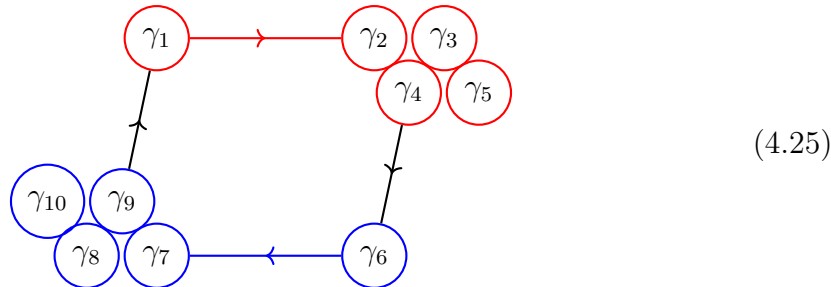

$$\tag{4.25}$$

As before, we use the block notation. We again expect that there is a chamber where the BPS spectrum of the KK theory decomposes into two copies of the massless $N_f = 3$ sub-quivers. To the best of our knowledge, this has not been discussed in the literature.

**Other examples.** It was recently argued in [24] that the global forms of a theory can be detected directly from the Seiberg–Witten geometry. More specifically, while the SW geometry for the 4d pure SU(2) theory is known to be described by the

---

[16]This is mutation equivalent to other known $dP_7$ quivers [123].

configuration $(I_4^*; 2I_1)$, the $\mathrm{SO}(3)_\pm$ global forms were found to be $(I_1^*; I_4, I_1)$, where the $I_4$ singularity is undeformable.

In a similar fashion, as the $D_{S^1}E_1$ theory has both a 0 and a 1-form symmetry, there are multiple global forms allowed for this theory. The geometry that we have mostly dealt with, $(I_8; 4I_1)$, was denoted in [24] as $E_1[\mathrm{SU}(2)]^{[0,1]}$, emphasising the choice of global form for the gauge group. Meanwhile, the $E_1[\mathrm{SO}(3)_\pm]^{[0,1]}$ forms are described by the configurations $(I_2; 2I_4, 2I_1)$. The Coulomb branches of these models can be shown to consist of two distinct copies of the 4d $\mathrm{SO}(3)_\pm$ theories.

### 4.3 Massive holonomy saddles

The correspondence between 4d and 5d theories can be refined in the presence of mass deformations, which allows to test the proposal (4.5) in a large part of the parameter space. While on the 5d side we are able to probe roughly half of the parameter space, in 4d this map characterises the whole parameter space.

When turning on arbitrary masses, the Coulomb branch geometry becomes rather involved, and we have to proceed more systematically in finding the explicit mapping between the CB order parameters.

#### 4.3.1 The 5d–4d dictionary

In order to find a 5d $\mathrm{SU}(2)+2N_f$ phase which allows a $\mathbb{Z}_2$-folding, we need to find the $E_{2N_f+1}$ curves whose functional invariant $\mathcal{J}(U)$ is symmetric, i.e. $\mathcal{J}(-U) = \mathcal{J}(U)$. Let us first focus on $N_f = 0, 1, 2, 3$, and we will comment on $N_f = 4$ below. If we express the $E_n$ curves in terms of the flavour symmetry characters $\boldsymbol{\chi}^{E_n}$, as is done for instance in [15, 17], we find that this $\mathbb{Z}_2$ symmetry appears whenever

$$\chi_{2j+1} = 0 \ , \quad \text{for } j = 1, \dots, N_f \ , \tag{4.26}$$

Indeed, the $E_1$ curve has a $\mathbb{Z}_2$ symmetry for any value of $\chi_1$ or $\lambda$,[17] while for the $E_{2N_f+1}$ curves with $N_f \geq 1$ these give $N_f$ constraints on the vanishing of all odd characters except $\chi_1$. We are then studying the geometry changing $\mathbb{Z}_2$-foldings with generic masses, which features only bulk $I_1$ fibres,

$$E_{2N_f+1}: \quad (I_{8-2N_f}; (2N_f+4)I_1) \xrightarrow{\ \mathbb{Z}_2\ } (I_{4-N_f}^*; (N_f+2)I_1) \ . \tag{4.27}$$

Under this mapping, the $2N_f + 1$ parameters of the $E_{2N_f+1}$ curves, reduce to $N_f + 1$ free parameters, due to the $N_f$ constraints needed for the $\mathbb{Z}_2$ symmetry. These free parameters can be matched with the $N_f$ mass parameters $m_i$ of the 4d $\mathrm{SU}(2)$ curves.

---

[17]For $E_1$, we have $\chi_1 = \sqrt{\lambda} + 1/\sqrt{\lambda}$.

**Matching the functional invariants.** Consider now the functional invariants $\mathcal{J}_{4d}(u_{N_f})$ for the 4d SU(2) SW curves [2], and denote the corresponding invariants for the $E_{2N_f+1}$ KK theories by $\mathcal{J}_{5d}(U_{2N_f+1})$. While it is difficult in general to find solutions to $\mathcal{J}_{4d}(u_{N_f}) = \mathcal{J}_{5d}(U_{2N_f+1})$, the previous considerations suggest the ansatz

$$u_{N_f} = a_{N_f} U_{2N_f+1}^2 + b_{N_f} . \tag{4.28}$$

The existence of such an identity is only possible if the 5d mass parameters are related to the 4d parameters. Assuming that this relation holds, the constants $a_{N_f}$ and $b_{N_f}$ can be easily found by eliminating the first two nonzero coefficients in the large $U_{2N_f+1}$ series of $\mathcal{J}_{5d}(U_{2N_f+1}) - \mathcal{J}_{4d}(a_{N_f} U_{2N_f+1}^2 + b_{N_f})$. Rather surprisingly, these take a very simple form for all $N_f = 0, \ldots, 3$, and we find

$$U_{2N_f+1} = \sqrt{(-1)^{N_f} \frac{2^{\frac{12}{4-N_f}}}{\Lambda_{N_f}^2} \left( u_{N_f} + \frac{1}{4-N_f} [\![ m_j^2 ]\!] \right) + Q_{N_f}(\boldsymbol{\chi})} , \tag{4.29}$$

where $[\![ m_j^2 ]\!] := \sum_{j=1}^{N_f} m_j^2$, and $Q_{N_f}(\boldsymbol{\chi}) \in \mathbb{Q}[\boldsymbol{\chi}]$ are polynomials in the characters $\chi_j$. Concretely, we have:

$$\begin{aligned}
Q_0(\boldsymbol{\chi}) &= 4\chi_1 , \\
Q_1(\boldsymbol{\chi}) &= \tfrac{1}{3}(12\chi_1 + \chi_2^2) , \\
Q_2(\boldsymbol{\chi}) &= \tfrac{1}{2}(12\chi_1 + \chi_4) , \\
Q_3(\boldsymbol{\chi}) &= 10\chi_1 + \chi_6 + 43 .
\end{aligned} \tag{4.30}$$

**The 5d–4d dictionary.** Finding a precise agreement between the 4d and 5d theories requires the curves to agree, under the correspondence (4.29). This can be checked efficiently by calculating the previously mentioned large $U_{2N_f+1}$ series. If the proposal (4.29) were to be trusted, this would collapse the infinite system of equations to a finite one. Indeed, this turns out to be the case, and we find the relations between the 5d characters and the 4d masses as shown below:

$$\begin{aligned}
N_f = 0 : \quad & \varnothing , \\
N_f = 1 : \quad & \chi_2 = 4m , \\
N_f = 2 : \quad & \chi_2 = -3 + 64m_1 m_2 , \\
& \chi_4 = -64(m_1^2 + m_2^2) - 4\chi_1 , \\
N_f = 3 : \quad & \chi_2 = 35 + 2^{12}(64\,T_3 - 3\,T_2) + 10\chi_1, \\
& \chi_4 = 35 - 2^{12}(3\,T_2 + 384\,T_3 - 2^{12}\,T_4) + (60 - 2^{14}\,T_2)\chi_1 + 15\chi_1^2 , \\
& \chi_6 = -15 + 2^{12}\,T_2 - 6\chi_1 .
\end{aligned} \tag{4.31}$$

Here, we define $T_2 = \sum_{j=1}^{3} m_j^2$, $T_3 = m_1 m_2 m_3$ and $T_4 = \sum_{i<j} m_i^2 m_j^2$, while we implicitly divide all masses by the scale $\Lambda_{N_f}$ in order to get dimensionless quantities. The

structure is clear: all odd characters except for $\chi_1$ vanish, while all even characters are polynomials in $\chi_1$ and the 4d masses $m_j$. More specifically, since the SW curves for 4d SQCD can be expressed in terms of $SO(2N_f)$ Casimirs (of order 1, 2 and 4 for $N_f = 1, 2, 3$), the even characters are then polynomials in $\chi_1$ and those Casimirs. Inserting these 4d-5d relations into the map (4.29), we find the universal relation:[18]

$$U_{2N_f+1} = \sqrt{(-1)^{N_f} 2^{\frac{12}{4-N_f}} \frac{u_{N_f}}{\Lambda_{N_f}^2} + 4\chi_1} \qquad (4.32)$$

This relation is interesting for various reasons. First, it should give a non-trivial check of the proposal (4.5) relating 5d BPS quivers to two copies of 4d quivers, which for arbitrary masses can be quite involved. Second, only massive expressions allow to tune to superconformal points. For instance, we can easily construct a 5d configuration which is $\mathbb{Z}_2$-folded to the simplest SQCD configuration featuring an AD point. Using the relation between the 4d and 5d masses (4.31), the $E_3$ configuration $(I_6; 2I_1, 2II)$ found for $\chi^{E_3} = (\chi_1, 3, 0)$ can be $\mathbb{Z}_2$-folded to $(I_3^*; I_1, II)$, i.e. the $N_f = 1$ theory with a type $II$ AD point. Finally, the case $N_f = 3$ is of special interest, as it involves the BPS quivers of the non-toric $E_7$ theory, which are more difficult to find in general.

For any given 4d SQCD configuration $(m_1, \ldots, m_{N_f})$, using (4.31) we can thus find a double cover of the $E_{2N_f+1}$ curve with all characters completely determined, except for $\chi_1$. This character is singled out of course since it is contained in all $E_{2N_f+1}$ curves. It is also clear that in the 4d-5d matching, only one character can remain undetermined: the $E_{2N_f+1}$ curve has $2N_f + 1$ characters, out of which $N_f$ are set to zero in order to obtain the $\mathbb{Z}_2$ symmetry. The remaining $N_f + 1$ characters are matched with the $N_f$ masses, such that one character $-\chi_1-$ remains. The fixed point under the folding on the 5d CB is $U = 0$, while from (4.32) we can see that it corresponds to $u_{N_f}/\Lambda_{N_f}^2 = (-1)^{N_f} 2^{-2(2+N_f)/(4-N_f)}\chi_1$. That is, the distance from the origin of the image of the fixed point under the folding is measured exactly by $\chi_1$.

**Distinction between foldings and geometric engineering.** As a final comment, we want to stress that the above-constructed map from the $D_{S^1} E_{2N_f+1}$ theory to 4d $\mathcal{N} = 2$ SU(2) SQCD with $N_f$ hypermultiplets is not the only map relating the KK theories to 4d SQCD. Of course, another well-studied limit is the geometric-engineering limit in type IIA, where one obtains 4d $\mathcal{N} = 2$ SU(2) gauge theories with $N_f \leq 4$ flavours as a limit of the $D_{S^1} E_{N_f+1}$ theories [131]. In this situation, two out of the $(4 + N_f)$ $I_1$ singularities merge the fibre at infinity, which becomes an $I_{4-N_f}^*$ after a quadratic twist. We clarify this distinction in the following diagram 5. Note that in both cases the 4d pure SU(2) theory can be obtained from the $E_1$ curve.

---

[18]For $N_f = 3$, we furthermore introduce a shift of $u_3/\Lambda_3^2$ by $-7/2^{10}$. Such shifts of $u$ for $N_f = 3$ are necessary in many other contexts [125–130].

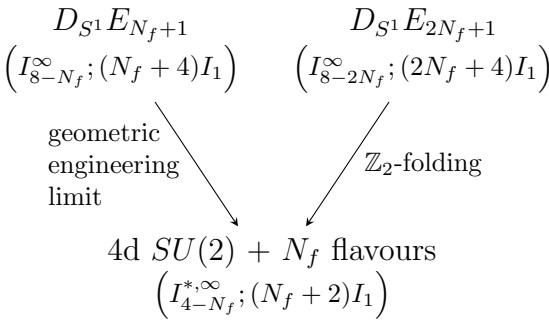

Figure 5: Distinction between the geometric engineering limit and the $\mathbb{Z}_2$-folding of the 4d KK theories to 4d SQCD.

### 4.3.2 6d–4d foldings

The previous analysis suggests that the $N_f = 4$ SCQD theory would be related to the six-dimensional $E$-string theory, which is a 6d $\mathcal{N} = (1,0)$ SCFT [15, 17, 81, 132]. The Seiberg–Witten geometry of the $E$-string theory on $\mathbb{R}^4 \times T^2$ depends modularly on the complex structure $\tau$ of the torus $T^2$ [17, 81]. Upon decoupling mass parameters, this parameter becomes the bare coupling for 4d SU(2) $N_f = 4$.

A relation analogous to (4.32) has already been found in [17] (see also [105]). The Weierstraß invariants $g_2$ and $g_3$ of the $E$-string curve are degree 4 and 6 polynomials with coefficients in the eight masses $M_i$ and the modulus $\tau$ of the torus. In analogy with the above four cases $N_f \leq 3$, the relation to SQCD requires setting all odd coefficients in the curve to zero.[19] Then, schematically we have [17, (6.16)]

$$U_{E\text{-string}} = \sqrt{N_4\, u_4 + c_4}\;, \tag{4.33}$$

where $N_4$ and $c_4$ depend on the modulus $\tau$ and the masses. Additionally, there is an explicit dictionary between the four 4d $N_f = 4$ SQCD masses $m_j$, and the four remaining masses $M_j$ [17, (6.12)–(6.15)]. Together with (4.32), this completes our discussion of massive holonomy saddles (4.27), which for $N_f = 4$ becomes

$$E_9: \quad (I_0; 12I_1) \xrightarrow{\mathbb{Z}_2} (I_0^*; 6I_1)\;. \tag{4.34}$$

This is a special case, where the fibre at infinity $F_\infty = I_0$ is smooth rather than singular.

A similar situation arises in the rank-one M-string theory [133–135], that is, the six-dimensional $\mathcal{N} = (2,0)$ $A_1$ SCFT which is the UV completion of 5d $\mathcal{N} = 2$ SYM. Introducing a mass $m$ for the adjoint breaks the theory to $\mathcal{N} = 1^*$, for which the corresponding SW curve has been studied in the context of integrable systems – see *e.g.* [136]. As was recently discussed in [24, 36], this theory is expected to be related

---

[19]In the conventions of [17, 105], this determines four masses to be zeros of theta functions.

from the point of view of holonomy saddles to two copies of the 4d $\mathcal{N} = 2^*$ theory. We give for completeness the relevant relation [24]

$$U_{\text{M-string}} = \sqrt{-\frac{4}{m^2 \, \wp(i\beta m, \tau_0)} \, u_{\mathcal{N}=2^*} + 1} \, , \qquad (4.35)$$

where $u_{\mathcal{N}=2^*}$ is the 4d $\mathcal{N} = 2^*$ Coulomb branch parameter, $U_{\text{M-string}}$ parametrises the M-string curve, and $\beta$ is the radius of the $S^1$. Meanwhile, $\wp$ is the Weierstraß function. A BPS quiver for the 6d M-string was proposed in [24], by making use of this map.

## 5 Mixed 't Hooft anomaly

Some of the theories that we have discussed so far have both non-trivial global 0-form as well as 1-form symmetries. In such cases, there can be a mixed 't Hooft anomaly between the two types of symmetries. This has the effect that if one symmetry is gauged, the other one disappears. Put differently, if a theory has a mixed 't Hooft anomaly between two symmetries, there is an obstruction to gauging both symmetries.

We have already seen that the group of automorphisms preserving the zero-section of the SW geometry, $\text{Aut}_\sigma(\mathcal{S})$, includes the 0-form symmetries of a given theory. Similarly, the Mordell–Weil group of the SW elliptic fibration, $\text{MW}(\mathcal{S})$, contains the 1-form symmetry of a theory [22]. The discrete gaugings of 1-form symmetries from this perspective was recently done in [24].

Mixed 't Hooft anomalies can often be expressed in terms of short exact sequences and group extensions (see *e.g.* [137, 138] for a review). This leads to the natural proposal that the automorphism group $\text{Aut}(\mathcal{S})$ of an elliptic surface $\mathcal{S}$ – taking the form of a semi-direct product $\text{Aut}(\mathcal{S}) = \text{MW}(\mathcal{S}) \rtimes_\varphi \text{Aut}_\sigma(\mathcal{S})$ – contains information on the mixed 't Hooft anomaly. For the rest of this section, we study the details of this proposal. We leave to Appendix A.2 some more details about the structure of $\text{Aut}(\mathcal{S})$.

### 5.1 Anomaly criterion

The semi-direct product structure of $\text{Aut}(\mathcal{S})$ gives an action of $\text{Aut}_\sigma(\mathcal{S})$ on $\text{MW}(\mathcal{S})$: any automorphism $\alpha \in \text{Aut}_\sigma(\mathcal{S})$ preserving the zero section acts as $\alpha \cdot t_P = t_{\alpha(P)}$, where $t_P$ denotes the translation by a section $P \in \text{MW}(\mathcal{S})$. Moreover, we have a short exact sequence [25, 27]:

$$1 \longrightarrow \text{MW}(\mathcal{S}) \hookrightarrow \text{Aut}(\mathcal{S}) \longrightarrow \text{Aut}_\sigma(\mathcal{S}) \longrightarrow 1 \, , \qquad (5.1)$$

with both $\text{MW}(\mathcal{S})$ and $\text{Aut}_\sigma(\mathcal{S})$ being subgroups of $\text{Aut}(\mathcal{S})$. This structure arises as a natural candidate for describing the mixed anomaly between the 0 and 1-form

symmetries. However, in general, neither of the two factors in $\mathrm{Aut}(\mathcal{S})$ are exactly the 0-form and 1-form symmetries; instead, these symmetry groups generally appear as strict subgroups of the two factors [22, 24]:

$$\Gamma^{(1)} \subset \Phi_{\mathrm{tor}}(\mathcal{S}) \subset \mathrm{MW}(\mathcal{S}) \,, \qquad \Gamma^{(0)} = \mathrm{Aut}_{\mathcal{S}}(\mathbb{P}^1) \subset \mathrm{Aut}_\sigma(\mathcal{S}) \,. \qquad (5.2)$$

Here $\Phi_{\mathrm{tor}}(\mathcal{S})$ denotes the torsion subgroup of $\mathrm{MW}(\mathcal{S})$, while the group $\mathrm{Aut}_\sigma(\mathcal{S})$ is a $\mathbb{Z}_2$ extension of the 0-form symmetry group (see Appendix A.2 for the precise definitions).[20] Importantly, $\Gamma^{(1)} \rtimes \Gamma^{(0)}$ is not a subgroup of $\mathrm{Aut}(\mathcal{S})$, since the product is not well-defined. Indeed, if $\Gamma^{(0)}$ were to act non-trivially on $\Gamma^{(1)}$, then $\Gamma^{(1)} \rtimes \Gamma^{(0)}$ would not be closed under multiplication.

We thus propose the following. Let

$$\mathcal{K}(\mathcal{S}) = \{\alpha \in \Gamma^{(0)} \,|\, \alpha(P) = P, \ \forall P \in \Gamma^{(1)}\} \qquad (5.3)$$

be the subgroup of $\Gamma^{(0)} = \mathrm{Aut}_{\mathcal{S}}(\mathbb{P}^1)$ preserving all torsion elements of $\mathrm{MW}(\mathcal{S})$ that we identify with the 1-form symmetry group $\Gamma^{(1)}$. As we focus on the 0-form symmetries that are cyclic, $\mathcal{K}$ will always be a normal subgroup of $\Gamma^{(0)}$. Thus, by taking a quotient, from this we can define an *anomaly group*

$$\mathcal{A}(\mathcal{S}) = \Gamma^{(0)}/\mathcal{K}(\mathcal{S}) \,. \qquad (5.4)$$

We propose that $\mathcal{A}(\mathcal{S})$ measures the mixed anomaly, and in particular:

$$\mathcal{A}(\mathcal{S}) = 1 \quad \Longleftrightarrow \quad \text{there is no mixed 't Hooft anomaly} \qquad (5.5)$$

Namely, if $\mathcal{A}(\mathcal{S})$ is trivial, then $\Gamma^{(0)}$ does not act on $\Gamma^{(1)}$, so $\Gamma^{(1)} \rtimes \Gamma^{(0)}$ is a well-defined group, being in fact a *direct* product $\Gamma^{(1)} \times \Gamma^{(0)}$. In a similar vein, the semi-direct product $\mathrm{Aut}(\mathcal{S}) = \mathrm{MW}(\mathcal{S}) \rtimes_\varphi \mathrm{Aut}_\sigma(\mathcal{S})$ is actually a direct product if $\ker(\varphi) = \mathrm{Aut}_\sigma(\mathcal{S})$, that is, if $\mathrm{Aut}_\sigma(\mathcal{S})/\ker(\varphi)$ is trivial. However, since $\mathrm{Aut}_\sigma(\mathcal{S})$ is a $\mathbb{Z}_2$ extension of $\Gamma^{(0)}$, one cannot only rely on this latter assertion.

**Mathematical intuition.** By definition, $\mathcal{K}$ is a subgroup of the 0-form symmetry group, $\Gamma^{(0)}$, that preserves the torsion sections which define the 1-form symmetry group of the theory, $\Gamma^{(1)} \subset \mathrm{MW}(\mathcal{S})$. Let us assume that $\Gamma^{(0)} = \mathbb{Z}_k$. Then, upon folding the RES $\mathcal{S}$, as achieved through a change of coordinates $\widetilde{U} = U^k$, the resulting SW geometry $\widetilde{\mathcal{S}}$ looses the $\Gamma^{(0)}$ symmetry.

Recall that the $\Gamma^{(1)}$ torsion sections of $\mathcal{S}$ are rational functions of the CB parameter $U$. Importantly, the sections that are left invariant by the $\Gamma^{(0)}$ action will remain torsion sections of $\widetilde{\mathcal{S}}$. For instance, this is easily seen in a $\mathbb{Z}_2$-folding, as follows:

$$P_{\mathbb{Z}_2} = \left(a\, U^2 + b,\, 0\right) \to \widetilde{P}_{\mathbb{Z}_2} = \left(a\, \widetilde{U} + b,\, 0\right) \,. \qquad (5.6)$$

---

[20]We denote by $\Gamma^{(0)}$ here the subgroup of the full 0-form symmetry that acts non-trivially on the Coulomb branch.

Note that a linear term in $P_{\mathbb{Z}_2}$ would instead imply that the folding does not preserve the torsion section. As such, whenever $\mathcal{K} \cong \Gamma^{(0)}$, the full 1-form symmetry group $\Gamma^{(1)}$ is preserved in the folded RES. Meanwhile, if $\mathcal{K}$ is a strict subgroup of $\Gamma^{(0)}$, only part of $\Gamma^{(1)}$ may be preserved.

Let us note that this argument does not have any implications on the full MW group of the folded surface. We will see in the case of $D_{S^1}E_0$, where $\Gamma^{(0)} \cong \mathbb{Z}_3$ and $\Gamma^{(1)} \cong \mathbb{Z}_3$ in the electric frame, that $\mathcal{K}$ is trivial, thus implying that the 1-form symmetry group is not preserved under a $\mathbb{Z}_3$-folding. Nevertheless, both the folded and unfolded surfaces have $\mathbb{Z}_3$ MW groups. Thus, our assertion is that the MW group of the folded surface does not originate from the 1-form symmetry group of the unfolded theory.

## 5.2   Examples

The criterion (5.5) is sufficient to detect the 't Hooft anomaly in the three 'electric' cases that we consider: the pure 4d SU(2) gauge theory, as well as the $D_{S^1}E_0$ and $D_{S^1}E_1$ KK theories. We furthermore comment on the M-string theory.

**Pure 4d SU(2).**   The pure $\mathcal{N} = 2$ SU(2) theory in 4d has a classical $U(1)_{\mathbf{r}}$ R-symmetry, which in the quantum theory is reduced to a $\mathbb{Z}_8^{(\mathbf{r})}$ symmetry due to the ABJ anomaly. This is spontaneously broken on the CB and acts on the CB parameter as $Z_8^{(\mathbf{r})}$. The $\mathbb{Z}_2^{(1)}$ 1-form symmetry itself is anomaly-free – of course, and gauging it leads to the theory with SO(3) gauge group.

While the SU(2) theory has a mixed anomaly between the $\mathbb{Z}_2^{(1)}$ electric 1-form symmetry and the $\mathbb{Z}_8^{(\mathbf{r})}$ 0-form symmetry, the SO(3) theory does not [96, 139]. This anomaly is related to the $\theta$-angle: in the SU(2) gauge theory, the $\theta$-angle is $2\pi$-periodic; meanwhile, for SO(3) theory has an extended $4\pi$ periodicity for $\theta$, and thus the ABJ anomaly breaks $U(1)_{\mathbf{r}}$ to $\mathbb{Z}_4^{(\mathbf{r})}$ instead.

Let us now check the proposal (5.5). Consider first the SU(2) curve, which has a $\mathbb{Z}_2$ torsion section[21] [24]

$$P_{\mathbb{Z}_2}^{\text{SU(2)}} = \left(\frac{u}{3}, 0\right) \ . \tag{5.7}$$

This section is not invariant under the $\mathbb{Z}_2$ symmetry $(u \mapsto -u) \in \text{Aut}_{\mathcal{S}}(\mathbb{P}^1)$. Thus $\mathcal{A}(\mathcal{S}) = \mathbb{Z}_2$, which agrees with the theory having a mixed anomaly. For the $\text{SO}(3)_{\pm}$ curve, $\text{Aut}_{\mathcal{S}}(\mathbb{P}^1) = 1$ is trivial and so is $\mathcal{A}(\mathcal{S}) = 1$. This again matches with the expectation that the SO(3) theory does not have a mixed anomaly.

The mixed anomaly of the SU(2) theory can also be studied from the perspective of the modular function parametrising the Coulomb branch. For the SU(2) global form, the monodromy group is $\Gamma^0(4)$ and the CB is parametrised by a Hauptmodul of $\Gamma^0(4)$. Gauging the $\mathbb{Z}_2^{(1)}$ symmetry gives the SO(3) curve, where the monodromy group is $\Gamma_0(4)$ and the CB parameter $u_{\Gamma_0(4)}$ is a Hauptmodul for that group [24].

---

[21]See Appendix A.2 for the definition of torsion sections.

The two groups are related by the map $\tau \mapsto \frac{\tau}{4}$, which corresponds to a composition of two 2-isogenies. In the SU(2) theory, the $\mathbb{Z}_2^{(0)}$ $R$-symmetry acts by $u \mapsto -u$, which corresponds to $\tau \mapsto \tau + 2$. Transferring this map along the isogeny to the $\Gamma_0(4)$ curve, it corresponds to $\tau \mapsto \tau + \frac{1}{2}$. Indeed, one can check using doubling formulas such as (A.20) that

$$u_{\Gamma_0(4)}(\tau + \tfrac{1}{2}) = -u_{\Gamma_0(4)}(\tau) . \tag{5.8}$$

The mixed anomaly of the SU(2) theory demands that $u \mapsto -u$ can not be a symmetry of the SO(3) curve. Indeed, we see from the relation above that it is not a (induced) symmetry of the underlying elliptic curve, since the map $\tau \mapsto \tau + \frac{1}{2}$ is not an element in $\mathrm{PSL}(2, \mathbb{Z})$ and the elliptic curve does not transform under it. Rather, it exchanges the $\mathrm{SO}(3)_+$ and $\mathrm{SO}(3)_-$ global forms [24]. This can then be viewed as a remnant of the broken 0-form symmetry.

In some cases, the topological defect of a 0-form symmetry after gauging a 1-form symmetry and coupling to a TQFT becomes a well-defined 3-dimensional non-invertible defect $\mathcal{N}$, satisfying Kramers-Wannier fusion rules [140]. This is also the case for pure $\mathcal{N} = 2$ SYM, where the non-invertible symmetry is subtly incoded in the $\mathbb{Z}_4$ torsion sections of the SO(3) curves, as recently discussed in [24].

**The $D_{S^1}E_1$ theory.** The $E_1$ SCFT admits a deformation to a 5d $\mathcal{N} = 1$ gauge theory with gauge SU(2) gauge group and thus has a global form with a 1-form symmetry $\Gamma_{5d}^{(1)}$. The circle compactification $D_{S^1}E_1$ has the corresponding $\mathbb{Z}_2^{(0)}$ and $\mathbb{Z}_2^{(1)}$ 0-form and 1-form symmetries, which do not have mixed anomalies. To make the distinction clear, we follow the conventions of [24] and denote this as $D_{S^1}E_1[SU(2)]^{[1,0]}$, with the superscript indicating the global symmetries. The $\mathbb{Z}_2^{(0)}$ symmetry acts as $U \to -U$. For $\lambda \neq 1$, the $D_{S^1}E_1$ curve has Mordell–Weil group $\mathrm{MW}(\mathcal{S}) = \mathbb{Z} \oplus \mathbb{Z}_2$ with torsion subgroup $\Phi_{\mathrm{tor}} = \mathbb{Z}_2$ identified with the $\mathbb{Z}_2^{(1)}$ 1-form symmetry. The $\mathbb{Z}_2$ torsion group is generated by [24]

$$P_{\mathbb{Z}_2} = \left( \frac{1}{12}(U^2 - 4\lambda - 4), 0 \right) , \tag{5.9}$$

for generic values of $\lambda$. For $\lambda = 1$, the torsion subgroup enlarges to $\mathbb{Z}_4$. Nevertheless, since the 1-form symmetry is independent of the value of $\lambda$, we can identify the $\mathbb{Z}_2$ subgroup of $\mathbb{Z}_4$ generated by (5.9) with the 1-form symmetry [22]. Since this subgroup is invariant under $U \mapsto -U$, both $\Gamma^{(0)}$ and $\mathcal{K}(\mathcal{S})$ are isomorphic to $\mathbb{Z}_2$, and so $\mathcal{A}(\mathcal{S})$ is trivial, in agreement with the theory being anomaly-free.

One can consider gauging the 1-form symmetry of the KK theory, which leads to some new theory denoted by $D_{S^1}E_1[SO(3)]^{[1,0]}$. This has $\mathbb{Z}_4$ torsion, and we are again only interested in the $\mathbb{Z}_2^{(1)}$ subgroup, generated by

$$P'_{\mathbb{Z}_2} = \left( \frac{1}{24}\left(-U^2 + 4(1 - 6\sqrt{\lambda} + \lambda)\right), 0 \right) . \tag{5.10}$$

This remains invariant under the $\mathbb{P}^1$ automorphism $U \mapsto -U$ as well, and thus $\mathcal{A}(\mathcal{S})$ is again trivial.

The 'magnetic' theory would be obtained by gauging both the 0 and 1-form symmetries of the KK theory, which is equivalent to gauging the whole 1-form symmetry for the 5d SCFT. However, our proposal will not apply in this case.

**The $D_{S^1}E_0$ theory.** The next example is the $E_0$ SCFT, which also has a 1-form symmetry, $\Gamma^{(1)}_{5d} = \mathbb{Z}_3$ [97, 98]. The resulting KK theory, $D_{S^1}E_0^{[1,0]}$, then has $\mathbb{Z}_3^{(0)}$ and $\mathbb{Z}_3^{(1)}$ 0-form and 1-form symmetries. Correspondingly, the $D_{S^1}E_0$ theory has $\mathbb{Z}_3$ torsion, generated by

$$P^{E_0}_{\mathbb{Z}_3,1} = \left(\frac{3}{4}U^2,\, 1\right), \qquad P^{E_0}_{\mathbb{Z}_3,2} = \left(\frac{3}{4}U^2,\, -1\right) . \tag{5.11}$$

The crucial difference compared to the previous cases is that the torsion sections do not respect the $\mathbb{Z}_3$ 0-form symmetry $U \to \zeta_3 U$. Thus, $\mathcal{K}(\mathcal{S})$ is trivial while $\mathcal{A}(\mathcal{S}) = \mathbb{Z}_3$, suggesting an anomaly in this theory, as pointed out in [24]. This anomaly originates from a cubic anomaly for the 1-form symmetry in the 5d SCFT [141]. Thus, gauging the 1-form symmetry in the KK theory will lead to theories without a 0-form symmetry.

**M-string theory.** Finally, let us consider the SW curve of the M-string theory. In the electric frame, the gauge theory interpretation is that of the 5d $\mathcal{N} = 1^*$ theory with gauge group SU(2). As previously argued, this curve is found from the 4d $\mathcal{N} = 2^*$ geometry, upon applying the map: $u_{\mathcal{N}=2^*} \mapsto U^2_{\text{M-string}} + \text{constant}$, or more precisely (4.35) (see also [24]). Then, the torsion sections of the M-string curve will simply follow from those of the 4d $\mathcal{N} = 2^*$ curve.

Recall that the absolute 4d $\mathcal{N} = 2^*$ $SU(2)$ curve is described by the $(I_0^*; I_4, 2I_1)$ configuration of singular fibres. The curve has a $\mathbb{Z}_2$ MW torsion group, with the $\mathbb{Z}_2$ section having the schematic form:

$$P^{\mathcal{N}=2^*}_{\mathbb{Z}_2} = (a + b\, u_{\mathcal{N}=2^*},\, 0) . \tag{5.12}$$

This section intersects non-trivially the bulk $I_4$ fibre, as well as the fibre at infinity. It immediately follows that the corresponding $\mathbb{Z}_2$ section of the M-string theory will only depend on $U^2_{\text{M-string}}$, being thus invariant under the $\Gamma^{(0)} = \mathbb{Z}_2$ action of the 0-form symmetry. Consequently, we have $\mathcal{K} = \mathbb{Z}_2$, leading to a trivial anomaly group. As such, there is no mixed 't Hooft anomaly in this theory, as expected from the analysis of [24].

**Discussion.** Indeed, in all six cases, $\mathcal{A}(\mathcal{S})$ (5.4) is trivial if and only if the theory is free of mixed 't Hooft anomalies. We conclude with some comments. First, it

would be great to have a more conceptual understanding of the group $\mathcal{K}(\mathcal{S})$ defined in (5.3), which determines the subgroup of $\Gamma^{(0)}$ interacting non-trivially with $\Gamma^{(1)}$.

Another intricacy is that the mixed anomalies can have different origins. In pure 4d SU(2), the anomaly can be computed from the Pontryagin square of the background gauge field $B \in H^2(X, \mathbb{Z}_2)$ associated with the $\mathbb{Z}_2^{(1)}$ centre symmetry, schematically of the form $B^2$. On spin manifolds $X$, this anomaly is $\mathbb{Z}_2$ valued, while on non-spin manifolds it is $\mathbb{Z}_4$ valued [96]. For the $E_0$ theory, there is a cubic anomaly for $\Gamma_{5d}^{(1)} = \mathbb{Z}_3$, corresponding to an anomaly term $B^3$ in 6d [141]. In the 4d KK theory, this reduces to a mixed anomaly $B_2^2 B_1$ between the 1-form and 0-form symmetry, with $B_2$ and $B_1$ their associated background gauge fields.

Let us also comment on a possible connection to 2-groups. The semi-direct product $\mathrm{Aut}(\mathcal{S})$ is induced by a homomorphism $\varphi : \mathrm{Aut}_\sigma(\mathcal{S}) \to \mathrm{Aut}(\mathrm{MW}(\mathcal{S}))$, as defined precisely in (A.11) in Appendix A.2. Meanwhile, 0-form and 1-form global symmetries can combine into 2-group global symmetries [97, 142–145]. Since 0-form symmetries can act on the 1-form charges, the symmetry defect demands an action $\rho : \Gamma^{(0)} \to \mathrm{Aut}(\Gamma^{(1)})$. Together with the Postnikov class $[\beta] \in H_\rho^3(B\Gamma^{(0)}, \Gamma^{(1)})$, this action defines a general 2-group $\mathbb{G} = (\Gamma^{(0)}, \Gamma^{(1)}, \rho, [\beta])$ [144]. Our discussion elucidates the way in which the homomorphism $\varphi$ descends to the action $\rho$. It would be interesting to connect these ideas to theories that indeed admit 2-groups, as well as non-invertible symmetries [22, 97, 140, 142–144]. See also [107] for a related approach.

# 6   Discussion and overview

Automorphisms of Seiberg-Witten geometries have been known to lead to discrete symmetry gaugings. In this work, we presented a general framework for analysing surgeries arising as quotients of the SW geometry by its automorphisms and extended these constructions beyond discrete gaugings. A key feature of our methods is the complete classification of automorphisms of rational elliptic surfaces [25, 26]. The SW curves for many higher-rank theories and their automorphisms have been studied already from various perspectives [5, 23, 99, 102, 103, 107, 108, 108, 146–149]. However, to the best of our knowledge, analogous classifications of automorphisms do not exist for these models, which renders a generalisation of our results more challenging.

The automorphisms originating from the Mordell–Weil group can sometimes relate to symmetries of the BPS quivers. Then, quotients by such automorphisms can lead to Galois covers [18, 31]. A more general discussion of these Galois covers will be presented in the future work [32]. Another interesting avenue for future research concerns the relation between 4d BPS quivers and 5d BPS quivers, as discussed in Section 4. Since we refined the correspondence between 4d and 5d theories in the

presence of mass deformations, it would be important to test the proposal (4.5) by an explicit calculation of the BPS spectra for general masses.

Section 3 explored the foldings of the $U$-plane based on the Coulomb branch symmetries. These symmetries are indeed predicted by the classification of automorphisms on the base $\mathbb{P}^1$ induced by automorphisms of the underlying elliptic surface. The classification of these automorphisms $\mathrm{Aut}_{\mathcal{S}}(\mathbb{P}^1)$ contains also non-abelian groups, such as the dihedral groups $D_k$ for $k = 3, 4$ or $6$, and the alternating group $A_4$. Such non-cyclic groups are indeed realised on the Coulomb branches of 4d $\mathcal{N} = 2$ theories. It would be interesting to understand better the physical origin of these non-abelian symmetries, and if a form of gauging of these symmetries is allowed or meaningful and can lead to new theories.

Finally, an important application of SW geometry is the topological twist of 4d $\mathcal{N} = 2$ theories, where topological correlation functions on compact four-manifolds can be expressed as an integral over the Coulomb branch – see *e.g.* [72–74, 121, 126, 128, 150–154]. In particular, correlation functions can be formally expressed as functionals of the rational elliptic surface associated with the theory [75, 76]. In this context, the automorphisms of the SW geometry are expected to leave correlation functions invariant, but their consequences on the topological theories remain to be studied.

## Acknowledgments

We are very grateful to Johannes Aspman, Cyril Closset and Jan Manschot for many interesting discussions, feedback and comments on the draft. EF is supported by the EPSRC grant "Local Mirror Symmetry and Five-dimensional Field Theory". HM was supported by a Royal Society Research Grant for Research Fellows for the most part of this work.

## A    Elliptic surfaces and modular forms

In this appendix, we review definitions and notions used throughout the main part of the paper, including elliptic surfaces, modular forms and congruence subgroups. See *e.g.* [60, 109, 155, 156] for more comprehensive treatments.

### A.1    Weierstraß model

Let us first review some basic aspects of elliptic curves and elliptic surfaces. As explained in Section 2.1, the SW geometry of a rank-one 4d $\mathcal{N} = 2$ theory is a rational elliptic surface $\mathcal{S}$, which we may describe by the Weierstraß model (2.3), namely:

$$y^2 = 4x^3 - g_2(U)\, x - g_3(U) \ . \tag{A.1}$$

The CB singularities correspond to the zero locus of the discriminant

$$\Delta(U) = g_2(U)^3 - 27g_3(U)^2. \tag{A.2}$$

We define the $\mathcal{J}$-invariant (or functional invariant) of the surface as

$$\mathcal{J}(U) = \frac{g_2(U)^3}{\Delta(U)} \ . \tag{A.3}$$

This object can be related to the modular $J$-invariant, which is a function of the complex structure parameter $\tau$ of the elliptic fibre. In this way, we can relate the base parameter $U$ to $\tau$, giving modular as well as non-modular functions $U(\tau)$ in many interesting examples (see *e.g.* [5, 6, 14, 22, 43]). The possible singularity structure of the SW geometry is captured by the Kodaira classification of singular fibres, which is expressed in terms of the orders of vanishing of the Weierstraß invariants $g_2$, $g_3$ and the discriminant:

$$g_2 \sim (U - U_*)^{\mathrm{ord}(g_2)} \ , \qquad g_3 \sim (U - U_*)^{\mathrm{ord}(g_3)} \ , \qquad \Delta \sim (U - U_*)^{\mathrm{ord}(\Delta)} \ . \tag{A.4}$$

The different types of fibres are listed in Table 4. There, we also list the monodromy induced by these singularities on the periods, and the associated flavour symmetry if the singularity is fully deformable. Note that the 4d low-energy description for the $II, III$, and $IV$ fibres is that of Argyres-Douglas SCFTs [3, 4], while the $II^*, III^*$, or $IV^*$ fibres correspond to either the Minahan-Nemeschansky SCFTs [7, 8], or other SCFTs with frozen singularities, such as the Argyres-Wittig theories [16]. Meanwhile, $I_k$ singularities can be described by U(1) gauge theories with $k$ flavours, while $I_k^*$ correspond to SU(2) gauge theories with $N_f = 4 + k$ fundamentals. Note also that all singularities except $I_0^*$ require a fixed value for the complex structure parameter $\tau$ at the singular point.

## A.2 Automorphisms of elliptic surfaces

Consider a rational elliptic surface $\mathcal{S}$. The rational sections $\beta : \mathcal{S} \to \mathbb{P}^1$ of $\mathcal{S}$ form the Mordell–Weil group $\mathrm{MW}(\mathcal{S})$, which is a subgroup of the group of automorphisms of $\mathcal{S}$, $\mathrm{Aut}(\mathcal{S})$ [25–27]. It can be shown (see [25]) that the automorphism group of $\mathcal{S}$ is isomorphic to the semi-direct product[22]

$$\mathrm{Aut}(\mathcal{S}) = \mathrm{MW}(\mathcal{S}) \rtimes_\varphi \mathrm{Aut}_\sigma(\mathcal{S}) \ , \tag{A.5}$$

where $\mathrm{Aut}_\sigma(\mathcal{S})$ is the subgroup of automorphisms preserving the zero section $\sigma$,

$$\mathrm{Aut}_\sigma(\mathcal{S}) = \{\tau \in \mathrm{Aut}(\mathcal{S}) \,|\, \tau(\sigma) = \sigma\} \ . \tag{A.6}$$

---

[22]Recall that for two groups $G$ and $H$, a group homomorphism $\varphi : G \to \mathrm{Aut}(H)$ defines a semi-direct product $H \rtimes_\varphi G \subset H \times G$ with the multiplication $(h_1, g_1)(h_2, g_2) := (h_1\varphi(g_1)(h_2), g_1g_2)$. For $(h, g) \in H \rtimes_\varphi G$, the inverse is found as $(\varphi(g^{-1})(h^{-1}), g^{-1})$.

| Fibre | $\tau$ | ord($g_2$) | ord($g_3$) | ord($\Delta$) | $\mathbb{M}_*$ | $\mathfrak{g}$ |
|:---:|:---:|:---:|:---:|:---:|:---:|:---:|
| $I_k$ | $i\infty$ | 0 | 0 | $k$ | $T^k$ | $\mathfrak{su}(k)$ |
| $I_{k>0}^*$ | $i\infty$ | 2 | 3 | $k+6$ | $PT^k$ | $\mathfrak{so}(2k+8)$ |
| $I_0^*$ | $\tau_0$ | $\geq 2$ | $\geq 3$ | 6 | $P$ | $\mathfrak{so}(8)$ |
| $II$ | $e^{\frac{2\pi i}{3}}$ | $\geq 1$ | 1 | 2 | $(ST)^{-1}$ | - |
| $II^*$ | $e^{\frac{2\pi i}{3}}$ | $\geq 4$ | 5 | 10 | $ST$ | $\mathfrak{e}_8$ |
| $III$ | $i$ | 1 | $\geq 2$ | 3 | $S^{-1}$ | $\mathfrak{su}(2)$ |
| $III^*$ | $i$ | 3 | $\geq 5$ | 9 | $S$ | $\mathfrak{e}_7$ |
| $IV$ | $e^{\frac{2\pi i}{3}}$ | $\geq 2$ | 2 | 4 | $(ST)^{-2}$ | $\mathfrak{su}(3)$ |
| $IV^*$ | $e^{\frac{2\pi i}{3}}$ | $\geq 3$ | 4 | 8 | $(ST)^2$ | $\mathfrak{e}_6$ |

Table 4: Kodaira classification of singular fibres based on orders of vanishing of $g_2$, $g_3$ and $\Delta$.

**The semi-direct product.** We can define a map

$$\psi : \begin{cases} \mathrm{Aut}(\mathcal{S}) \longrightarrow \mathrm{Aut}_\sigma(\mathcal{S}) \,, \\ \tau \qquad \longmapsto \alpha := \psi(\tau) := t_{-\tau(\sigma)} \circ \tau \,, \end{cases} \tag{A.7}$$

which is called the *linearlisation* of $\tau$ [25, 27]. It can be shown that $\psi$ is a group homomorphism. The kernel of this group homomorphism is a normal subgroup of $\mathrm{Aut}(\mathcal{S})$ which acts by translations by sections. Indeed, it is the Mordell–Weil group,

$$\ker(\psi) = \mathrm{MW}(\mathcal{S}) \lhd \mathrm{Aut}(\mathcal{S}) \,. \tag{A.8}$$

In fact, we have a short exact sequence:

$$1 \longrightarrow \mathrm{MW}(\mathcal{S}) \overset{t}{\hookrightarrow} \mathrm{Aut}(\mathcal{S}) \overset{\psi}{\longrightarrow} \mathrm{Aut}_\sigma(\mathcal{S}) \longrightarrow 1 \,, \tag{A.9}$$

with both $\mathrm{MW}(\mathcal{S})$ and $\mathrm{Aut}_\sigma(\mathcal{S})$ being subgroups of $\mathrm{Aut}(\mathcal{S})$. The first non-trivial map is an embedding, defined on smooth fibres by translation,

$$t : \begin{cases} \mathrm{MW}(\mathcal{S}) \longhookrightarrow \mathrm{Aut}(\mathcal{S}) \,, \\ P \qquad \longmapsto t_P \,. \end{cases} \tag{A.10}$$

The Mordell–Weil group is sometimes identified with its image in $\mathrm{Aut}(\mathcal{S})$. The semi-direct product (A.5) requires an action of $\mathrm{Aut}_\sigma(\mathcal{S})$ on $\mathrm{MW}(\mathcal{S})$. This is the group homomorphism

$$\varphi : \begin{cases} \mathrm{Aut}_\sigma(\mathcal{S}) \longrightarrow \mathrm{Aut}(\mathrm{MW}(\mathcal{S})) \,, \\ \alpha \qquad \longmapsto \{\varphi_\alpha : t_P \mapsto \varphi_\alpha(t_P) := t_{\alpha(P)}\} \,, \end{cases} \tag{A.11}$$

Then a general element of $\mathrm{Aut}(\mathcal{S})$ can be written as $\tau = (t_P, \alpha)$ with $P \in \mathrm{MW}(\mathcal{S})$ and $\alpha \in \mathrm{Aut}_\sigma(\mathcal{S})$. The group law on $\mathrm{Aut}(\mathcal{S})$ is

$$
\begin{aligned}
(t_{P_1}, \alpha_1) \circ (t_{P_2}, \alpha_2) &= (t_{P_1} \varphi_{\alpha_1}(t_{P_2}), \alpha_1, \alpha_2) \\
&= (t_{P_1} t_{\alpha_1(P_2)}, \alpha_1 \alpha_2) \\
&= (t_{P_1 + \alpha_1(P_2)}, \alpha_1 \alpha_2) \ .
\end{aligned}
\tag{A.12}
$$

Note that the inverse of $(t_P, \alpha) \in \mathrm{Aut}(\mathcal{S})$ is $(t_{\alpha^{-1}(-P)}, \alpha^{-1})$.

The kernel of the homomorphism (A.11) is a normal subgroup of $\mathrm{Aut}_\sigma(\mathcal{S})$, being given by

$$
\begin{aligned}
\ker(\varphi) &= \{\alpha \in \mathrm{Aut}_\sigma(\mathcal{S}) \,|\, \varphi_\alpha = \mathrm{id}\} \\
&= \{\alpha \in \mathrm{Aut}_\sigma(\mathcal{S}) \,|\, t_{\alpha(P)} = t_P, \forall P \in \mathrm{MW}(\mathcal{S})\} \\
&= \{\alpha \in \mathrm{Aut}_\sigma(\mathcal{S}) \,|\, \alpha(P) = P, \forall P \in \mathrm{MW}(\mathcal{S})\} \\
&\lhd \mathrm{Aut}_\sigma(\mathcal{S}) \ .
\end{aligned}
\tag{A.13}
$$

**Induced automorphisms.** The automorphism group preserving the zero section is related to the automorphism group on the base $\mathbb{P}^1$ in a simple way: For every automorphism $\tau$ of $\mathcal{S}$, there is an induced automorphism $\tau_\mathcal{S}$ on $\mathbb{P}^1$ such that the obvious corresponding diagram with rational sections $\beta : \mathcal{S} \to \mathbb{P}^1$ is commutative. There is another group homomorphism

$$
\phi : \begin{cases} \mathrm{Aut}(\mathcal{S}) \longrightarrow \mathrm{Aut}(\mathbb{P}^1) \ , \\ \tau \hspace{2.5em} \longmapsto \tau_{\mathbb{P}^1} \ , \end{cases}
\tag{A.14}
$$

which associates to every automorphism $\tau$ its action on the base $\mathbb{P}^1$. The automorphism group of $\mathbb{P}^1$ is, of course, $\mathrm{PGL}(2, \mathbb{C})$, which manifests through Möbius transformations. If $\mathcal{S}$ has a non-constant $\mathcal{J}$-map, $\mathrm{Aut}(\mathcal{S})$ is a finite group, and thus the induced automorphisms on the base $\mathrm{Aut}(\mathbb{P}^1)$ is a discrete subgroup of $\mathrm{PGL}(2, \mathbb{C})$. Every finite subgroup of $\mathrm{PGL}(2, \mathbb{C})$ is isomorphic to a finite subgroup of $\mathrm{SO}(3, \mathbb{R})$, which is the group of isometries of the unit sphere. All such subgroups are known: they are the cyclic groups, dihedral groups, the tetrahedral, octahedral and icosahedral groups (see for instance [157, p. 184]).

The group of induced automorphisms on $\mathbb{P}^1$ is defined as the image $\mathrm{Aut}_\mathcal{S}(\mathbb{P}^1) = \phi(\mathrm{Aut}(\mathcal{S}))$. Any automorphism which induces the identity on $\mathbb{P}^1$ and also preserves the zero section acts on each smooth fibre as either the identity or the inversion,[23] and hence $\ker\!\big(\phi|_{\mathrm{Aut}_\sigma(\mathcal{S})}\big) = \mathbb{Z}_2$. This yields $\mathrm{Aut}_\sigma(\mathcal{S})$ as a $\mathbb{Z}_2$ extension of $\mathrm{Aut}_\mathcal{S}(\mathbb{P}^1)$,

$$
1 \longrightarrow \mathbb{Z}_2 \longrightarrow \mathrm{Aut}_\sigma(\mathcal{S}) \longrightarrow \mathrm{Aut}_\mathcal{S}(\mathbb{P}^1) \longrightarrow 1 \ .
\tag{A.15}
$$

---

[23]The inversion here is the inversion of a point on the elliptic fibre with respect to the group law of the elliptic curve. The inverse of a point $P = (x, y)$ is given by $-P = (x, -y)$

Here, the $\mathbb{Z}_2$ can be understood as the map $(x, y) \mapsto (x, -y)$ on the elliptic fibres. For instance, if $\mathrm{Aut}_{\mathcal{S}}(\mathbb{P}^1) = \mathbb{Z}_n$ is cyclic, then [25]

$$\mathrm{Aut}_\sigma(\mathcal{S}) = \begin{cases} \mathbb{Z}_{2n} , & \text{or} \\ \mathbb{Z}_2 \times \mathbb{Z}_n , \end{cases} \tag{A.16}$$

and $\mathrm{Aut}_{\mathcal{S}}(\mathbb{P}^1) \subset \mathrm{Aut}_\sigma(\mathcal{S})$ is a subgroup.

**Mordell–Weil group.** The other component of (A.5) is the Mordell–Weil group $\mathrm{MW}(\mathcal{S})$. It is constructed from the group structure of the elliptic fibres as follows. For an elliptic surface $\mathcal{S}$, the Weierstraß invariants $g_2$ and $g_3$ in (A.1) are valued in $\mathbb{C}(U)$, that is, the field of rational functions of $U$. A *rational section* of this elliptic fibration is a rational solution to the equation (A.1), $P = (x(U), y(U))$, where $x(U)$ and $y(U)$ are in $\mathbb{C}(U)$. By the Mordell–Weil theorem, the sections of $\mathcal{S}$ form a finitely generated abelian group,

$$\mathrm{MW}(\mathcal{S}) \cong \mathbb{Z}^{\mathrm{rk}\,\mathrm{MW}(\mathcal{S})} \oplus \mathbb{Z}_{k_1} \oplus \cdots \oplus \mathbb{Z}_{k_t}. \tag{A.17}$$

This has two components: the free part has $\mathrm{rk}\,\mathrm{MW}(\mathcal{S})$ independent generators, which also define the rank of the group. The point at infinity, $O = (\infty, \infty)$, is the neutral element and does not contribute to the rank. The second component is the torsion subgroup, which we will sometimes denote by $\Phi_{\mathrm{tor}}$. The addition of sections in $\mathrm{MW}(\mathcal{S})$ is given by the standard addition law of rational points of an elliptic curve. In particular, the inverse of a point $P = (x, y)$ is $-P = (x, -y)$, such that $P - P = O$. A section $P$ is $\mathbb{Z}_k$ torsion if $kP = P + P \cdots + P = O$. As such, 2-torsion sections have the particularly simple form $P = (x, 0)$.

The possible Mordell–Weil groups of rational elliptic surfaces have been classified in [158], with the result that $\mathrm{MW}(\mathcal{S})$ is one of 26 distinct groups, with $\mathrm{rk}\,\mathrm{MW}(\mathcal{S}) \leq 8$, and the possible torsion subgroups being $\mathbb{Z}_k$ with $k = 2, 3, 4, 5, 6$, and $\mathbb{Z}_2 \times \mathbb{Z}_2$, $\mathbb{Z}_4 \times \mathbb{Z}_2$, and $\mathbb{Z}_3 \times \mathbb{Z}_3$. The Mordell–Weil group is uniquely determined by the singular configuration of a RES and has been determined for each such configuration in [112].

### A.3 Modular forms and congruence subgroups

The Jacobi theta functions $\vartheta_j : \mathbb{H} \to \mathbb{C}$, $j = 2, 3, 4$, are defined as

$$\vartheta_2(\tau) = \sum_{r \in \mathbb{Z} + \frac{1}{2}} q^{r^2/2} , \quad \vartheta_3(\tau) = \sum_{n \in \mathbb{Z}} q^{n^2/2} , \quad \vartheta_4(\tau) = \sum_{n \in \mathbb{Z}} (-1)^n q^{n^2/2} , \tag{A.18}$$

with $q = e^{2\pi i \tau}$. These functions transform under $T, S \in \text{SL}(2, \mathbb{Z})$ as

$$S : \quad \begin{aligned} \vartheta_2(-1/\tau) &= \sqrt{-i\tau} \, \vartheta_4(\tau) \;, \\ \vartheta_3(-1/\tau) &= \sqrt{-i\tau} \, \vartheta_3(\tau) \;, \\ \vartheta_4(-1/\tau) &= \sqrt{-i\tau} \, \vartheta_2(\tau) \;, \end{aligned} \tag{A.19}$$

$$T : \quad \begin{aligned} \vartheta_2(\tau + 1) &= e^{\frac{\pi i}{4}} \vartheta_2(\tau) \;, \\ \vartheta_3(\tau + 1) &= \vartheta_4(\tau) \;, \\ \vartheta_4(\tau + 1) &= \vartheta_3(\tau) \;. \end{aligned}$$

They furthermore satisfy the doubling formulas

$$\begin{aligned} \vartheta_2(\tfrac{\tau}{2})^2 &= 2\vartheta_2(\tau)\vartheta_3(\tau) \;, \\ \vartheta_3(\tfrac{\tau}{2})^2 &= \vartheta_2(\tau)^2 + \vartheta_3(\tau)^2 \;, \\ \vartheta_4(\tfrac{\tau}{2})^2 &= \vartheta_3(\tau)^2 - \vartheta_2(\tau)^2 \;. \end{aligned} \tag{A.20}$$

We also define the modular $j$-invariant

$$j = 256 \frac{(\vartheta_3^8 - \vartheta_3^4 \vartheta_4^4 + \vartheta_4^8)^3}{\vartheta_2^8 \vartheta_3^8 \vartheta_4^8} \;, \tag{A.21}$$

which is a modular function for $\text{SL}(2, \mathbb{Z})$. Another normalisation we use throughout the text is $J := j/12^3$.

We use modular forms for the congruence subgroups $\Gamma_0(n)$ and $\Gamma^0(n)$ of $\text{SL}(2, \mathbb{Z})$. They are defined as

$$\begin{aligned} \Gamma_0(n) &= \left\{ \begin{pmatrix} a & b \\ c & d \end{pmatrix} \in \text{SL}(2, \mathbb{Z}) \middle| c \equiv 0 \mod n \right\} \;, \\ \Gamma^0(n) &= \left\{ \begin{pmatrix} a & b \\ c & d \end{pmatrix} \in \text{SL}(2, \mathbb{Z}) \middle| b \equiv 0 \mod n \right\} \;, \end{aligned} \tag{A.22}$$

and are related by conjugation. The *principal congruence subgroup* $\Gamma(n)$ is the subgroup of $\text{SL}(2, \mathbb{Z}) \ni A$ defined by elements $A \equiv \mathbb{1} \mod n$. A subgroup $\Gamma$ of $\text{SL}(2, \mathbb{Z})$ is called a congruence subgroup if it contains $\Gamma(n)$ for some $n \in \mathbb{N}$.

**Index.** Let us also study the projective indices of several subgroups of $\text{PSL}(2, \mathbb{Z})$,[24]

$$\Gamma(n) \triangleleft \Gamma_1(n) \triangleleft \Gamma_0(n) \leqslant \Gamma(1) \;. \tag{A.23}$$

Consider first the principal congruence subgroup $\Gamma(n)$. Its index in $\text{PSL}(2, \mathbb{Z})$ is

$$[\text{PSL}(2, \mathbb{Z}) : \Gamma(n)] = \frac{n}{2} J_2(n) = \frac{n^3}{2} \prod_{p|n} (1 - \tfrac{1}{p^2}) \;, \qquad n \geq 3 \;, \tag{A.24}$$

---

[24]See for instance [155, 159]

where the product runs over all prime divisors of $n$, and $J_2$ is Jordan's totient function. For $\Gamma(2)$, the projective index is 6, and this formula does not work since $\Gamma(2)$ is not torsion-free, as it contains $-\mathbb{1}$. For $\Gamma_0(n)$, the index in $\mathrm{PSL}(2,\mathbb{Z})$ is

$$[\mathrm{PSL}(2,\mathbb{Z}) : \Gamma_0(n)] = n \prod_{p|n}(1 + \tfrac{1}{p}) , \qquad (A.25)$$

where the product runs again over prime divisors of $n$. This is precisely the Dedekind psi function, $\psi(n)$. We also have the inclusion $\Gamma(n) \subset \Gamma_0(n)$. The index of $\Gamma(n)$ in $\Gamma_0(n)$ is thus

$$[\Gamma_0(n) : \Gamma(n)] = \frac{n^2}{2} \prod_{p|n}(1 - \tfrac{1}{p}) = \tfrac{n}{2}\phi(n) , \qquad (A.26)$$

with $\phi$ being Euler's totient function.

If $n|m$, then $\Gamma_0(m) \leqslant \Gamma(n)$ is a subgroup. The index of $\Gamma_0(m)$ in $\Gamma_0(n)$ is

$$[\Gamma_0(n) : \Gamma_0(m)] = \tfrac{m}{n} \prod_{\substack{p|m \\ p\nmid n}}(1 + \tfrac{1}{p}) , \qquad (A.27)$$

where the product again runs over primes.

Since $\Gamma(n)$ is normal in $\mathrm{SL}(2,\mathbb{Z})$, we can study the quotients

$$\mathrm{SL}(2,\mathbb{Z})/\Gamma(n) \cong \mathrm{SL}(2,\mathbb{Z}_n) . \qquad (A.28)$$

For small $n$ these have been analysed in [160], where the first few groups are:

| $n$ | 2 | 3 | 4 | 5 | 7 |
|---|---|---|---|---|---|
| $\Gamma/\Gamma(n) \cong$ | $S_3$ | $A_4$ | $S_4$ | $A_5$ | $\mathrm{PSL}(2,\mathbb{F}_7)$ |
| $[\Gamma(1) : \Gamma(n)]$ | 6 | 12 | 24 | 60 | 168 |

$$(A.29)$$

The order of $\Gamma/\Gamma(n)$ matches of course with the index $[\Gamma(1) : \Gamma(n)]$. It is clear that $\mathrm{SL}(2,\mathbb{Z})/\Gamma(n)$ for larger $n$ cannot be isomorphic to $S_m$ or $A_m$, since $|S_m| \sim m!$, while $[\Gamma(1) : \Gamma(n)] \sim n^3$.

# B    Foldings of fundamental domains

In this appendix, we discuss how $U$-plane foldings can be obtained from both modular and non-modular configurations. This construction is relevant in particular in Section 3.

**Foldings from modularity.** An alternative perspective on $U$-plane foldings can be given in terms of the fundamental domains $\mathcal{F}_{\mathcal{T}}$ of theories $\mathcal{T}$. Focusing on the case of modular rational elliptic surfaces with monodromy group $\Gamma' \subset \mathrm{PSL}(2,\mathbb{Z})$, the existence of a $\mathbb{Z}_N$ symmetry on the $U$-plane generally implies the existence of a subgroup $\Gamma \subset \mathrm{PSL}(2,\mathbb{Z})$, such that:

$$\Gamma(1) \geqslant \Gamma \geqslant \Gamma' \, , \qquad [\Gamma : \Gamma'] = N \, . \tag{B.1}$$

As argued in [6, 14, 43, 161, 162], inclusion sequences of this type are in one-to-one correspondence with field extensions of $\mathbb{C}(\Gamma(1))$, *i.e.* the field of meromorphic modular functions of $\Gamma(1)$ over $\mathbb{C}$:

$$\mathbb{C}(\Gamma(1)) \subset \mathbb{C}(\Gamma) \subset \mathbb{C}(\Gamma') \, . \tag{B.2}$$

Such extensions can be also understood from the fundamental domains, as they typically involve a 'folding' of the fundamental domain of $\Gamma'$.[25] For the case where $\Gamma'$ is a *normal* subgroup of $\Gamma$, the proof of the above statement is based on the fact that the Galois group of the field extension is isomorphic to the quotient group [163][26],

$$\mathrm{Gal}\left(\mathbb{C}(\Gamma')/\mathbb{C}(\Gamma)\right) \cong \Gamma/\Gamma' \, . \tag{B.3}$$

When $\Gamma' \leqslant \Gamma$ is a normal subgroup, the quotient $\Gamma/\Gamma'$ is well-defined and forms a group of order $[\Gamma : \Gamma']$. For instance, $\Gamma(n)$ is a normal subgroup of $\Gamma(1)$, where the quotients $\Gamma(1)/\Gamma(n)$ are isomorphic to $\mathrm{SL}(2,\mathbb{Z}_n)$, and for small $n$ there are accidental isomorphisms to the symmetric and alternating groups [160] (see Appendix A.3). However, pairs of subgroups $\Gamma' \leqslant \Gamma$ which are both congruence subgroups of $\mathrm{PSL}(2,\mathbb{Z})$ are usually not normal and thus the Galois group of the field extension cannot be found from (B.3). Indeed, the quotient group $\Gamma/\Gamma'$ is only well-defined if $\Gamma'$ is normal in $\Gamma$.

In Section 3, we find in many examples that modular foldings are allowed if there exist non-trivial cyclic groups $\mathbb{Z}_N$ which induce them. This raises the question: in the case where $\Gamma'$ is not a normal subgroup of $\Gamma$, can we still find cyclic groups in pairs $(\Gamma', \Gamma)$ of subgroups of $\mathrm{PSL}(2,\mathbb{Z})$, even if they are not normal?

Consider first the congruence subgroups $\Gamma_0(n)$ and $\Gamma^0(n)$. It is known that (see for instance [155])

$$\Gamma(n) \triangleleft \Gamma_1(n) \triangleleft \Gamma_0(n) \leqslant \Gamma(1) \, . \tag{B.4}$$

It is important to note that the normality property of subgroups is generally not transitive. However, since $\Gamma(n) \triangleleft \Gamma(1)$, by definition $\Gamma(n)$ is also normal in any subgroup of $\Gamma(1)$ containing it.[27] For instance, we have that $\Gamma(n) \triangleleft \Gamma_0(n)$. The

---

[25]Despite the connection to Galois theory, these foldings will not lead to the Galois covers discussed in [18, 31, 32].

[26]See also Exercise III §3 7(b) of [159].

[27]In particular, any congruence subgroup of level $l$ by definition contains $\Gamma(l)$ as a normal subgroup.

groups $\Gamma_0(n)$ are however never normal in $\Gamma(1)$, and similarly $\Gamma_0(n)$ is never normal in $\Gamma_0(m)$ unless $m = n$.[28]

An approach to define an analogue group for pairs of non-normal subgroups is the following. Let $\Gamma(1) = \mathrm{PSL}(2, \mathbb{Z})$, and let $\Gamma$ be any finite index subgroup of $\Gamma(1)$. Then we can decompose $\Gamma(1)$ into a finite union

$$\Gamma(1) = \bigcup_{\alpha \in C_\Gamma} \Gamma\alpha \tag{B.5}$$

of right cosets, where $C_\Gamma \subset \Gamma(1)$ is a set of right coset representatives of $\Gamma$. The number of coset representatives is of course just the index $|C_\Gamma| = [\Gamma(1) : \Gamma]$ of $\Gamma$ in $\Gamma(1)$.

Then we can consider another subgroup $\Gamma' < \Gamma \leqslant \Gamma(1)$, which we assume is a proper subgroup of $\Gamma$. We can always choose the coset representatives of $\Gamma$ and $\Gamma'$ such that $C_\Gamma \subset C_{\Gamma'}$. We are now interested in relations between $C_\Gamma$ and $C_{\Gamma'}$. Namely, we can find a finite set $Z \subset \Gamma$ with $[\Gamma : \Gamma']$ elements, such that

$$C_{\Gamma'} = \bigcup_{\beta \in Z} \beta C_\Gamma \;. \tag{B.6}$$

The existence of $Z$ is clear, with its elements being precisely the coset representatives of $\Gamma'$ within $\Gamma$. If $\Gamma' \lhd \Gamma$ is a normal subgroup for instance, then $Z$ has a natural group structure, it is precisely the quotient group $\Gamma/\Gamma'$. Our aim is to endow $Z$ with a group structure for some special cases where $\Gamma' \ntriangleleft \Gamma$.

**Foldings of modular surfaces.** We are particularly interested in groups $\Gamma$ and $\Gamma'$ that arise as monodromy groups of modular rational elliptic surfaces [164, 165]. To each modular RES $\mathcal{S}$ we can associate a modular subgroup $\Gamma_\mathcal{S} \leq \Gamma(1)$. As discussed in detail in [6, 14, 22], the singular fibres of $\mathcal{S}$ are then mapped in a 1-to-1 fashion to the cusps and elliptic points of the modular group.[29]

For each RES corresponding to a SW geometry, the singular fibre $F_\infty$ at infinity plays a special role. However, for the monodromy group $\Gamma_\mathcal{S}$, there is no such dedicated cusp since cusps are just $\Gamma_\mathcal{S}$-equivalence classes of $\mathbb{Q} \cup \{\infty\}$. Nevertheless, since the monodromies at infinity in the $U$-plane are $T^n$ for some $n$, this fixes a particular cusp of $\Gamma_\mathcal{S}$: it is the cusp $\tau = \infty$, which is stabilised by an abelian group of translations. This group is generated by $T^{w_{\Gamma_\mathcal{S}}^\infty}$, where $w_{\Gamma_\mathcal{S}}^\infty$ is the width of the cusp $\infty$ for $\Gamma_\mathcal{S}$.

Aside from the fibre at infinity and possible additive fibres, the RES have a singular configuration of the form $(I_{k_1}, I_{k_2}, \ldots, I_{k_m})$. These bulk fibres are matched with the remaining cusps of $\Gamma_\mathcal{S}$. In order to obtain a consistent folding, we need

---

[28]See Problem 1, III §1 of [159].

[29]This is not true for the $I_0^*$ fibres, which are not associated with a cusp but rather an arbitrary point in the upper half-plane, as specified by the RES. We momentarily exclude the surfaces containing $I_0^*$ fibres from the discussion.

to guarantee that the singular configuration is mapped to the fundamental domain in a coherent fashion. For this, we put a further constraint on the choices of cosets (B.5). Each $\alpha \in C_{\Gamma_{\mathcal{S}}}$ maps $\infty$ to the cusp $\alpha(\infty)$ of $\Gamma_{\mathcal{S}}$. One necessary condition for a physically consistent fundamental domain for $\Gamma_{\mathcal{S}}$ is that the singular configuration of $\mathcal{S}$ should be reflected in the decomposition of the index $[\Gamma(1) : \Gamma_{\mathcal{S}}]$ into the widths of all the cusps. Namely, an $I_k$ fibre should correspond to a width $k$ cusp of $\Gamma_{\mathcal{S}}$. The width of a given cusp can be read off from (B.5) as the number of cosets $\alpha \in C_{\Gamma_{\mathcal{S}}}$ such that $\alpha(\infty)$ is equal to that cusp, and that number should equal $k$ if it corresponds to an $I_k$ singular fibre.[30]

With this additional constraint, we are interested in the cases when $Z$, defined in (B.6), has a natural group structure:

> *Whenever $Z \cong \mathbb{Z}_{[\Gamma:\Gamma']}$ is isomorphic to a cyclic group, we say that $\Gamma'$ can be folded to $\Gamma$, and we denote this by $\Gamma' \sqsubset \Gamma$.*

In practice, we will find a choice for $Z$ such that it is generated by one element $g$, *i.e.* $Z = \langle g \rangle_n = \{g^n | n = 0, \ldots, [\Gamma : \Gamma'] - 1\}$, where we identify $g^{[\Gamma:\Gamma']}$ with the neutral element of $Z$. After discussing a few examples, we make this notion more precise below.

**Examples.** Let $\Gamma' = \Gamma(n)$ for $n \geq 2$ and $\Gamma = \Gamma(1)$, then $\Gamma' \triangleleft \Gamma$ is a normal subgroup. The index of $\Gamma'$ in $\Gamma$ is given in (A.24). The quotients $\Gamma/\Gamma'$ for the first few $n$ are isomorphic to symmetric and anti-symmetric groups, and in particular, they are never cyclic unless $n = 1$. Thus $\Gamma(n)$ can never be folded to $\Gamma(1)$.

Consider now the subgroups $\Gamma = \Gamma^0(2)$ and $\Gamma' = \Gamma^0(4)$. We have the coset representatives

$$
\begin{aligned}
C_{\Gamma^0(2)} &= \{\mathbb{1}, S, T\} \,, \\
C_{\Gamma^0(4)} &= \{\mathbb{1}, S, T, T^2, T^3, T^2 S\} \,.
\end{aligned}
\tag{B.7}
$$

It is straightforward to find a subset $Z = \{\mathbb{1}, T^2\} \subset \Gamma^0(2)$ such that (B.6) holds. This subset $Z$ is generated by the element $T^2$, which in $\Gamma' = \Gamma^0(4)$ has order 2. In this way we find that $Z$ is isomorphic to the cyclic group $\mathbb{Z}_2$ of order 2, where 2 is the index of $\Gamma^0(4)$ in $\Gamma^0(2)$. Thus $\Gamma^0(4) \sqsubset \Gamma^0(2)$.

Similarly we find that $\Gamma^0(9) \sqsubset \Gamma^0(3)$, where $Z = \{1, T^3, T^6\}$ is generated by $T^3 \in \Gamma^0(3)$. For $\Gamma(3) \triangleleft \Gamma_0(3)$ we can also easily check that $Z = \{\mathbb{1}, T, T^2\} \cong \mathbb{Z}_3$. In general, for $n \geq 3$ the index of $\Gamma(n)$ in $\Gamma_0(n)$ is $\frac{n}{2}\phi(n)$ (see Appendix A.3). It is equal to $n$ for $n = 1, 2, 3, 4, 6$ but is larger than $n$ for all other integers.

Let us now consider $\Gamma^0(m) \leqslant \Gamma^0(n)$ with $n|m$. A necessary condition for $\Gamma^0(m) \sqsubset \Gamma^0(n)$ is thus the index $[\Gamma^0(n) : \Gamma^0(m)]$ being equal to $\frac{m}{n}$, then it is possible to

---

[30]This constraint makes $\Gamma^0(8)$ for instance not foldable to $\Gamma^0(2)$. This is because $\Gamma^0(8)$ has a fundamental domain which is a four-fold copy of one for $\Gamma^0(2)$, but this choice of domain for $\Gamma^0(8)$ does not reflect the correspondence between singular fibres and widths of the cusps.

write $C_{\Gamma^0(m)}$ as a union $\cup_{j=0}^{m/n} T^{nj} C_{\Gamma^0(n)}$ (this is because $\Gamma^0(m)$ contains $T^m$, which is generated by $\frac{m}{n}$ copies of $T^n \in \Gamma^0(n)$). But from (A.27) we see that this is true if and only if the sets of prime divisors of $m$ and $n$ coincide. In other words, the prime factorisations of $m$ and $n$ have the same bases, but $m$ can have larger exponents. This agrees with the two examples above, $\Gamma^0(4) \sqsubset \Gamma^0(2)$ and $\Gamma^0(9) \sqsubset \Gamma^0(3)$. A simple example is when $p$ is a prime number, then $[\Gamma^0(p) : \Gamma^0(p^k)] = p^{k-1}$ for $k \in \mathbb{N}$ is precisely the right index. However, if $\Gamma^0(p^k)$ can be folded to $\Gamma^0(p)$ depends also on the cusp structure and remains to be determined.

We can make the above isomorphism between $Z$ and a cyclic group more specific as follows: Let $\Gamma' \leqslant \Gamma$ be a subgroup, and denote by $\Gamma' : \Gamma$ the set of all right cosets. There is a group homomorphism

$$
H : \begin{cases} \Gamma \longrightarrow \mathrm{Sym}(\Gamma' : \Gamma) \\ g \longmapsto H(g) : \begin{cases} \Gamma' : \Gamma \to \Gamma' : \Gamma \\ \Gamma' \tilde{g} \mapsto \Gamma' \tilde{g} g \end{cases} \end{cases} \tag{B.8}
$$

from $\Gamma$ into the symmetric group on the coset space $\Gamma' : \Gamma$. Thus $H(g)$ is a permutation of elements in $\Gamma' : \Gamma$. For instance, $H(e) = \mathrm{id}_{\mathrm{Sym}(\Gamma':\Gamma)}$ is the identity permutation. The symmetric group $\mathrm{Sym}(\Gamma' : \Gamma) \cong S_{|\Gamma:\Gamma'|}$ contains a cyclic subgroup $\mathbb{Z}_{|\Gamma:\Gamma'|}$, which is however not necessarily in the image of $H$. This enables a definition: Let $\Gamma' \leqslant \Gamma$ be a subgroup. If there exists an element $g \in \Gamma$ such that $H(g) \cong \mathbb{Z}_{|\Gamma:\Gamma'|}$, then $\Gamma' \sqsubset \Gamma$.

**Foldings for non-modular domains.** The extension to non-modular configurations is rather simple: To any rational elliptic surface $\mathcal{S}$ we can associate a fundamental domain [6]

$$
\mathcal{F}(\mathcal{S}) = \bigcup_{\alpha \in C_{\mathcal{S}}} \alpha \mathcal{F} , \tag{B.9}
$$

where $\mathcal{F}$ is the standard $\mathrm{PSL}(2,\mathbb{Z})$ fundamental domain, and $C_{\mathcal{S}} \subset \mathrm{PSL}(2,\mathbb{Z})$ is a choice of cosets for $\mathcal{S}$. An important feature of non-modular configurations is the appearance of branch points and cuts in the fundamental domain. These can obstruct the foldings: Any branch point appears in the fundamental domains at least twice. If a branch point lies in the interior of the fundamental domain, it must be connected with a branch cut to another representative of the same branch point. The boundary of the fundamental domain consists of pairwise identified contours, and thus if a branch point lies on the boundary then there is another representative also on the boundary.

In order to obtain the correct descriptions for such $\mathbb{Z}_N$ covers, we first need to make sure that the fundamental domains of some given theory are consistent. In order to guarantee a consistent folding of $\mathcal{F}(\mathcal{S})$, we thus impose not only the same consistency on the coset representatives $C_{\mathcal{S}}$ as for the modular case, but also that all branch points in $\mathcal{F}(\mathcal{S})$ must be mapped to each other by the elements of $Z$, defined

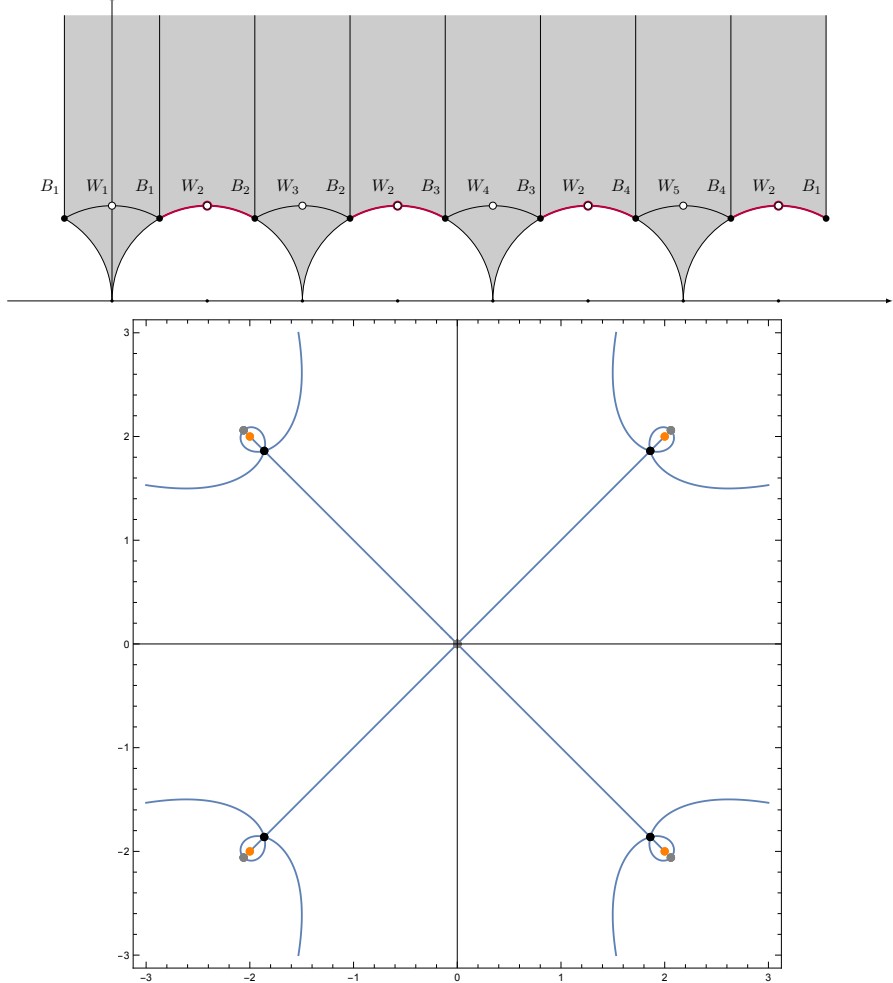

Figure 6: Top: Fundamental domain for $D_{S^1} E_1$ with $\lambda = -1$. The four purple dots are the branch points, which are identified under $U(\tau, -1)$. Bottom: Partition of the $U$-plane of the $D_{S^1} E_1$ theory at $\lambda = -1$. The $\mathbb{Z}_4$ symmetry $U \mapsto e^{\frac{\pi i}{2}} U$ is apparent.

analogous to (B.6). An example is the non-modular configuration $(I_3^*, 3I_1)$ of $N_f = 1$ SQCD, where the branch points are $\tau \in e^{2\pi i/3} + \mathbb{Z}$. These are all related by powers of $T$, which generates the folding in that case. Another example is the configuration $(I_8, 4I_1)$ of the $D_{S^1} E_1$ theory with $\lambda = -1$ (see Fig. 6). Only for this choice of $\lambda$, the branch points are identified under the generator of the folding.

With the consistency of the branch point kept in mind, the comparison of representatives $C_{\mathcal{S}}$ and $C_{\mathcal{S}'}$ for two rational elliptic surfaces $\mathcal{S}$ and $\mathcal{S}'$ proceeds as above. Since the branch points are identified under the generator of $Z$, this in particular means that the branch point is at the origin of the folding in the $U$-plane. See for instance Fig. 6 or Fig. 5 in [6].

The examples of $\Gamma' \sqsubset \Gamma$ which are of interest are those when $Z$ is generated by a power of $T$. Let $w_\Gamma$ be the smallest integer such that $T^{w_\Gamma} \in \Gamma$ for any subgroup

$\Gamma \leqslant \mathrm{PSL}(2,\mathbb{Z})$: This is the width of the cusp $i\infty$ of $\Gamma$. Then any proper subgroup $\Gamma' < \Gamma$ with $w_{\Gamma'} > w_\Gamma$ can have a folding generated by $T^{w_\Gamma} \in \Gamma$, whose $[\Gamma : \Gamma']$-th power $T^{w_{\Gamma'}}$ is in $\Gamma'$. For these cases, we find that the functional invariants $J_\Gamma$ associated with the subgroups $\Gamma$ satisfy

$$J_\Gamma(U^{[\Gamma:\Gamma']}) = J_{\Gamma'}(U) , \tag{B.10}$$

where we implicitly include also shifts and scalings in $U$. This is only possible if $J_{\Gamma'}$ is a rational function of $U^{[\Gamma:\Gamma']}$. In particular, it is invariant under $U \mapsto \zeta_{[\Gamma:\Gamma']}U$, where $\zeta_n$ is an $n$-th root of unity, $\zeta_n^n = 1$. Therefore, the action of $T^{w_{\Gamma'}}$ on $U$ is the map

$$U(\tau + w_{\Gamma'}) = \zeta_{[\Gamma:\Gamma']}U(\tau) . \tag{B.11}$$

To conclude this section, let us note that for the description of foldings from modular domains, we have implicitly used that the fixed point of the folding is a smooth fibre. If the fixed point itself is singular, the discussion needs to be extended. However, the characterisation (B.10) is agnostic to the fixed point of the folding and works in either case.

## C  Non-cyclic symmetries

The Coulomb branch automorphisms induced by automorphisms of the SW geometry are discrete subgroups of $\mathrm{Aut}(\mathbb{P}^1) \cong \mathrm{PGL}(2,\mathbb{Z})$ of order less than or equal to 12.[31] They are: the cyclic groups $\mathbb{Z}_n$ with $n \leq 12$, the dihedral groups[32] $D_n$ (of order $2n$) with $n = 2, 3, 4, 6$, and the tetrahedral group $A_4$. The family of RES supporting the non-cyclic $\mathrm{Aut}_\mathcal{S}(\mathbb{P}^1)$ is classified by their singular fibres, as given in [25, Proposition 4.2.3].

In this appendix, we discuss these non-cyclic symmetries, as well as their physical interpretation, and present some examples for the 5d $E_n$ theories. This extends the short discussion in Section 3.5.

### C.1  Möbius symmetries

We consider furthermore automorphisms of the Coulomb branch that are not induced by those of the elliptic surface $\mathcal{S}$. Such automorphisms can have distinct physical origins. Consider the four groups

$$\mathrm{Aut}_\mathcal{S}(\mathbb{P}^1) \subset \mathrm{Aut}_\mathcal{J}(\mathbb{P}^1) \subset \mathrm{Aut}_\Delta(\mathbb{P}^1) \subset \mathrm{Aut}(\mathbb{P}^1) , \tag{C.2}$$

---

[31]Automorphisms on $\mathbb{P}^1$ are bijective conformal maps.

[32]For future reference, recall that the dihedral group $D_n$ with $2n$ elements is the symmetry group of the regular $n$-gon and has a presentation

$$D_n = \langle r, s \,|\, r^n = s^2 = (sr)^2 = e \rangle , \tag{C.1}$$

such that it is isomorphic to $D_n \cong \mathbb{Z}_n \rtimes \mathbb{Z}_2$.

which we define in the following. The first, $\mathrm{Aut}_{\mathcal{S}}(\mathbb{P}^1)$, was defined as the auto-morphisms on the base $\mathbb{P}^1$ induced by all elements of $\mathrm{Aut}(\mathcal{S})$. This is, of course, subgroups of $\mathrm{PGL}(2, \mathbb{C})$ – i.e. the group of automorphisms of $\mathbb{P}^1$ – acting by Möbius transformations

$$\mathrm{Aut}(\mathbb{P}^1) \ni \varphi : U \mapsto \frac{aU + b}{cU + d} \,, \qquad \begin{pmatrix} a & b \\ c & d \end{pmatrix} \in \mathrm{PGL}(2, \mathbb{C}) \,. \qquad \text{(C.3)}$$

Due to the rationality condition of the rational elliptic surface $\mathcal{S}$, $\mathrm{Aut}_{\mathcal{S}}(\mathbb{P}^1)$ is finite and thus a finite subgroup of $\mathrm{Aut}(\mathbb{P}^1)$. Moreover, these automorphisms preserve the elliptic fibres, and thus the $\mathcal{J}$-invariant of $\mathcal{S}$ is preserved under such transformations.

We can furthermore study all Möbius transformations preserving the $\mathcal{J}$-invariant,

$$\mathrm{Aut}_{\mathcal{J}}(\mathbb{P}^1) = \left\{ \varphi \in \mathrm{Aut}(\mathbb{P}^1) \,|\, \mathcal{J}(\varphi(U)) = \mathcal{J}(U) \right\} \,. \qquad \text{(C.4)}$$

Clearly, this group contains $\mathrm{Aut}_{\mathcal{S}}(\mathbb{P}^1)$. However, in Section C.3 we find examples where it contains the latter as a strict subgroup: not all transformations preserving the $\mathcal{J}$-invariant of $\mathcal{S}$ are induced by automorphisms of $\mathcal{S}$. We will discuss these additional symmetries in examples below.

Finally, we can consider the symmetries preserving the CB singularities,

$$\mathrm{Aut}_{\Delta}(\mathbb{P}^1) = \left\{ \varphi \in \mathrm{Aut}(\mathbb{P}^1) \,|\, \mathcal{L}_{\Delta \circ \varphi} = \mathcal{L}_{\Delta} \right\} \,, \qquad \text{(C.5)}$$

where $\mathcal{L}_{\Delta}$ is the vanishing locus of the discriminant $\Delta$. Clearly, this group contains $\mathrm{Aut}_{\mathcal{J}}(\mathbb{P}^1)$, and indeed, in an example we show that it can also be a strictly larger group. One purpose of this appendix is to show that the four groups (C.2) are generally mutually distinct. We further hope that this structure (C.2) will clarify distinct notions of automorphisms of curves and surfaces related to Seiberg–Witten curves, and hope it will be beneficial in other contexts [5, 46, 103].

To illustrate the differences between these various groups, consider the $D_{S^1}E_5$ curve with the singular fibre configuration $\mathcal{S} : (I_4; 8I_1)$, found for $\chi_i = 0$ except for $\chi_2 = 37 + 24\sqrt{3}$. In this case, the discriminant locus $\mathcal{L}_{\Delta}$ has a $\mathrm{Aut}_{\Delta}(\mathbb{P}^1) = \mathbb{Z}_8$ symmetry [22]. However, the $U$-plane itself is only $\mathbb{Z}_4$ symmetric, that is, $\mathrm{Aut}_{\mathcal{S}}(\mathbb{P}^1) = \mathrm{Aut}_{\mathcal{J}}(\mathbb{P}^1) = \mathbb{Z}_4$. This can be checked for instance from the fact that the $\mathcal{J}$-invariant is rational in $U^{1/4}$, but not in $U^{1/8}$. Equivalently, the partitioning [6] of the $U$-plane will only have a $\mathbb{Z}_4$ symmetry, but no $\mathbb{Z}_8$ symmetry. These two distinct symmetries can be clearly seen in Fig. 7. This 'counterexample' also clarifies why it does not show up in Table 2, that is, why it is not a 'maximal' cyclic symmetry of the theory.

## C.2 Examples of non-cyclic symmetries

We have previously studied gaugings of the $\mathbb{Z}_n$ cyclic symmetries of the Coulomb branch. In general, one can also consider gauging non-cyclic abelian groups [33]. The only such group relevant for rank-one theories is the Klein four-group $D_2 =$

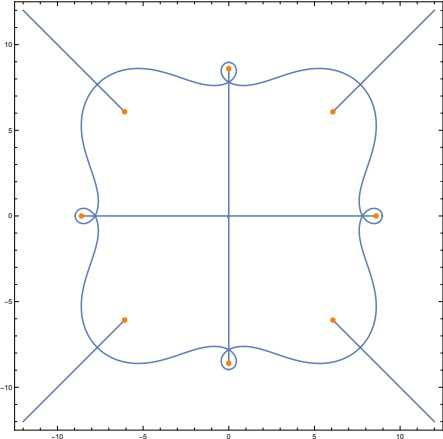

Figure 7: $D_{S^1}E_5$ configuration with $\mathbb{Z}_8$ symmetric singular locus (orange), while the $U$-plane partitioning itself only has a $\mathbb{Z}_4$ symmetry (blue).

$\mathbb{Z}_2 \times \mathbb{Z}_2$ [25]. This group acts by Möbius transformations on $\mathbb{P}^1$ and is generated by $u \mapsto 1/u$ and $u \mapsto -u$. We will denote these generators by $s = \left(\begin{smallmatrix} 0 & 1 \\ 1 & 0 \end{smallmatrix}\right)$ and $r = \left(\begin{smallmatrix} -1 & 0 \\ 0 & 1 \end{smallmatrix}\right)$, respectively, and refer to the corresponding symmetries as *inversion* and *reflection* symmetries.

There are many rational elliptic surfaces $\mathcal{S}$ whose induced automorphisms $\mathrm{Aut}_{\mathcal{S}}(\mathbb{P}^1)$ on the CB are this simplest non-cyclic abelian group. Indeed, this symmetry is realised for 18 distinct configurations [25]. Any such surface satisfies the property that $4 | \deg(J)$. The $\mathbb{Z}_2$ extension to the automorphism group preserving the zero-section is $\mathrm{Aut}_{\sigma}(\mathcal{S}) = D_4 = \mathbb{Z}_4 \rtimes \mathbb{Z}_2$ [25]. A discrete gauging of the 4d superconformal SU(2) SQCD with four flavours by the Klein four-group has recently been considered in [166].

The transformation $u \mapsto 1/u$ interchanging the fibre at infinity with a bulk singularity appears to be rather peculiar. Mathematically, these singular fibres are treated on equal grounds. For instance, if $F_\infty = I_n$, then this fibre must be interchanged by $u \mapsto 1/u$ to an equivalent $I_n$ fibre in the bulk. Indeed, the elliptic fibration $\mathcal{S} \to \mathbb{P}^1$ is obtained by a compactification of the $U$-plane to $\mathbb{P}^1$ by adding the point $U = \infty$. Physically, however, we distinguish the fibre at infinity $F_\infty$, as characterising the 'UV definition' of the theory, as explained around (2.7). Therefore, any automorphism that does not fix $F_\infty$ is not a 'proper' symmetry of the theory. As is clear from the examples below, this implies that any non-cyclic automorphism cannot be a discrete symmetry of a given theory.

In the remainder of this appendix, we rather want to contemplate the geometric structure if we do not pose this physical constraint, and thus consider abelian non-cyclic and also non-abelian symmetries.

### C.2.1 The Klein four-group

As an example, consider the $D_{S^1}E_5$ curve with $\lambda = 1$ and $M_j = i$ for all $j = 1,\ldots,4$. This curve is found by setting the $E_5$ characters to $\boldsymbol{\chi} = (-2,-3,0,8,0)$. This configuration has singular fibres $(I_4; I_4, 2I_2)$, the Mordell–Weil group $\mathrm{MW}(\mathcal{S}) = \Phi_{\mathrm{tor}}(\mathcal{S}) = \mathbb{Z}_4 \times \mathbb{Z}_2$, and is modular with monodromy group $\Gamma^0(4) \cap \Gamma(2)$ [22]. To study the $\mathbb{Z}_2 \times \mathbb{Z}_2$ symmetry, we rescale the CB parameter, $U = 4u$, such that the $\mathcal{J}$-invariant becomes

$$\mathcal{J}(u) = \frac{4\left(u^4 + u^2 + 1\right)^3}{27u^4\left(u^2 + 1\right)^2} \ . \tag{C.6}$$

This has the interesting property that the transformations $u \mapsto -u$ and $u \mapsto 1/u$ leave it unaffected. These two transformations are, of course, the generators of the Klein four-group previously mentioned. In fact, we have the stronger statement that $\mathrm{Aut}_{\mathcal{S}}(\mathbb{P}^1) = D_2$ [25], as can be seen from the partitioning of the $u$-plane sketched in Fig. 4a.

**Reflection quotient.** It is compelling to consider a quotient by this symmetry, even though an interpretation in terms of gauging of a discrete symmetry is more elusive. We can nevertheless consider taking quotients by the full automorphism $\mathbb{Z}_2^r \times \mathbb{Z}_2^s$, as well as quotients by either factor $\mathbb{Z}_2^r$, or $\mathbb{Z}_2^s$. For $\mathbb{Z}_2^r$ quotients, we identify $u$ with $-u$ by defining $x = u^2$. For the $\mathbb{Z}_2^s$ transformation, we identify $u$ with $1/u$ and we can thus consider $x = u + 1/u$. These identifications can give interesting patterns of quotients.

Consider first the quotient by $\mathbb{Z}_2^r$. This quotient is a $\mathbb{Z}_2$-folding of the $E_5$ curve, with the folded curve having the configuration of singular fibres $(I_2^\infty; I_2^*, I_2)$. Note that this is a $D_{S^1}E_7$ configuration, with monodromy group $\Gamma(2)$. The partitioning of the folded $u$-plane is shown in Fig. 8a.

**Inversion quotient.** The quotient by the $\mathbb{Z}_2^s$ symmetry makes use of the redefinition $x \equiv u + 1/u$, leading to the new curve with $\mathcal{J}$-invariant

$$\mathcal{J}(x) = \frac{4(x^2 - 1)^3}{27x^2} \ . \tag{C.7}$$

This map identifies the $I_4$ and the $I_4^\infty$ fibres, as well as the two bulk $I_2$'s. The smooth points $u = \pm 1$ are the self-dual points under $u \mapsto 1/u$, and they become in fact two $III$ singularities. Thus the singular structure is $(I_4^\infty; I_2, 2III)$, which is a particular configuration of the $D_{S^1}E_5$ theory. It is in fact a modular configuration, with monodromy group $4C^0$, in the notation of [167]. A fundamental domain is obtained in [14, Fig. 7]. The $x$-plane is shown in Fig. 8b.

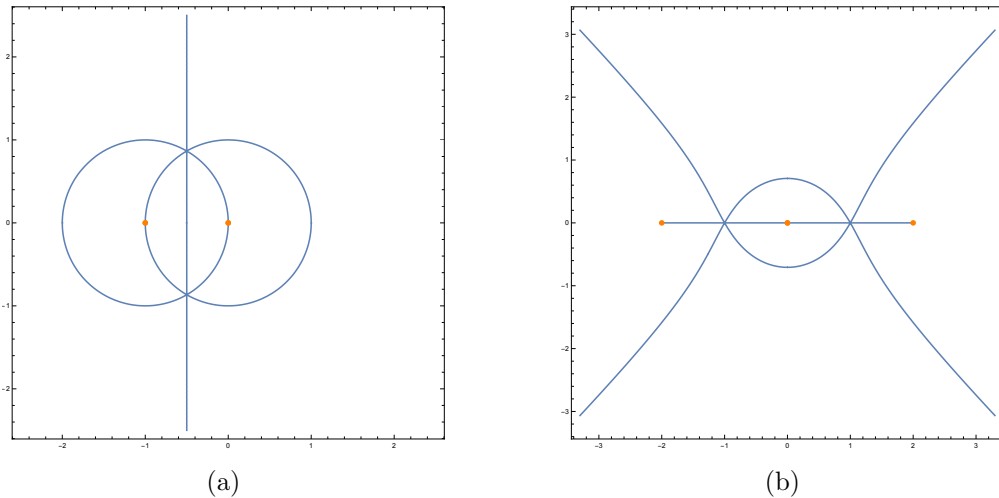

Figure 8: (a) $D_{S^1}E_7$ configuration obtained from a quotient of the $D_{S^1}E_5$ curve (C.6) by the $\mathbb{Z}_2^r$ symmetry. (b) $D_{S^1}E_5$ configuration obtained from a quotient of the same curve by the $\mathbb{Z}_2^s$ symmetry.

**Full quotient.** We can complete the quotient diagram by either taking a further $\mathbb{Z}_2^s$ or $\mathbb{Z}_2^r$ quotient of the previously two discussed quotients, respectively. The singular fibre configuration for the full quotient becomes $(I_2^\infty; I_1, III^*)$, which is again modular with monodromy group $\Gamma^0(2)$. The resulting CB partitioning is shown in Fig. 9. These quotients are summarised in Figure 10.

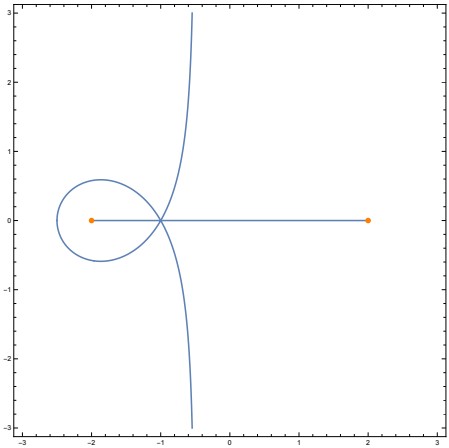

Figure 9: $u$-plane after taking the quotient by the $\mathbb{Z}_2 \times \mathbb{Z}_2$ symmetry of the $D_{S^1}E_5$ curve (C.6).

This diagram commutes up to an irrelevant shift of the order parameter. This is because the full quotient is realised in the two cases from the change in coordinates $u^2 + 1/u^2$, and $(u + 1/u)^2$, respectively.

### C.2.2 The tetrahedral group

The possible non-abelian groups $\mathrm{Aut}_{\mathcal{S}}(\mathbb{P}^1)$ are the dihedral groups $D_n$ $(n = 3, 4, 6)$ and the tetrahedral group $A_4$. Rational elliptic surfaces with tetrahedral symmetry

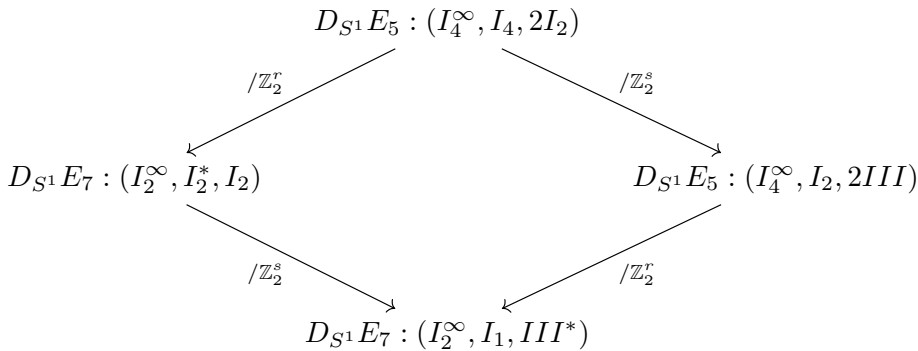

Figure 10: The quotients of the $D_{S^1}E_5$ configuration by all subgroups of its automorphism group $\mathrm{Aut}_{\mathcal{S}}(\mathbb{P}^1) = \mathbb{Z}_2^r \times \mathbb{Z}_2^s$, where $\mathbb{Z}_2^r : u \mapsto -u$ and $\mathbb{Z}_2^s : u \mapsto 1/u$.

are highly restricted: the possible singular configurations are $(4I_3)$ and $(12I_1)$. The latter, $(12I_1)$, is the generic configuration of the 6d $E$-string curve curve [15, 17].

Meanwhile, the configuration $(I_3; 3I_3)$ can be realised for the $D_{S^1}E_6$ curve with characters $\boldsymbol{\chi} = (0, 0, -3, 9, 0, 0)$, having $\Phi_{\mathrm{tor}}(\mathcal{S}) = \mathbb{Z}_3 \times \mathbb{Z}_3$. It is in fact a modular elliptic surface, with monodromy group $\Gamma(3)$. Due to the non-trivial torsion, this surface plays an important role in the context of Galois covers [31, 32]. In order to make the $A_4$ symmetry manifest, let us rescale $U = 4u$. Then we find that

$$\mathcal{J}(u) = \frac{\left(u^4 + 8u\right)^3}{2^6 \left(u^3 - 1\right)^3} , \tag{C.8}$$

is invariant under the Möbius transformations $g_1 : u \mapsto \omega_3 u$ and $g_2 : u \mapsto \frac{u+2}{u-1}$, with $\omega_3 = e^{2\pi i/3}$, which satisfy $g_1^3 = g_2^2 = \mathbb{1}$ and $g_1 g_2 g_1 = g_2 g_1^2 g_2$. Thus, these offer a presentation for the tetrahedral group $A_4$ (see also [25, Section 5.3.5]). The $u$-plane with $A_4$ symmetry is plotted in Fig. 4b.

The symmetry can be seen as follows: Connect the three bulk singularities $(1, \omega_3, \omega_3^2)$ (orange) by a line, then add the point infinity and embed the four points in the Riemann sphere $\mathbb{P}^1$. The $A_4$ symmetry is then the symmetry of the tetrahedron with vertices $P = (1, \omega_3, \omega_3^2, \infty)$. We can think of the action of $A_4$ on $\mathbb{P}^1$ as orientation-preserving transformations of this tetrahedron inside the Riemann sphere $\mathbb{P}^1$. Then $A_4$ permutes these vertices $P$ as follows: The $g_1$ action is a rotation by $\frac{2\pi}{3}$, having $u = 0$ and the point at infinity as fixed points. Meanwhile, the order 2 Möbius transformation acts as $g_2(P) = (\infty, \omega_3^2, \omega_3, 1)$, having $u = 1 \pm \sqrt{3}$ as fixed points.

In terms of the monodromy group $\Gamma(3)$, the $A_4$ symmetry is carried by the cosets, as can be seen from the fact that $\mathrm{SL}(2, \mathbb{Z})/\Gamma(3) \cong A_4$ (see Table A.29). We can take either quotients of the $\mathbb{Z}_3^{g_1} \subset A_4$ or $\mathbb{Z}_2^{g_2} \subset A_4$, but taking the quotient by one cyclic subgroup removes the other symmetry. This is due to the map implementing the full $A_4$ group not being a valid base change of the rational elliptic surface.

We note also that the finite group $A_4 \subset \mathrm{SU}(2)$ is associated with the $E_6$ Dynkin diagram through the McKay correspondence (see for instance [168]). This suggests that the $U$-plane symmetry $A_4$ is related to the flavour symmetry $E_6$. For this configuration of characters, the theory has flavour algebra $\mathfrak{g}_F = A_2 \oplus A_2 \oplus A_2$ [14]. This is a rank 6 lattice that embeds into $E_8$. The three copies of $A_2$ are each associated with the $I_3$ singularities, while the $I_3^\infty$ singularity at infinity does not contribute to the flavour symmetry. The $(4I_3)$ configuration is extremal, so $\mathrm{MW}(\mathcal{S})$ does not have a free part. One possible way to derive $A_4$ from the flavour symmetry is to study how the symmetry $\mathrm{SU}(3)^3/(\mathbb{Z}_3 \times \mathbb{Z}_3)$ is embedded in $E_6/\mathbb{Z}_3$. We leave it for future work to explore this in more detail.

## C.3 Non-induced symmetries

Returning to the inclusion sequence (C.2) of automorphisms groups, we defined in (C.4) the Möbius transformations $\mathrm{Aut}_{\mathcal{J}}(\mathbb{P}^1)$ preserving the $\mathcal{J}$-invariant. In this subsection, we briefly discuss the difference between this group and the smaller group of induced automorphisms $\mathrm{Aut}_{\mathcal{S}}(\mathbb{P}^1)$. The simplest examples can be found in 4-dimensional massless $\mathcal{N} = 2$ SQCD.

**Massless SU(2) $N_f = 2$.** Consider the massless SU(2) $N_f = 2$ SW curve with $\Lambda_2 = \sqrt{8}$,[33]

$$\mathcal{J}(u) = \frac{(u^2 + 3)^3}{27 (1 - u^2)^2} , \tag{C.9}$$

which has a configuration $(I_2^*; 2I_2)$ and is modular with monodromy group $\Gamma(2)$. The flavour symmetry is $\mathfrak{g}_F = A_1 \oplus A_1$. This curve is invariant under the Möbius transformations $m : u \mapsto \frac{u-3}{u+1}$ and $r : u \mapsto -u$.[34] We have $m^3 = r^2 = (mr)^2 = 1$, giving a presentation of the dihedral group $D_3$ of order 6 (see (C.1)). In this case, the $m$-transformation is induced by a modular transformation $ST : \tau \mapsto -\frac{1}{\tau+1}$, while the $r$-transformation is induced by $T : \tau \mapsto \tau + 1$. These can easily be proven on the level of the modular functions, as have been found in [40]. Similar to the $\Gamma(3)$ modular surface described in the previous section, where $\Gamma/\Gamma(3) \cong A_4$, here we have $\Gamma/\Gamma(2) \cong S_3 \cong D_3$. Thus in this case again the group of Möbius maps preserving the $\mathcal{J}$-invariant is given by the $\mathrm{SL}(2, \mathbb{Z})$ duality group. These duality transformations clearly permute the singularities $P = (1, -1, \infty)$: The map $r(P) = (-1, 1, \infty)$ interchanges the strongly coupled singularities $\pm 1$, while $m(P) = (-1, \infty, 1)$ rotates all three singularities.

---

[33]Or, alternatively, we can set $\Lambda_2 = 1$ and rescale $u$.

[34]The 'Möbius invariants' $\mathrm{Aut}_{\mathcal{J}}(\mathbb{P}^1)$ of rational functions $\mathcal{J}$ can be found as follows. The PGL$(2, \mathbb{C})$ transformation $\mathcal{J}(\frac{au+b}{cu+d})/\mathcal{J}(u)$ of $\mathcal{J}(u)$ is a new rational function $R(u)/Q(u)$ with numerator and denominator polynomials $R$ and $Q$. This quotient is equal to 1 if the polynomial $R(u) - Q(u)$ is identically zero. This gives a system of $\deg(\mathcal{J}) + 1$ polynomial equations for $a, b, c, d$, subject to the constraint $ad - bc \neq 0$.

The elliptic surface $\mathcal{S}$ for massless SU(2) $N_f = 2$ thus seems to enjoy a 'large' symmetry. However, by the classification of [25], $D_3$ symmetries can never be induced by automorphisms of $\mathcal{S}$ if $\deg J = 6$ (see p. 17). Indeed, the only RES with $D_3 = \text{Aut}_\mathcal{S}(\mathbb{P}^1)$ are $(3I_3, 3I_1)$, $(3I_2, 6I_1)$, $(6I_1)$ and $(12I_1)$, all of $\deg J = 12$. Furthermore, the only possible order $n$ of induced automorphisms on $\mathbb{P}^1$ for the surface $(I_2^*; 2I_2)$ is $n = 2$ [25, Tables 2 and 11]. This is precisely the symmetry $u \mapsto -u$. It is clear that the extra symmetry cannot originate from the surface $\mathcal{S}$ itself: It exchanges necessarily the $I_2^*$ with some $I_2$, which is not an automorphism.

To conclude, in the example of massless $N_f = 2$ the $D_3$ symmetry decomposes as $D_3 \cong \mathbb{Z}_3 \rtimes \mathbb{Z}_2$. The $\mathbb{Z}_2$ is due to the non-anomalous $R$-symmetry, and is an induced symmetry of the surface. Meanwhile, there is an extra $\mathbb{Z}_3$ 'duality' symmetry, which does not originate from automorphisms of $\mathcal{S}$. We see that the group $\text{Aut}_\mathcal{J}(\mathbb{P}^1)$ can exceed $\text{Aut}_\mathcal{S}(\mathbb{P}^1)$, as in this example we have $\text{Aut}_\mathcal{J}(\mathbb{P}^1) = D_3 \supset \mathbb{Z}_2 = \text{Aut}_\mathcal{S}(\mathbb{P}^1)$.

**Massless SU(2) $N_f = 3$.** We can check explicitly that for $N_f = 0$, as well as massless $N_f = 1$, there is no $U$-plane symmetry other than the expected $\mathbb{Z}_2$ and $\mathbb{Z}_3$ cyclic symmetry, respectively. For massless $N_f = 3$ with $\Lambda_3 = 16$, we have

$$\mathcal{J}(u) = -\frac{(u^2 - 16u + 16)^3}{108(u-1)u^4} .\qquad(\text{C.10})$$

It is the modular RES with singularities $(I_1^*, I_4, I_1)$ and monodromy group $\Gamma_0(4)$. The flavour symmetry is $\mathfrak{g}_F = A_3$ due to the $I_4$ singularity, and there is no residual action of the $R$-symmetry on the Coulomb branch [2].

However, one easily shows that $\mathcal{J}(\frac{u}{u-1}) = \mathcal{J}(u)$. This Möbius transformation generates a $\mathbb{Z}_2$ automorphism, since $g^2 = \mathbb{1}$ for $g = \left(\begin{smallmatrix} 1 & 0 \\ 1 & -1 \end{smallmatrix}\right)$. It interchanges the cusps $u = \infty$ and $u = 1$, while leaving the $I_4$ singularity $u = 0$ invariant. This automorphism of $\mathbb{P}^1$ is again not induced by the surface $\mathcal{S}$ [25, Table 3]: For any surface with singular configuration $(I_1^*, I_4, I_1)$, the order $n$ of the induced automorphism is $n = 1$. The situation is yet again different here to massless $N_f = 2$. Indeed, this configuration is modular for $\Gamma_0(4)$, but $\Gamma_0(4)$ is not a normal subgroup of $\text{SL}(2, \mathbb{Z})$, thus the quotient $\Gamma/\Gamma_0(4)$ is not well-defined and consequently not isomorphic to an order 6 group. Rather, it is apparent that the duality group is reduced in this case to a $\mathbb{Z}_2$. Indeed, we can check, starting from the solution[35]

$$u(\tau) = -4\frac{\vartheta_3(\tau)^2 \vartheta_4(\tau)^2}{(\vartheta_3(\tau)^2 - \vartheta_4(\tau)^2)^2} ,\qquad(\text{C.11})$$

that the $S^{-1}T^{-2}S$ transformation $\tau \mapsto \frac{\tau}{2\tau+1}$ is encoded in the Möbius map

$$u\left(\frac{\tau}{2\tau+1}\right) = \frac{u(\tau)}{u(\tau) - 1} .\qquad(\text{C.12})$$

---

[35]See Appendix A.3 for a definition of the Jacobi theta functions and their transformation properties

One can easily show that this is not true for the other four cosets $S$, $ST$, $ST^{-1}$ and $ST^{-2}$ of $\Gamma_0(4)$ in $\mathrm{SL}(2,\mathbb{Z})$, which in this case correspond to the four expansions at the $I_4$ cusp $u = 0$. These four $\mathrm{SL}(2,\mathbb{Z})$ transformations correspond to non-rational functions of $u$.

**Duality symmetries.** We can understand this more generally: From the perspective of the degree $\mathrm{ord}(\mathcal{J})$ equation (2.5) associated with a generic surface $\mathcal{S}$ with functional invariant $\mathcal{J}$, this picture is quite clear: The $\mathrm{ord}(\mathcal{J})$ zeros of the polynomial $P_{\mathcal{S}}$ correspond to the expansions of $u(\tau)$ at all cusps, and any non-trivial map $m$ with the property $\mathcal{J}(m(u)) = \mathcal{J}(u)$ necessarily permutes the zeros of $P_{\mathcal{S}}$. The cusp expansions are of course related by modular duality transformations [6]. If $m$ is a Möbius transformation, then there exists a corresponding $\mathrm{SL}(2,\mathbb{Z})$ duality transformation inducing that Möbius transformation on the base $\mathbb{P}^1$. As has been observed in [118, 119], clearly not every $\mathrm{SL}(2,\mathbb{Z})$ transformation induces a Möbius transformation on the base.

We then have some evidence that any such Möbius transformation is an electric-magnetic duality transformation. When $u \to \zeta_n u$ cyclic symmetry, it originates from an (accidental) $R$-symmetry and can be induced by a $T$-transformation. Möbius transformations that are pure translations $u \mapsto u + b$ can never be symmetries of $\mathcal{J}$, since they do not leave the singularities invariant. Then any non-cyclic Möbius transformation is induced by a duality transformation, and involves an $S$-transformation.

The fundamental domain for a surface $\mathcal{S}$ is given by a set of coset representatives. In general, they do not form a group [6]. When the rational function $\mathcal{J}$ has a Möbius symmetry $\mathrm{Aut}_{\mathcal{J}}(\mathbb{P}^1)$, this exchanges some of the coset representatives. Since the action of Möbius transformations gives a group, the subset of relevant coset representatives form a group. In the example of massless $N_f = 3$, the coset representatives $\mathbb{1}$ and $S^{-1}T^{-2}S$ form a $\mathbb{Z}_2$ group, since $(S^{-1}T^{-2}S)^2 \in \Gamma_0(4)$ and thus is identified with the identity representative $\mathbb{1}$.

# D   Seiberg–Witten curves

In this Appendix, we list some explicit expressions for the Seiberg–Witten curves we study in the body of the paper.

**4d SU(2) SQCD.** The Seiberg–Witten surfaces for 4d $\mathcal{N} = 2$ supersymmetric QCD with $N_f$ massive fundamental hypermultiplets are given by [2]:

$$
\begin{aligned}
N_f = 0 : \quad & y^2 = x^3 - ux^2 + \frac{1}{4}\Lambda_0^4 x \ , \\
N_f = 1 : \quad & y^2 = x^2(x - u) + \frac{1}{4}m\Lambda_1^3 x - \frac{1}{64}\Lambda_1^6 \ , \\
N_f = 2 : \quad & y^2 = (x^2 - \frac{1}{64}\Lambda_2^4)(x - u) + \frac{1}{4}m_1 m_2 \Lambda_2^2 x - \frac{1}{64}(m_1^2 + m_2^2)\Lambda_2^4 \ , \quad \text{(D.1)} \\
N_f = 3 : \quad & y^2 = x^2(x - u) - \frac{1}{64}\Lambda_3^2(x - u)^2 - \frac{1}{64}(m_1^2 + m_2^2 + m_3^2)\Lambda_3^2(x - u) \\
& \quad + \frac{1}{4}m_1 m_2 m_3 \Lambda_3 x - \frac{1}{64}(m_1^2 m_2^2 + m_2^2 m_3^2 + m_1^2 m_3^2)\Lambda_3^2 \ .
\end{aligned}
$$

See also footnote 18 for an important comment on the normalisation for $N_f = 3$.

$E_n$ **theories.** The Seiberg–Witten curves for the $D_{S^1} E_n$ theories were studied in [15, 17, 79–90]. Here, we list the Weierstraß invariants for the toric $E_n$ curves, $n = 0, \ldots, 3$:

$$
\begin{aligned}
g_2^{E_0}(U) &= \frac{3}{4}U\left(9U^3 - 8\right) \ , \\
g_3^{E_0}(U) &= \frac{1}{8}\left(-27U^6 + 36U^3 - 8\right) \ , \\
g_2^{E_1}(U) &= \frac{1}{12}\left(16\lambda^2 - 16\lambda + U^4 - 8\lambda U^2 - 8U^2 + 16\right) \ , \\
g_3^{E_1}(U) &= -\frac{1}{216}\left(-4\lambda + U^2 - 4\right)\left(16\lambda^2 - 40\lambda + U^4 - 8\lambda U^2 - 8U^2 + 16\right) \ , \\
g_2^{\tilde{E}_1}(U) &= \frac{1}{12}\left(U^4 - 8U^2 - 24\lambda U + 16\right) \ , \\
g_3^{\tilde{E}_1}(U) &= -\lambda^2 + \frac{1}{6}\lambda U\left(U^2 - 4\right) - \frac{1}{216}\left(U^2 - 4\right)^3 \ , \\
g_2^{E_2}(U) &= \frac{1}{12}\left(16\lambda^2 - 16\lambda - 24\lambda M_1 U + U^4 - 8\lambda U^2 - 8U^2 + 16\right) \ , \quad \text{(D.2)} \\
g_3^{E_2}(U) &= \frac{1}{216}\left(64\lambda^3 - 24\lambda^2\left(9M_1^2 + 6M_1 U + 2U^2 + 4\right) \right. \\
& \quad \left. + 12\lambda\left(U^2 - 4\right)\left(3M_1 U + U^2 + 2\right) - \left(U^2 - 4\right)^3\right) \ , \\
g_2^{E_3}(U) &= \frac{1}{12}\left(16\left(\lambda^2\left(M_1^2 M_2^2 - M_1 M_2 + 1\right) - \lambda(M_1 M_2 + 1) + 1\right) - \right. \\
& \quad \left. 8U^2(\lambda + \lambda M_1 M_2 + 1) - 24\lambda U(M_1 + M_2) + U^4\right) \ , \\
g_3^{E_3}(U) &= \frac{1}{216}\left(-24U^2\left(\lambda + \lambda^2\left(2M_1^2 M_2^2 + M_1 M_2 + 2\right) + \lambda M_1 M_2 + 2\right)\right. \\
& \quad - 8\left(3\lambda^2\left(M_1^2\left(4M_2^2 + 9\right) + 2M_1 M_2 + 9M_2^2 + 4\right)\right. \\
& \quad \left. - 4\lambda^3\left(2M_1^3 M_2^3 - 3M_1^2 M_2^2 - 3M_1 M_2 + 2\right) + 12\lambda(M_1 M_2 + 1) - 8\right)
\end{aligned}
$$

$$+ 12U^4(\lambda + \lambda M_1 M_2 + 1) + 36\lambda U^3(M_1 + M_2)$$
$$- 144\lambda U(M_1 + M_2)(\lambda + \lambda M_1 M_2 + 1) - U^6\Big) \ .$$

In the Mathematica notebook [56], we furthermore give the explicit curves for the non-toric $D_{S^1}E_n$ theories.

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
