# Peer review of "Coulomb branch surgery: Holonomy saddles, S-folds and discrete symmetry gaugings"

_SciPost Physics_

## Round 1 · Referee Report · Anonymous (Referee 1) · 2024-6-29

Report

This paper explores the symmetry of Seiberg-Witten geometry of the Coulomb branch of 4d N=2 theories, including 4d N=2 KK theories compactified from 5d and 6d. It extensively utilizes the mathematical results on the classification of automorphisms of rational elliptic surfaces found in references [25] and [26], providing physical interpretations and implications for rank-1 Seiberg-Witten geometry. Particularly, it studies the folding of the Coulomb branch with respect to the symmetry of the U-plane induced by the automorphism of the rational elliptic surface. Depending on the behavior of the fiber at infinity​ under the quotient, the authors explore two physical scenarios: 5d (fractional) S-folding and 5d/6d holonomy saddles. Notably, the latter provides a new perspective on the non-toric del-Pezzo BPS quiver. By further including the Mordell-Weil lattice, the authors interpret the structure of the entire automorphism of the rational elliptic surface in terms of mixed anomalies of 0-form and 1-form symmetries.
The paper is well-written and self-contained. I recommend its publication in SciPost Physics after addressing the following requested changes.

Requested changes

Below, I point out some minor changes related to phrasing and clarity:

  1. The term "surgery" used in this paper should be clearly distinguished from surgery operations in surgery theory in algebraic topology. Please add a brief explanation or note to avoid confusion.

  2. Please change the phrasing "the rational sections \beta: S to \mathbb{P}^1 of S" in the first two lines of Appendix A.2 to "the rational sections of \beta: S to \mathbb{P}^1 ". The same request applies to the phrasing "the rational sections \beta: S to \mathbb{P}^1 " above Eq. (A.14). The point is that \beta: S to \mathbb{P}^1 represents the elliptic fibration rather than the sections.

  3. Please correct the typo in the second row, the first column of Table (A.29) from \Gamma/\Gamma(n) to \Gamma(1)/\Gamma(n). Similarly, correct the \Gamma/\Gamma(n) in the first row below Table (A.29).

  4. Before Eq. (A.23), please explicitly state that " \Gamma(1) = SL(2, \mathbb{Z})". Consequently, the text "Let \Gamma(1) = SL(2, \mathbb{Z})" above Eq. (B.5) can be deleted to avoid repetition. Given the repeated interchange between \Gamma(1) and SL(2, \mathbb{Z}) in Appendix A.3 and Appendix B, I suggest the authors stick to one notation as consistently as possible to improve readability.

Recommendation

Ask for minor revision

---

## Round 1 · Referee Report · Anonymous (Referee 2) · 2024-7-13

Report

In this paper, the authors discuss the automorphism of Seiberg-Witten geometries based on the complete classification of automorphisms of rational elliptic surfaces (RES) for rank-one theories. The automorphism group of RES takes the form of a semi-direct product of the Coulomb branch symmetry and the Mordell-Weil group. The primary objective is to examine the quotients of Seiberg-Witten curves of rank-one theories in four, five, and six dimensions with eight supercharges. The paper presents a comprehensive and systematic framework for analyzing the quotients of Seiberg-Witten geometry by its automorphisms, based on the complete classification of automorphisms of rational elliptic surfaces. The article offers new insights into the interpretation of the quotients in connection with the S-folds of 5d superconformal theories or particular quotients of (p,q) 5-brane webs, folding across dimensions, and mixed 't Hooft anomalies originating from the semi-direct product structure of the automorphism group.

The article is well-written and presents new insights into the Coulomb branch surgeries and the automorphisms of Seiberg-Witten geometries. However, the authors' analysis remains challenging for higher-rank theories due to the lack of a complete classification of automorphisms of Seiberg-Witten geometries. Nevertheless, the paper contains significant potential for further implications for uncovering various BPS quivers/spectra. Therefore, I recommend this paper for publication, provided the following minor points are addressed or updated.

Requested changes

  1. About the proposal, presented in Eq. (4.5), that circle compactification of the 5d SU(2) superconformal theories of $E_{2N_f+1}$ flavour symmetry leads in the IR to 4d SU(2) SQCD with $N_f$ flavours, the authors presented examples of $D_{S^1}E_{2N_f+1}~(N_f=0,1,2,3)$ in a systematic way by discussing the corresponding BPS quivers. To ensure consistency with the cases for $N_f=0,1,2$, it would be good to present a basis of light BPS states for the $N_f=3$ case.

  2. In section 4.3, the authors discussed the rank-one E-string and the rank-one M-string, along with their 6d-4d foldings. In the case of the M-string theory, the corresponding BPS quiver is presented in the reference [24], which appears to be the same BPS quiver as that given in Eq. (4.16) but with double arrows instead of single arrows. It is well-known that the BPS quiver for 4d $\mathcal{N}=2^*$ theory is composed of three nodes with double arrows. For instance, see Eq.(3.7) in [2308.10225]. Therefore, this evidence also lends support to the proposal (4.5). If this is indeed the case, it is reasonable to anticipate that the M-string theory will also exhibit a $\mathbb{Z}_6$ symmetry, as observed in the $D_{S^1}E_{3}$ theory. On the other hand, it is not clear how to see this $\mathbb{Z}_6$ symmetry. It would be good if the authors elaborate on this point. Also, it would be beneficial to include the BPS quiver for the M-string theory, which supports the proposal (4.5).

  3. A bit more explanation on the 'magnetic' theory discussed in section 5.2 would be helpful for the readers to understand why the proposal will not apply.

  4. In page 49, in Eq. (5.10), an inner bracket $``)"$ on the right-hand side is missing: \begin{align} P'_{\mathbb{Z}_2}= \bigg( \frac{1}{24} \Big(-U^2+4(1-6\sqrt{\lambda}+\lambda)\Big)\,,\,0\bigg)\ \end{align}

Recommendation

Ask for minor revision

---

## Editorial Decision

resubmitted